# Special Observing Period (SOP) Data for the Year of Polar Prediction site Model Intercomparison Project (YOPPsiteMIP)

Zen Mariani[1], Sara M. Morris[2,11], Taneil Uttal[2], Elena Akish[3,2], Robert Crawford[1], Laura Huang[1], Jonathan Day[4], Johanna Tjernström[12], Øystein Godøy[12], Lara Ferrighi[12], Leslie M. Hartten[3,2], Jareth Holt[6], Christopher J. Cox[2], Ewan O'Connor[9], Roberta Pirazzini[9], Marion Maturilli[13], Giri Prakash[10], James Mather[8], Kimberly Strong[5], Pierre Fogal[5], Vasily Kustov[7,14], Gunilla Svensson[6], Michael Gallagher[3,2], Brian Vasel[11]

[1]Meteorological Research Division, Environment and Climate Change Canada, Toronto, Canada
[2]NOAA Physical Sciences Laboratory, Boulder, CO, USA
[3]Cooperative Institute for Research in Environmental Science, University of Colorado, Boulder, Colorado, USA
[4]European Centre for Medium-Range Weather Forecasts, Reading, UK
[5]Department of Physics, University of Toronto, Toronto, Canada
[6]Department of Meteorology, Stockholm University, Sweden
[7]Arctic and Antarctic Research Institute, Air-sea interaction department, St. Petersburg, Russia
[8]Pacific Northwest National Laboratory, Richland, WA, USA
[9]Finnish Meteorological Institute, Finland
[10]Environmental Sciences Division, Oak Ridge National Laboratory, Oak Ridge, TN, USA
[11]NOAA Global Monitoring Laboratory, Boulder, CO, USA
[12]Norwegian Meteorological Institute, Norway
[13]Alfred Wegener Institute, Helmholtz Centre for Polar and Marine Research, Potsdam, Germany
[14]Freelance entrepreneur, Belgrade, Serbia

*Correspondence to*: Zen Mariani (zen.mariani@ec.gc.ca) and Sara Morris (Sara.Morris@noaa.gov)

**Abstract.** The rapid changes occurring in the polar regions require an improved understanding of the processes that are driving these changes. At the same time, increased human activities such as marine navigation, resource exploitation, aviation, commercial fishing, and tourism require reliable and relevant weather information. One of the primary goals of the World Meteorological Organization's Year of Polar Prediction (YOPP) Project is to improve the accuracy of numerical weather prediction (NWP) at high latitudes. During YOPP, two Canadian supersites were commissioned and equipped with new ground-based instruments for enhanced meteorological and system process observations. Additional pre-existing supersites in Canada, the United States, Norway, Finland, and Russia also provided data from ongoing long-term observing programs. These supersites collected a wealth of observations that are well-suited to address YOPP objectives. In order to increase data useability and station interoperability, novel Merged Observatory Data Files (MODFs) were created for the seven supersites over two Special Observing Periods (February to March 2018 and July to September 2018). All observations collected at the supersites were compiled into this standardized NetCDF MODF format, simplifying the process of conducting pan-Arctic NWP verification and process evaluation studies. This paper describes the seven Arctic YOPP supersites, their instrumentation, data collection and processing methods, the novel MODF format, and examples of the

observations contained therein. MODFs comprise the observational contribution to the model intercomparison effort, termed YOPP supersite Model Intercomparison Project (YOPPsiteMIP). All YOPPsiteMIP MODFs are publicly accessible via the YOPP Data Portal (Whitehorse: https://doi.org/10.21343/a33e-j150, Iqaluit: https://doi.org/10.21343/yrnf-ck57, Sodankylä: https://doi.org/10.21343/m16p-pq17, Utqiaġvik: https://doi.org/10.21343/a2dx-nq55, Tiksi: https://doi.org/10.21343/5bwn-w881, Ny-Ålesund: https://doi.org/10.21343/y89m-6393, Eureka: https://doi.org/10.21343/r85j-tc61), hosted by MET Norway, with corresponding output from NWP models.

## 1 Introduction

In the Arctic there is a recognized lack of process-level information supplementing meteorological observations to characterize the atmosphere and the cryosphere for operational forecasting (Cassano et al., 2011; Illingworth et al., 2015; Lawrence et al., 2019). As the climate continues to change, information on weather and climate is becoming more critical in ensuring the health and safety of local communities. Unfortunately, climate models do a poor job of capturing key features of Arctic climate, such as the Arctic amplification factor, likely as a result of inaccurate representation of key physical processes, as shown by Rantanen et al. (2022). Similarly, the accuracy of weather forecasts in the Polar Regions is also lower than in mid-latitudes (Jung et al., 2016) partly due to the scattered and limited availability of observing networks (Lawrence et al., 2019). Advances in Polar weather forecast prediction are expected to improve weather forecasts and climate predictions elsewhere (Jung et al., 2016 and Day et al., 2019), but understanding the causes of poor model performance in the Arctic is limited by the availability of observatory data. Data from observatories, where sometimes hundreds of parameters are measured, are needed for detailed investigations into the cause of model error, such as boundary-layer processes and turbulent exchanges (e.g., Day et al., 2024).

To address the need to improve Numerical Weather Prediction (NWP) performance in the Polar Regions, the World Meteorological Organization (WMO) launched the international Polar Prediction Project with its flagship activity, the Year of Polar Prediction (YOPP). During YOPP's core phase, from mid-2017 to mid-2019, several intensive observing periods were conducted with close coordination between the international network of polar observatories and weather forecast centers. The aim was to produce highly-concentrated sets of observed and modelled data for supporting forecast evaluation and process studies (Koltzow et al., 2019; Goessling et al., 2016; Jung et al., 2016).

One of the flagship activities of YOPP was the YOPP supersite Model Intercomparison Project (YOPPsiteMIP), an initiative to assess the performance of NWP systems at the process level by comparing with observatory data (Day et al., 2024). To achieve this, a dataset of weather forecasts was produced by various NWP centers for supersite locations. In the Arctic the dataset covers two Special Observing Periods (SOPs), SOP1 (February 1 – March 31, 2018) and SOP2 (July 1 – September 30, 2018). During this period the number of routine observations (e.g. radiosonde launches, buoy deployments, etc.) were

enhanced in the Arctic (doubled in the case of radiosondes), field campaigns were conducted, and enhanced observations from the designated YOPP "supersite" observatories were taken. In general, the suite of several additional instruments that enable an enhanced measurement program, including remote sensing, radiation, and other meteorological sensors, is what distinguishes a 'supersite' from a typical weather site. This paper documents the efforts to compile the supersite (hereafter referred to as "sites") data collected during this period as part of the YOPPsiteMIP. These sites (Figure 1) are distributed over a diverse range of geographical locations capturing some of the diversity in the terrestrial high-latitude climate zones.

Prior to YOPP, data collection, processing, geophysical variable reporting cadences, and file output type and format were not standardized across the sites, which are operated by different international agencies and consortiums. This lack of interoperability made performing multi-site comparisons, evaluations, and process studies difficult and time consuming, deterring potential users of the data (Wohner et al., 2022). In order to address this problem, the concept of standardized Merged Observatory Data Files (MODFs) was developed as part of the YOPPsiteMIP (Uttal et al., 2024). This concept is based on combining measurements from multiple international research observatories' instruments into a single NetCDF file that complies with established data management standards. Prior to MODFs, there generally existed no standardized procedures for coordinated data management at these research sites such as those that have been developed for operational datasets. Thus, the data from these sites' separate instruments were scattered between separate files with different authors, formats, metadata, post-processing techniques, physical archive locations, and requirements for usage. As such, they could not be amalgamated to provide a pan-Arctic observational dataset.

MODF files bring together observations from different earth system components in a standardized NetCDF file format to enable utilization of research-grade, process-level observations for model evaluation and parameterization development. At the same time, MODFs are compatible with and mirror Merged Model Data Files (MMDFs) that are produced by each NWP centre participating in YOPP (Day et al., 2024). Each geophysical variable observed at a site is matched to its corresponding NWP model geophysical variable using a standardized data format, cadence, and file structure. Uttal et al. (2024) provides a generalized overview for the content and data structure of MODFs, i.e., a single NetCDF data file containing measurements from multiple sources, and a series of tools to facilitate their creation. Table 1 provides information regarding the on-site facility location where measurements were collected and their coordinates for reference. For some sites (e.g., Sodankylä), certain geophysical variables are measured at multiple locations; these are all reported in the MODF with their corresponding measurement coordinates embedded within the file so as to distinguish each measurement. Final DOIs for the $MODF_{yms}$ are listed in Table 2.

The MODF's standardized file structure directly aligns with the NWP's MMDFs. Thus, MODFs easily facilitate observation-model comparisons at any/all of the seven sites (Gallagher and Tjernström, 2024). The purpose of the present work is to

describe the construction and contents of MODFs for seven of the YOPP-designated Arctic sites during SOPs 1 and 2 (hereafter, "MODF$_{ysm}$"): Whitehorse, Canada (60.71 °N, 135.07 °W, 682 m a.s.l.); Iqaluit, Canada (63.74 °N, 68.51 °W, 11 m a.s.l.); Sodankylä, Finland (67.367 °N, 26.629 °E, 179 m a.s.l.); Utqiaġvik (Barrow), Alaska (71.325 °N, 156.625 °W, 8 m a.s.l.); Tiksi, Russia (71.596 °N, 128.889 °E, 30 m a.s.l.); Ny-Ålesund, Norway (78.923 °N, 11.926 °E, 15 m a.s.l.); and Eureka, Canada (80.083 °N, 86.417 °W, 89 m a.s.l.). Methods used to organize a site's dataset and develop MODFs are provided. Each sites' instrumentation and data processing are also described in this work to provide users with additional context and information about the source of the geophysical variables contained in the MODF. The MODFs' counterpart, MMDFs, are described in Uttal et al. (2024).

Creating a standardized dataset such as MODF that contains observations from different meteorological and research agencies' sites is an extremely complex, non-trivial task. For the sake of brevity and to reduce redundancy, this paper references site- or instrument-specific publications in order to fully describe all of the aspects of the MODF dataset, including instrumentation, quality control, and processing techniques. In the case where non-trivial aspects about the MODF data arise, the data's origin, reference publications (e.g., dataset dois), and site contacts have been provided. Section 2 describes the data processing chain conducted at each site, including information about the site's local topography, climate, and instrumentation in order to provide site-specific context to aid the interpretation of model-observation comparisons. Section 3 describes the instrumentation and calculated variables. Section 4 describes the standardized MODF dataset file format, quality control, and post processing, which in some cases differed slightly from site-to-site. Section 5 describes the MODF data structure, attributes, and example Figures that illustrate the available dataset. Data and code availability is provided in Section 6, and concluding remarks are provided in Section 7.

## 2 Site Descriptions

To properly contextualize and interpret the observations contained within the MODF since they come from vastly different sites. A map of the distribution of the sites is shown in Figure 1. While all sites are also designated surface synoptic observation (SYNOP) stations, the meteorological data provided in the MODFs is significantly more detailed and includes additional geophysical variables and thus is not the same as the SYNOP data. Table 3 lists the geophysical variables observed at each site that are stored in the standardized MODF format, their measurement location(s), and other attributes; the MODF featureType corresponds to the type of geophysical variable being observed at each site (they are split up into broad categories).

### 2.1 Whitehorse, Canada

The Whitehorse site (Figure 2) was commissioned as part of the Canadian Arctic Weather Science (CAWS) project (Mariani
et al., 2018; Joe et al., 2020). CAWS was initiated to evaluate upper air observing technologies that can complement and
improve Polar forecasts, perform satellite calibration / validation over Arctic terrain, and to provide recommendations to
optimize the Canadian Arctic observing network. The site's instruments (Figure 2 and Table 4) are installed on an elevated
platform, all within a few meters of each other. The site is located at the Erik Nielsen Whitehorse International Airport, which
is situated on a plateau ~50 m above the rest of the city. The city is located in a valley between the Yukon Ranges to its West
(~1.6 km a.s.l.) and East (~1.4 km a.s.l.); this complex mountainous terrain strongly influences the weather systems that reach
Whitehorse, which mostly originate from the Eastern Pacific or over Alaska.
Whitehorse experiences cold to temperate average monthly temperatures ranging from -15 to 14 ℃ (annual mean of -2 ℃)
and average monthly precipitation ranging from 7 to 38 mm (annual total of ~500 mm). Since the city is in the rain shadow of
the Coast Mountains, precipitation totals are relatively low year-round. The primary surface wind direction follows the valley
(NNW). The soil type at and around the site is a mixture of grained alluvial and colluvial slopes and, as part of the Boreal
Cordillera ecozone, the surface type is primarily Boreal Forest, including complex plateaus, mountains, valleys and Cordilleran
vegetation. With a population greater than 26,000 inhabitants, Whitehorse is the primary gateway for air traffic for all of the
Yukon Territories, parts of Alaska, and the Western Canadian Arctic. During the YOPP SOPs, radiosondes were launched
four times daily.
**2.2 Iqaluit, Canada**
Like Whitehorse, the Iqaluit site (Figure 3) was commissioned as part of the CAWS project (Mariani et al., 2022). The site is
located ~200 m from the airport runway and all instruments (Figure 3 and Table 5) are co-located to within no more than 140
m of each other on flat terrain. The city itself is located along the coast in a valley that runs in the NW to SE direction; thus,
the primary direction of surface winds, which are frequently severe (> 15 ms$^{-1}$), follows this direction. The surrounding region
is relatively flat Arctic tundra except for nearby hills (~300 m a.s.l.) approximately two kilometers to the NE of the site.
Iqaluit experiences an extreme range of average monthly temperatures ranging from -28 to 8 ℃ (annual mean of -9 ℃) and
average monthly precipitation ranging from 18 to 70 mm (annual total of ~460 mm). The soil type at and around the site is
cryosolic and the surface type is ~70% tundra and ~30% ocean within a 10 km radius of the site. Most storm tracks that reach
Iqaluit originate over the Western Canadian Arctic or the Prairies; these storms can produce strong Easterly winds which
frequently cause blowing snow that severely reduces visibility during non-summer months. Given the site's proximity to
Frobisher Bay (< 600 m), the site is influenced by sea surface conditions during onshore flow (NW). Co-located instrument
evaluation studies were conducted for several remote sensing and upper air observations (Mariani et al., 2020, 2021), including

preliminary model verification studies during the YOPP SOPs and beyond. Iqaluit has over 8,000 inhabitants and is the primary gateway for air and sea traffic for the central and Eastern Canadian Arctic. During the YOPP SOPs, radiosondes were launched four times daily.

## 2.3 Sodankylä, Finland

The Sodankylä site (Figure 4) is managed by the Arctic Space Centre of the Finnish Meteorological Institute (FMI-ARC). It is located in the Scandinavian taiga, which consists of a mix of spruces, pines and birches. The instruments (Figure 4 and Table 6) at the Sodankylä site are distributed over seven main observational sites, each of them including several installations (48m, 24m, 20m or 16m towers, automatic weather stations (AWS), structures supporting snow and soil measurements) that cover an area of approximately 1.5 km$^2$. The environment of the observational sites varies between dense forest, sparse forest, forest openings, and wetland, each of these environments having its own particular surface characteristics.

Sodankylä experiences monthly temperatures ranging from -11 to 15 ℃ (annual mean of 1 ℃) and average monthly precipitation ranging from 35 to 85 mm (annual total of ~660 mm). The site is a calibration/validation site for numerous satellite products (such as snow water equivalent and snow extent (Luojus et al., 2021), and soil freeze-thaw (Cohen et al., 2021 and Rautiainen et al., 2016). The spatial distribution of the observational sites reflects the need of measuring the spatial variability of observed parameters over different spatial scales and satellite footprints (Hannula et al., 2016). During the YOPP SOPs, radiosondes were launched four times daily.

## 2.4 Utqiaġvik (formerly Barrow), USA

The Utqiaġvik site (Figure 5) consists of observatories located ~3 km southeast from the coastline where the Beaufort and Chukchi Seas meet. The site is situated over tundra interspersed with thermokarst lakes having a coverage of up to 40% area (Sellmann et al., 1975). There are two primary observatories located outside of Utqiaġvik (formerly Barrow), Alaska: The Atmospheric Radiation Measurement (ARM) North Slope of Alaska (NSA) observatory operated by the Department of Energy (DOE), and the Barrow Atmospheric Baseline Observatory facility operated by the National Oceanic and Atmospheric Administration (NOAA) Global Monitoring Laboratory (GML). These observatories are equipped with a suite of meteorological instruments (Figure 5 and Table 7) located 8 km east of the town of Utqiaġvik. This is likely beyond the influence of a local heat island in town (Hinkel et al., 2007) and disturbance to snow cover by human activity (Stone et al., 2002). The site includes several towers and space for guest instruments.

Utqiaġvik experiences monthly temperatures ranging from -26 to 9 °C (annual mean of -10 °C) and average monthly precipitation ranging from 35 to 85 mm (annual total of ~770 mm). The climate in Utqiaġvik, and much of the Alaskan North Slope, is regulated by seasonal sea ice cover and the dominance of easterlies that circulate around the Beaufort High. This atmospheric pattern is punctuated by episodes of southerly advection of air masses from the north Pacific, which frequently arrive from the direction of the Bering Strait and are influential the timing of seasonal transitions of terrestrial snow cover and sea ice coverage in both autumn and spring (Cox et al., 2017). The GML Barrow Atmospheric Baseline Observatory recently built a newly furnished on-site laboratory that was completed in 2020. The site's previous facility was constructed in 1972 (https://gml.noaa.gov/obop/brw/history/index.html), and was deconstructed in 2021. The ARM NSA observatory was established in 1997 (Verlinde et al., 2016). Together, the GML and ARM observatories provide an extensive set of long-term measurements at this coastal location. Measurements include properties of aerosols, clouds, precipitation, trace gases, the atmospheric state and the surface energy balance. Unlike the other YOPP sites, radiosondes were launched three times daily during the SOPs.

**2.5 Tiksi, Russia**

The Tiksi observatory (Figure 6) is 7 km away from the town of Tiksi, Russia, in the Sakha Republic of northern Siberia and is staffed by personnel that commute from the town. Tiksi hosts a 20-m flux tower, a clean air facility, a weather station, a Climate Reference Network (CRN) platform, and a Baseline Surface Radiation Network (BSRN) platform, among other instruments (Figure 6 and Table 8) (Ohmura et al., 1998; Driemel et al., 2018). It is a coastal site, with facilities built in a high latitude tundra regime, comprising several different types of tundra land classifications including shrub (most predominant), lichen, wet/dry fen, grassy, bog, water, bare and meadow (Mikola et al., 2018). Meteorologically, Tiksi is located in a boundary region between Atlantic and Pacific air masses. The resulting variability in atmospheric conditions with air masses originating from various source regions in Russia, Northern America, Europe and Central Asia require careful attention and interpretation of in-situ measurements. Tiksi is also influenced by its location at the mouth of the Lena River, the second largest river draining into the Arctic Ocean and the only major Russian river underlain by permafrost which has impacts on the processes and evolution of surface fluxes. Tiksi is also situated on the coast of the Laptev Sea, which is historically a region of large sea-ice production.

Tiksi experiences monthly temperatures ranging from -29 to 11 °C (annual mean of -10 °C) and average monthly precipitation ranging from 15 to 65 mm (annual total of ~510 mm). The original Tiksi science station was established in 1932 and at its height had 60-80 staff and families that lived onsite with a school and grocery store comprising an independent community. In collaboration with the Russian Federal Service for Hydrometeorological and Environmental Monitoring (Roshydromet), a partnership was established with NOAA and the FMI in 2005 to collect climate grade meteorological, surface energy budget,

greenhouse gases and aerosol data (Uttal et al., 2013). Radiosonde data were incorporated into the Integrated Global
Radiosonde Archive (IGRA) and are available through NOAA's National Centers for Environmental Information (NCEI)
portal (Durre et al., 2018). Unlike the other YOPP sites, radiosondes had twice daily launches during the SOPs.

## 2.6 Ny-Ålesund, Norway

At Ny-Ålesund Research Station (Figure 7) in Svalbard, Norway, multi-disciplinary observations are operated by several
institutions of different nationalities. The Norwegian Meteorological Institute (aka MET Norway; www.met.no) is operating
the standard meteorological surface and synoptic observations (Figure 7 and Table 9) reported to the WMO (Maturilli et al.,
2013). The settlement at 78.9°N, 11.9°E, is situated on the south coast of the Kongsfjord, which opens at the west coast of
Svalbard towards the Fram Strait. The fjord stretches in southeast-northwest direction from the large glacier plateau to the
open ocean, and is surrounded by glaciated mountains with altitudes up to 1 km. This geographical setting impacts the local
wind field in the lowermost kilometer, resulting in a mainly southeastern wind direction at Ny-Ålesund, which is temporarily
replaced by a north-westerly wind direction when large-scale synoptic wind is also coming from the according direction. Only
in calm conditions with wind speed < 2 ms$^{-1}$ do katabatic winds from the glaciers south of Ny-Ålesund prevail.
Ny-Ålesund experiences monthly temperatures ranging from -8 to 9 °C (annual mean of -6 °C) and average monthly
precipitation ranging from 17 to 46 mm (annual total of ~590 mm). Ny-Ålesund may be located in the high Arctic, but due to
its location in a coastal environment affected by the West Spitsbergen Current, the local climate is quite maritime and relatively
warm. During the summer months, air temperatures above freezing and the otherwise snow-covered landscape exhibits tundra
ground and the active layer soil surface of permafrost. An overview of the climate conditions and changes in Svalbard is given
by the Norwegian Centre for Climate Services (NCCS, 2018), while the specific atmospheric and radiation conditions in Ny-
Ålesund are described by Maturilli et al. (2019). For the YOPP SOPs, the radiosonde launch frequency was increased from
daily to 6-hourly. Radiosonde launches, four times daily, are contributed by the Alfred Wegener Institute (AWI), and carried
out by the German-French AWIPEV research base that AWI jointly operates with the French Polar Institute Paul-Émile Victor
(IPEV). The radiosondes and weekly ozone sondes are launched from a balloon platform about 200m west of the MET Norway
weather mast. Atmospheric trace gases and cloud condensation nuclei are observed at the Zeppelin Observatory at about 474
m a.s.l. on Zeppelin Mountain south of Ny-Ålesund, operated by the Norwegian Polar Institute (NPI), the Norwegian Institute
for Air Research (NILU), Stockholm University, the Japanese National Institute of Polar Research (NIPR), and others. The
full complement of atmospheric measurements at Ny-Ålesund highlights the interwoven research community that contributes
to making Ny-Ålesund an observational site. More information on the Ny-Ålesund Research Station is available at
https://nyalesundresearch.no.

**2.7 Eureka, Canada**

The Canadian Network for the Detection of Atmospheric Change (CANDAC) runs the Polar Environment Atmospheric Research Laboratory (PEARL) (Figure 8) near the Environment and Climate Change Canada (ECCC) Eureka Weather Station (EWS) in Nunavut, Canada. PEARL has three facilities: the Ridge Laboratory (RL), the Zero Altitude PEARL Auxiliary Laboratory (0PAL), and the Surface and Atmospheric Flux Irradiance Extension (SAFIRE). PEARL collects a wide variety of measurements across all three facilities (Figure 8 and Table 10). The observations used from the Eureka station for the MODF$_{ysm}$ (Akish and Morris, 2023a) were primarily measured at the 0PAL and SAFIRE on-site facilities. The 0PAL lab is situated at approximately 10 m a.s.l. elevation to capture measurements in the lowermost atmosphere. The SAFIRE facility is located about 5 km from the EWS, and it is located away from any structures. At SAFIRE, there is a former BSRN station, a flux tower, and additional remote sensing instrumentation. Additional details about the site including its instrumentation, dataset validation and uncertainties, etc., can be found in Fogal et al. (2013) and at https://www.pearl-candac.ca/website/index.php/facilities. Only a subset of the available measurements collected have been included in the MODF$_{ysm}$ (Akish and Morris, 2023a) due to time constraints and processing resources. Ellesmere Island, where Eureka is situated, is characterized by complex topography that generates mesoscale atmospheric circulations, such as downsloping winds (e.g., Persson and Stone, 2007). The local summertime atmosphere is likely regulated also by nearby ice conditions (Persson and Stone, 2007; Tremblay et al., 2019), which vary between the northern side of the island where multiyear pack ice persists (e.g., Alert) and other coastal areas, which are generally adjacent to seasonal ice cover (e.g., Eureka). However, the general dryness of the atmosphere over Ellesmere is likely a regional anomaly related to location relative to dominant pressure patterns over the Beaufort Sea and near the pole rather than being local (Cox et al., 2012).

Eureka has a minimum monthly average temperature of -37.4 °C in February, a maximum of 6.1 °C in July, and a yearly average of -19 °C. Average monthly precipitation ranges from 9 to 53 mm (annual total of ~285 mm). Details of Eureka's climatology are described in Lesins et al. (2010) and water vapor climatology in Weaver et al. (2017). For the period from 1954–2007, the monthly average dry bulb air temperature minimum occurs in February at approximately -37 °C, with the maximum in July at approximately 5 °C. ECCC also publishes climate normals for Eureka at https://climate.weather.gc.ca/climate_normals/results_1981_2010_e.html?stnID=1750&autofwd=1. Eureka is generally colder and drier than Utqiaġvik (Cox et al., 2012). The soils are mostly marine deposits, and the topography, apart from the stony ridges, is driven mostly by ground ice (Pollard and Bell, 1998; Pollard et al., 2015). Cloud cover over Eureka is anomalous relative to other Arctic observatories, with generally higher cloud bases, a smaller proportion of supercooled liquid, and a seasonal cycle offset from the typical pattern observed elsewhere (Shupe, 2011; Shupe et al., 2011). Eureka increased their twice daily radiosonde launches to four daily launches during the SOPs.

## 3 Instrumentation and Derived Variable Calculation

Standard surface meteorological observations (winds, temperature, pressure, humidity, precipitation) were conducted by instruments of similar design, operation, and accuracy at the different sites. The MODF files have an attribute "Instrument," which specifies the exact instrument model used for each variable at each site. For each site, the full list of measured variables, instrument model and manufacturer, temporal resolution, measurement uncertainty, and operating configuration is provided in Tables 4-10 (note that the information in these tables is also documented in the attributes of the MODFs themselves). The uncertainties provided in these tables originate from the manufacturer and often depend on the meteorological conditions (e.g., relative humidity observations are less accurate during very low temperatures); as such, the largest reported uncertainty was provided for each geophysical variable to provide a conservative error estimate.

For all sites, Vaisala RS92 or RS41 radiosondes were used to collect vertical profile observations from the surface up to the stratosphere. For Iqaluit and Whitehorse, however, the radiosonde manufacturer changed during SOP2 from Vaisala (RS92) to GRAW on September 12, 2018 (no impact on the data quality is anticipated). The radiation flux, cloud base height, and snowfall flux observations are the only derived variables that were explicitly calculated in the MODF (as opposed to the direct observations described in the paragraphs above). The heat flux observations were processed using the eddy correlation and bulk method (see for instance Baldocchi, 2014). Additional processing and quality control methods for these observations are discussed in Section 4. Cloud-base height observations were output by the Vaisala CL51 ceilometer at most sites (where available) using a proprietary algorithm to determine the lowest cloud base height; the uncertainty of this algorithm isn't reported but the ceilometer has a reported distance accuracy of $\pm$ 5 m from the manufacturer. ARM technical reports, instrument validation / evaluation, and quality control measures are linked and available within the Utqiaġvik/Barrow MODF$_{ysm}$ (Akish and Morris, 2023c).

For all observations, instantaneous time is reported at the instruments' raw sampling cadence in UTC. The typical temporal cadence for most observations are around 1 minute or less. No temporal interpolation or averaging was performed on the data. The only exception to this is for turbulent fluxes (the only calculated variable), where some averaging (1 to 30 minutes, depending on the variable) is implicit in the calculation of fluxes. Heights are reported as above ground level (AGL), with the exception of the soil thermistor string, which reports depths below the surface in units of cm. For more information on the instrumentation used or further details on the instrument accuracy, precision, and co-located validation studies for certain instruments, refer to the site-specific references listed in Section 2 and/or the WMO Guide to Instruments and Methods of Observation (WMO, 2021).

## 4 Dataset Preparation, quality control, and post-processing

Guidelines for creating MODFs were published as a table in both human-readable (PDF file) and machine-readable (JSON files) formats by Hartten and Khalsa (2022). This "H-K Table" adopts the standards and conventions commonly used in the earth sciences, including NetCDF encoding with Climate and Forecast (CF) Conventions and following CMIP6 naming, as agreed upon by the YOPP community (Uttal et al., 2024). This H-K standard facilitates the creation of MODFs using current requirements and the creator's software of choice, with the MODF toolkits providing tools to assist the user in creating MODFs (Section 6). For the present work, we used H-K Table version 1.3 to guide the criteria for the generation and standardization of naming conventions, units, and global/variable attribute metadata. Observational datasets were collated and formatted for each of the seven sites into a set of NetCDF files in accordance with the table's criteria. The native variable name is saved as an attribute in the MODFs and as previously discussed, no resampling was performed to harmonize different time stepping (the instrument's instantaneous raw sampling frequency is reported, usually about minutely). Acceptance of data into the $MODF_{ysm}$ was generally determined by the variable list described in the table. The processing script is openly available and described in Section 6.

Radiosonde (timeSeriesProfileSonde variables) data in the MODF were binned into 5 m intervals (10 m for Iqaluit and Whitehorse) of geopotential height and all measurements within each bin were averaged. In the case of 5-meter intervals, this most often results in 0, 1, or 2 measurements in each bin: 8%, 82%, 9%, respectively, in SOP1 and 6%, 80%, 13% in SOP2. In both SOP1 and SOP2 at least 99.9% of the measurements have two or fewer measurements, but a given bin can have up to 14 measurements. The number of measurements per bin has been included in the dataset to filter for these situations, as have the actual time and height of each measurement (though also averaged within each bin). For surface precipitation observations, no corrections for solid precipitation under-catchment were performed (the dataset is raw in the MODF); where appropriate, users are recommended to process under-catchment corrections via Kochendorfer et al. (2020).

The present phase of the MODF concept is to use standardized data organization, metadata, and interoperability. While data quality assurance and measurement operation procedures remain in the purview of the contributing stations, considerable effort was undertaken to ensure MODF production followed a transparent, consistent, and standardized data processing chain. This includes efforts to standardize post-processing and filtering techniques (e.g., quality control methods) as much as possible for the same geophysical variable across the different sites. This consistent processing chain is another unique feature of the MODF dataset as it enforces a level of consistency across vastly different observation sites that normally follow their agencies' own data production procedures and methods. A summary of the processing and quality control applied for each site's observations is provided in Tables 4-10. As discussed in more detail in the below subsections, there are some cases where site-specific data processing could not be avoided; data should be used cautiously and with due consideration to each site's processing techniques and quality control (QC) methods for the $MODF_{ysm}$.

**4. 1 Whitehorse and Iqaluit, Canada**

All geophysical variables observed at the Iqaluit and Whitehorse sites were processed in the same manner and included in the MODF$_{ysm}$ (Huang et al., 2023a; 2023b). For most geophysical variables, limited QC was performed on the raw dataset with the intention to remove obvious outliers only. Details regarding the QC performed are provided in Tables 4-5. A very small number (<5%) of observations were flagged by the QC algorithm. The radiation flux observations should be treated with caution since they typically require additional QC processing prior to analysis; no additional QC was performed on these observations to account for potential frost or snow deposition on the sensors, for instance. No additional QC was performed on the cloud base height data, which was processed by the Vaisala software. Vaisala also processed the raw data feed from the radiosonde observations, which was obtained at 2 s resolution; no additional QC was performed. When no data was available (due to the instrument being down, loss of power at the site, or because it was flagged by the QC algorithm), a missing value (-9999.0) was reported in the MODF$_{ysm}$ (Huang et al., 2023a; 2023b) and is notated via the "missing_value" attribute associated with each variable. Mariani et al. (2020, 2021) provides instrument validation studies and more detailed information on the quality control processing routines for the remote sensing and upper air observations.

**4. 2 Sodankylä, Finland**

The Sodankylä observations included in the MODF$_{ysm}$ (O'Connor, 2023) are automatically uploaded every day to the FMI open access web site https://litdb.fmi.fi/ where the data are organized on the basis of platforms and stations. Before being uploaded to the web page, the data undergo an automatic quality check to remove outliers, as described in Table 6. In the current MODF$_{ysm}$ version (O'Connor, 2023), no further quality check was applied to the data, implying that errors from several sources are occasionally included. These sources of error may include snow/frost deposition on radiation and temperature sensors or absorption of solar radiation by unsheltered temperature sensors. In a future version of the MODF$_{ysm}$, a deeper quality check will be applied to some of the variables included in the current MODF$_{ysm}$ (O'Connor, 2023). This quality check is based on the comparison among the same variables measured at different sites, on visual inspection and, in the case of global radiation, on the comparison with radiative transfer model calculations. This processing will enable the identification of the shortwave data affected by the shadows casted by the vegetation, of errors caused by frost formation on the domes of pyranometers, and of the error in unshaded thermometers caused by the absorption of solar radiation. As in the case of the Eureka observatory, the radiosonde data in the MODF was ingested and processed by IGRA and is available through NOAA's NCEI portal (Durre et al., 2018).

**4. 3 Utqiaġvik (formerly barrow), USA, Tiksi, Russia and Eureka, Canada**

The Utqiaġvik/Barrow data within the MODF$_{ysm}$ (Akish and Morris, 2023c) originated from both DOE/ARM and NOAA GML datasets, with GML proving datasets for ozone, snow thickness, skin temperature, soil temperature profile. Value added products were generated and disseminated to the users using the ARM Data Discovery interface. Both the ARM and GML datasets were ingested into a single MODF$_{ysm}$ with variable attribution detailing how each variable and data set was quality controlled, processed and accessed, as described in Tables 7-8, 10. The surface ozone data was collected in 1-minute intervals and was manually quality controlled and submitted to NCEI. The measurements collected by the ARM facility were processed, QC analyzed, and archived at the ARM Data Center archive. The long-term Eureka and Tiksi datasets (flux tower and radiation) are hosted by the NOAA Physical Sciences Laboratory (PSL), in collaboration with ECCC (Eureka site only), and Roshydromet (Tiksi site only).

For the three sites, the radiation measurements were QC'd and processed following Long and Shi (2008) and improved correction of the infrared loss in diffuse shortwave measurements was included (Younkin and Long, 2003). Turbulent heat fluxes were processed and QC'd via Eddy correlation corrections including stability correction, Webb-Pearman correction, frequency correction, sensor separation correction, filtering correction, line-averaging correction, and volume-averaging correction (Cook et al. 2008, Fuehrer and Friehe 2002). Bulk corrections were also employed and utilized ARM data from the radiation, ground, met, and tower. Radiosonde data were ingested and processed by NOAA's NCEI and was processed through IGRA, following their standards (Durre et al., 2018) and is available through NOAA's NCEI portal. The IGRA 2 QA system processed the sonde data, which is based largely on the QA procedures in the IGRA 1 system (Durre et al. 2006; Durre et al. 2008). Like the IGRA 1 system, it consists of a deliberate sequence of specialized algorithms, each of which makes a binary decision on the quality of a value, level, or sounding; either the data item passes the check and remains available, or it is identified as erroneous and thus set to missing. For all observations, a second level of manual QC was performed whereby data was reviewed by instrument mentors and visually assessed by the site scientist/data quality office. This included removing non-physical values and outliers, after confirming that they were either biased, incorrect, or collected during site maintenance periods. If data was not available for any of the collected measurements across any of the variables, due to the instrument being down, loss of power at the site, or because it was flagged by the QC algorithm, a missing value (-9999) was reported in the MODF$_{ysm}$ (Akish and Morris, 2023b).

**4. 5 Ny-Ålesund, Norway**

The meteorological measurements used for the MODF$_{ysm}$ (Holt, 2023) are taken from the AWIPEV weather mast (Driemel et al., 2018; Maturilli, 2020b). Except for precipitation, all other data used in the MODF$_{ysm}$ for Ny-Ålesund originated from the following data sets: Maturilli (2020a, 2020b, 2020c, 2022). The precipitation data reported in the MODF$_{ysm}$ are the direct

instrument output and no quality checks were applied; as such this data should be treated with caution (Holt, 2023). The Ny-Ålesund observations included in the MODF$_{ysm}$ are a subset of those regularly uploaded in the PANGAEA data repository (www.pangaea.de). Before being uploaded, all data undergo an automatic quality check (described in Table 9). Following this, additional manual/visual inspection was performed as for Utqiaġvik, Tiksi, and Eureka. Surface radiation data were validated and have undergone all quality checks of BSRN before archiving (Maturilli, 2020a).

**5 MODF Data Structure**

The data inside a MODF comprises of all the observations listed in Table 3 for a given observation site. The data itself follows the same standardized format and structure for all observations and sites and is stored into a single NetCDF file using CF conventions. NetCDF file formatting was chosen to best accommodate the high-level of metadata detail required for merging such large quantities of individual measurements together, particularly given the need to be as transparent as possible when reporting instrument-specific details for each observation. NWP model output was stored in MMDFs, matching the MODF format to facilitate model-observation comparisons. Local maps showing the synoptic region around each site are provided in Figure 9, with native spatial grids of the forecast models that participated in YOPPsiteMIP overlaid. This provides visual context of where the site and the nearest NWP grid points exist in and around each site.

All MODF$_{ysm}$ measurements provided in the data files maintained their native time cadence (typically on the order of minutely) with no averaging undertaken, and details of the collection and processing techniques can be found in the variable attributes within the files. Each DOI in Table 2 contains four (e.g., Whitehorse) or six (e.g., Utqiaġvik) files, depending on whether the site had timeSeriesProfile observations on a tower/mast. The filename convention for each MODF is as follows: site name + "obs" + MODF_featureType + start_date + end_date.nc.

Guidelines for creating inventories of variable and attribute information (metadata) necessary for the MODF file attributes were published in spreadsheet format by Morris and Akish (2022). This "A-M Template" uses variable content criteria from the H-K Table to generate a metadata matrix of attribute and variable information for each of the measurements contained within the MODFs. The template has individual tabs for each of the corresponding CF metadata featureTypes (i.e., timeSeries and timeSeriesProfile) of the MODF NetCDF files, as well as one tab for the Global Attributes of the MODFs. The CF Conventions can be found here: https://cfconventions.org/cf-conventions/cf-conventions.html. The attributes within the template are mandatory when applicable, and serve as a guideline for MODF creators. The A-M Template is machine-readable and can be ingested into MODF software to create the final output.

The file content is well-illustrated in Table 3; other details of the MODF$_{ysm}$ format and structure are outlined in Uttal et al.
(2024). MODFs can contain featureTypes such as timeSeries and timeSeriesProfile, which refer to time series having one and
two data dimensions, respectively. In cases where data subcategories exist, featureType modifications can be depicted in the
file name, for example timeSeriesProfileSonde exist for the MODF$_{ysm}$. Currently, more than one featureType can be used
within an individual MODF file, but all subscribe to the same formatting structure and nomenclature. To generate an MODF,
creators would first visit the H-K Table to determine the variables that will be included in their MODF, and then they should
utilize the A-M Template to fill in the needed attribute and variables information requested by existing MODF software. Once
the A-M Template has been completed, then users can ingest the template into their MODF software to create the final MODF
outputs. For the MODF$_{ysm}$, individual toolkits were developed by MODF makers for each YOPP site.  Python code was
developed for Whitehorse, Iqaluit and Ny-Alesund, and MATLAB code for Utqiagvik, Tiksi, Eureka and Soldankyla (see
Section 6). After the generation of the MODF$_{ysm}$ outputs, the files were run through an MODF checker that identifies the
various inconsistencies or issues with the files before their upload to the MET Norway data portal. The MODF$_{ysm}$ checker
developed for the YOPPsiteMIP files is part of a larger toolkit being designed to continue the creation of MODFs.

As an example of the uniformity of the observations (in terms of data format, post-processing, temporal cadence, etc.)
contained within each site's MODF$_{ysm}$ and their data coverage during the two YOPP SOPs, Figures 10 and 11 provide the
surface downwelling longwave radiation and near-surface temperature observations from each site's MODF$_{ysm}$ during SOP1,
respectively and Figures 12 and 13 show the same except for SOP2. The MET Norway data portal and MODF maker toolkit
(Sect. 6) also provides plotting tools that work with any MODF or MMDF and can produce similar figures automatically.
Periods of interest can be quickly identified by users and analyzed for further investigation and/or comparison with their
corresponding MMDFs. MODFs significantly simplify the process of analyzing observations from multiple sites and multiple
instruments, as analyses and Figures can be produced for each site using a single code that works for any observed geophysical
variable and (if desired) their corresponding NWP model output in the MMDF. In contrast, without MODFs a user would have
to contact each meteorological agency individually, find each sites' data repository, obtain data access privileges, find the files
they need from multiple instruments, reprocess and reformat multiple uniquely-formatted datasets and file types, then develop
several different codes (e.g., readers) specific to each instruments' dataset to ingest the multi-variate datasets and plot them.

The MODF$_{ysm}$ at Sodankylä are unique in that their measurements are collected across a series of sub-sites in the area;
therefore, it is important to describe here the possible methods for extracting the data for specific locations, or for co-located
measurements. The Sodankylä station comprises at least 25 distinct locations, the precise number of which is given by the
dimension 'site_id' inside the MODF data file. Each distinct location is given a unique index key in the variable 'subsite_name',
with these indices also identifying the 'lat', 'lon' and 'soil_type' for each location. The corresponding FMI names for each
location are identified in the attribute 'flag_meanings' for the variable 'subsite_name' via their indices; for example, the index
value of 16 pointing to IOA003_spot_8, which is one of the automatic weather stations located in the Intensive Observations
Area (IOA). There may be multiple locations providing the same measurement. However, not all locations provide the same
set of measurements, and to keep the MODF compact, each measurement variable has the location dimension truncated to
include only locations which measure that variable; i.e., the location dimension for the measurement variables is 'nsubsites_X',
where X is the number of locations making the particular measurement. This set of locations is accessed through the indices
given in the attribute 'subsite_name' for the measurement variable, which corresponds to the key given in the 'subsite_name'
variable; i.e., a subsite_name attribute of "1, 3, 10" means that these measurements were made at the locations identified by
their indices, from which their locations (latitudes and longitudes) and soil_type can also be determined.
This method permits diverse options of collecting measurements for particular uses. All measurements, for example, at one
location can be obtained by identifying the appropriate 'subsite_name' index inside the MODF data file, iterating through the
'subsite_name' attribute of each variable to see if it contains the selected index, and, if so, selecting the column or slice of data
for the data that matches the location of the index (i.e. if subsite_name = 10 and the subsite_name attribute for a timeSeries
variable is "1, 3, 10", the measurement timeSeries for the requested location is in the third column, the next variable may have
a subsite_name attribute of "1, 3, 5, 6, 10" and the measurement timeSeries for the requested location is in the fifth column).
The user could also select a specific area of interest and identify all measurements made within this region as follows: select
the indices for the locations within a specified latitude and longitude range, then iterate through the 'subsite_name' attribute of
each variable to see if it contains the selected indices and return the columns or slices that match them.
Note that each site conducts additional observations not listed in table 3 that will be included in upcoming updates to the
MODF$_{ysm}$ with the intent to eventually incorporate all observations into the MODF$_{ysm}$ for each site. This process of developing
and appending to MODFs can be extended to other sites and/or research programs that wish to create MODFs of their
observations. Given the standardized nature of the MODFs, reading and analyzing datasets from any of the YOPP sites is
simplified. Quick-look plotting tools have been developed via the MET Norway YOPP data portal and the MODF maker
toolkit (Sect. 6), which enable near-instantaneous plotting of the observations contained within the MODF$_{ysm}$.

## 6 Data and Code Availability

The MODF$_{ysm}$ for each site are available via the MET Norway YOPP Data Portal (https://yopp.met.no/) where they are
indexed through FAIR compliant discovery metadata and can be directly accessed at:
https://thredds.met.no/thredds/catalog/alertness/YOPP_supersite/obs/catalog.html (Whitehorse:
https://doi.org/10.21343/a33e-j150, Iqaluit: https://doi.org/10.21343/yrnf-ck57, Sodankylä: https://doi.org/10.21343/m16p-
pq17, Utqiaġvik: https://doi.org/10.21343/a2dx-nq55, Tiksi: https://doi.org/10.21343/5bwn-w881, Ny-Ålesund:
https://doi.org/10.21343/y89m-6393, Eureka: https://doi.org/10.21343/r85j-tc61).

Proper data citation ensures appropriate credits to authors of both input data sources and merged MODF$_{ysm}$ datasets. Data from
each station has been assigned a DOI. The variable attributes of the merged data products contain information about the source
datastreams and their DOIs, to more clearly establish data provenance in a traceable manner. When using data from the
MODF$_{yms}$, it is expected that the user references the MODF$_{ysm}$ DOI, and any subsidiary variable DOIs when available.
Assigning citations for merged data streams such as the MODF$_{ysm}$ is a challenging and still evolving concept. For example,
the US DOE ARM Program uses a combination of DOI and citation structure for continuous data streams, as outlined in
Prakash et al. (2016). They recommend when registering DOIs for derived and higher-order data, source DOIs in the metadata
of the newly created DOI should be added and linked when possible.

The source code used to produce the MODF$_{ysm}$ for each site (and MODFs in general) are available via gitlab:
https://gitlab.com/mdf-makers/mdf-toolkit. This MODF toolkit is openly available for anyone interested in developing their
own MODF file or generating quick-look plots of the data contents inside the MODFs. The toolkit is regularly updated as the
MODF community grows and new geophysical variables and/or functions are added. Additional site-specific python and
MATLAB codes that were used to prepare the observation data files for MODF ingestion are available upon request (e.g.,
contact the site principle investigator).

**7 Concluding Remarks**
The enhanced ground-based observations conducted at both Poles during the YOPP fill significant and identified gaps in our
current meteorological observation capabilities for the Polar Regions. YOPPsiteMIP MODFs (MODF$_{ysm}$) have been published
for seven of the YOPP Arctic sites, whereby all geophysical variables are stored in an identical, standardized format in a single
NetCDF file following CF conventions. This fulfills a key objective of the program to perform single- or multi-variate model-
observation comparisons. These MODFs archive data in a manner as similar as possible to corresponding MMDF (see Uttal
et al., 2024) that contain high-resolution forecast variables from a single NWP model at and around a site (Figure 9). Thus,
combined, MODFs and MMDFs greatly simplify integration of these complex datasets, enabling further scientific study as
demonstrated in the recent publications using the latest MODF$_{ysm}$ and MMDF$_{ysm}$ (Day et al., 2024).

Standardized geophysical variable nomenclature, cadences, metadata, basic QC, and file structure were employed to create
these files. MODFs provide the first standardized files for archiving all the different ground-based observation site

observations, containing a multitude of geophysical variables observed by (at times) different instruments. This amalgamation of different sites' observations into a standardized, user-friendly MODF format enables easier analysis of the MODF dataset, inter-site comparisons, and detailed NWP model validation, evaluation, intercomparisons, and process-based diagnostic studies that are currently underway (e.g., Figures 10 to 13). The further adoption, creation, and use of MODFs outside of YOPP is encouraged; a suite of tools and documentation is openly available via Gitlab (Sect. 6) for other site managers, researchers, and users to develop and create their own site-specific MODFs outside of YOPP or to analyze an observation sites' dataset.

The YOPP MODF$_{ysm}$ discussed here provide novel access to datasets of enhanced meteorological observations collected at several sites across the Arctic. The MODF concept is not limited for use in polar regions and could be exported elsewhere. Seven YOPP-designated sites in the Arctic developed and published MODF$_{ysm}$ covering both SOP periods (February – March 2018 and July – September 2018), including Iqaluit, Whitehorse, and Eureka in Canada, Utqiaġvik in the United States, Tiksi in Russia, Sodankylä in Finland, and Ny-Ålesund in Norway. Additional geophysical variables observed at each of these seven sites will be included in a future update of their MODF$_{ysm}$, with the goal of having almost all of a site's observations available. Observations at most of these sites continue today beyond YOPP and are available for subsequent analyses, in some cases using updated MODFs generated in near-real time. MODF$_{ysm}$ for the other YOPP sites, including ship-based platforms and sites in the Antarctic, will be made available in the future to complete the YOPP dataset. The MODF$_{ysm}$ described here directly ties to process-oriented verification studies aiming to improve NWP predictions at the Poles by contributing and enabling NWP inter-comparisons.

## Author contributions

SM, ZM, and TU wrote the first draft of the manuscript. SM and ZM conducted scientific analyses and created tables and figures with JD and JT. All authors managed data archiving, creation of the $MODF_{ysm}$, and publication to the MET Norway YOPP Data Portal. All authors contributed to the writing and the editing of the manuscript.

## Competing interests

The authors declare that they have no conflict of interest.

## Disclaimer

Use of specific instrument manufacturers/models and suppliers mentioned in the manuscript and/or used at the sites is not a commercial endorsement of their products.

## Acknowledgements

This is a contribution to the Year of Polar Prediction (YOPP), a flagship activity of the Polar Prediction Project (PPP), initiated by the World Weather Research Programme (WWRP) of the World Meteorological Organisation (WMO). We acknowledge the WMO WWRP for its role in coordinating this international research activity. This study was supported by NOAA's Global Ocean Monitoring and Observing Program through the Arctic Research Program (FundRef: https://doi.org/10.13039/100018302). Special thanks to the station technicians and operators at the sites for deploying instruments, maintenance, and technical services. In particular, thank you to the radiosonde operators for providing extra daily sonde launches during the two SOP periods. Thank you to Jenn Glaser for her contract work in creating the station graphic in Figure 1, and to Kyrie Newby and Kalvin Jesse for creating the Google Earth images in Figure 9. JD was supported by the European Union funded INTERACTIII project (Grant Agreement: 871120). AK and LMH were supported in part by NOAA cooperative agreements NA17OAR4320101 and NA22OAR4320151. Potions of the $MODF_{ysm}$ data were obtained from the Atmospheric Radiation Measurement (ARM) user facility, a U.S. Department of Energy (DOE) office of science user facility managed by the biological and environmental research program. Thank you to MET Norway for hosting the YOPP data portal. All data products are produced by their respective institutions and are available via the YOPP data portal (https://yopp.met.no) and directly at: https://thredds.met.no/thredds/catalog/alertness/YOPP_supersite/obs/catalog.html.

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

**Table 1.** List of facility coordinates for locations where $MODF_{ysm}$ measurements were collected at each site. The measured variables that are observed at each site are listed (refer to Table 3). In some cases, the same variable is measured at multiple locations for a single site; these observations and their corresponding coordinates are embedded within the MODF. "All" refers to the entire list of the measured variables in Table 3, whereas "All radiation" refers to all radiation-related measured variables.

| | **Facility Name** | **Coordinates** | **Measured Variables** (from Table 3) |
|---|---|---|---|
| **Whitehorse** | Whitehorse | N60.71, W135.07 | All |
| **Iqaluit** | Iqaluit | N63.74, W68.51 | All |
| **Sodankylä** | Operative Sounding Station Area; Automatric Weather Station (LUOxxxx) | N67.366618 – N67.367220, E26.628253 - E26.63144 | Pressure, Visibility |
| | CO2 Flux Mast Area (VUOxxxx) | N67.361883, E26.643003 - E26.64323 | Total precipitation of water, all wind, vertical velocity, temperature, dew-point temperature, relative humidity, snow thickness, all radiation, cloud base height |
| | Intensive Observation Area (IOAxxxx) | N67.361654 - N67.361950, E26.633190 - E26.634191 | Temperature, relative humidity, snow thickness, snowfall flux, snow water equivalent, all short-wave radiation, soil temperature profile, soil moisture, snow temperature |
| | Lichen Fence (JAKxxxx) | N67.36710 - N67.36716, E26.634740 - E26.63513 | All radiation |
| | Micrometeorological Mast Area (METxxxx) | N67.361711 - N67.36216, E26.63726 - E26.65117 | All wind, temperature, vertical velocity, relative humidity, snow thickness, all radiation, all heat fluxes, friction velocity, soil temperature profile, soil moisture, snow temperature |
| | Peatland Area (SUOxxxx) | N67.361903 - N67.36707, E26.633802 - E26.654067 | Temperature, dew-point temperature, relative humidity, snow thickness, all short-wave radiation, soil temperature profile, soil moisture, snow temperature |
| **Utqiaġvik** | ARM Facility | N71.19228, W156.3654 | All except ozone concentration, snow thickness, and soil temperature profile |
| | GML Barrow Atmospheric Baseline Observatory | N71.3230, W156.6114 | Ozone concentration, snow thickness, and soil temperature profile |
| **Tiksi** | Baseline Surface Radiation Network (BSRN) | N71.5862, E128.9188 | All radiation observations |
| | Fluxtower | N71.595, E128.882 | All except radiation observations |
| **Ny-Ålesund** | Baseline Surface Radiation Network (BSRN) | N78.92278, E11.92725 | All radiation observations, pressure, cloud base height |
| | AWIPEV Met.Tower | N78.92226, E11.92667 | All wind, temperature, relative humidity, specific humidity |
| | Balloon Launch Facility | N78.92301, E11.92271 | All timeSeriesProfileSonde observations |

| | | | |
|---|---|---|---|
| **Eureka** | Baseline Surface Radiation Network (BSRN) | N79.989, W85.9404 | All radiation observations |
| | Fluxtower | N80.083, W86.417 | Pressure, all wind, temperature, relative humidity, snow thickness, ground heat flux, soil temperature profile |
| | Sonde Launch | N79.9833, W85.9333 | All timeSeriesProfileSonde observations |


**Table 2.** List of final DOIs for each site's MODF$_{ysm}$.

| | DOI | Title | Citation |
|---|---|---|---|
| **Whitehorse** | https://doi.org/10.21343/a33e-j150 | MODF for Erik Nielsen Airport, Whitehorse, Canada during YOPP SOP1 and SOP2 | Huang et al., 2023a |
| **Iqaluit** | https://doi.org/10.21343/yrnf-ck57 | MODF for Iqaluit Airport, Iqaluit, Nunavut, Canada during YOPP SOP1 and SOP2 | Huang et al., 2023b |
| **Sodankylä** | https://doi.org/10.21343/m16p-pq17 | Merged observation data file for Sodankylä | O'Connor, 2023 |
| **Utqiaġvik** | https://doi.org/10.21343/a2dx-nq55 | MODF for Utqiaġvik, Alaska, during YOPP SOP1 and SOP2 | Akish and Morris, 2023c |
| **Tiksi** | https://doi.org/10.21343/5bwn-w881 | MODF for Tiksi, Russia, during YOPP SOP1 and SOP2 | Akish and Morris, 2023b |
| **Ny-Ålesund** | https://doi.org/10.21343/y89m-6393 | Merged Observatory Data File (MODF) for Ny Ålesund | Holt, 2023 |
| **Eureka** | https://doi.org/10.21343/r85j-tc61 | MODF for Eureka, Canada, during YOPP SOP1 and SOP2 | Akish and Morris, 2023a |



**Table 3.** List of the geophysical variables currently included in each site's MODF. Note that this table only includes variables currently in the existing MODF$_{ysm}$, and does not indicate the complete list of variables that are observed at each site. An asterisk (*) denotes a variable not included in the H-K table (Hartten and Khalsa, 2022) and a double asterisk (**) denotes a calculated variable. The level and type(s) of additional processing for the heat fluxes are also provided, where EC = eddy covariance and bulk = bulk method.

| MODF featureType | Measured Variables | Whitehorse *lat: 60.71 N lon: 135.07 W* | Iqaluit *lat: 63.74 N lon: 68.51 W* | Sodankylä *lat: 67.367 N lon: 26.629 E* | Utqiaġvik *lat: 71.325 N lon: 156.625 W* | Tiksi *lat: 71.596 N lon: 128.889 E* | Ny-Ålesund *lat: 78.923 N lon: 11.926 E* | Eureka *lat: 80.083 N lon: 86.417 W* |
|---|---|---|---|---|---|---|---|---|
| **timeSeries Variables** | Pressure (Pa) | surface | surface | surface, mean sea-level | surface | surface | surface | surface |
| | Total precipitation of water in all phases per unit area (kg m$^{-2}$ s$^{-1}$) | surface | surface | surface | | | surface | |
| | Eastward Wind (m s$^{-1}$) | surface | near-surface | near-surface | near-surface (2m) | near-surface (4m) | near-surface (10m) | near-surface (6m) |
| | Northward Wind (m s$^{-1}$) | surface | near-surface | near-surface | near-surface (2m) | near-surface (4m) | near-surface (10m) | near-surface (6m) |
| | *Wind gust (m s$^{-1}$) | | | near-surface (10m) | | | | |
| | Vertical velocity (m s$^{-1}$) | | | near surface (2 m) | | | | |
| | Temperature (K) | near-surface (2m) | near-surface (2m) | skin, near-surface (2m) | skin, near-surface (2m) | skin, near-surface (2m) | near-surface (2m) | skin, near-surface (2m) |
| | Dew-point Temperature (K) | near-surface (2m) | near-surface (2m) | near-surface (2m) | near-surface (2m) | | | |
| | Relative Humidity (1 or %) | near-surface (2m) | near-surface (2m) | near-surface (2m) | near-surface (2m) | near-surface (2m) | near-surface (2m) | near-surface (2m) |
| | Specific Humidity (1 or kg kg$^{-1}$) | | | | | | near-surface (2m) | |
| | Ozone Concentration in Air (mole fraction) | | | | surface | | | |
| | Snow thickness (m) | | surface | surface | surface | surface | | surface |
| | Snowfall Flux (kg m$^{-1}$ s$^{-2}$) | | | | surface | | | |
| | Snow water equivalent (kg m$^{-2}$) | | | | surface | | | |
| | Upward Short-wave Radiation (W m$^{-2}$) | | surface | surface | surface | surface | surface | surface |
| | Downward Short-wave Radiation (W m$^{-2}$) | | surface | surface | surface | surface | surface | surface |
| | Upward Long-wave Radiation (W m$^{-2}$) | | surface | surface | surface | surface | surface | |
| | Downward Long-wave Radiation (W m$^{-2}$) | | surface | surface | surface | surface | surface | surface |
| | Net Short-wave Radiation at the Surface (W m$^{-2}$) | | | surface | | | | |
| | *Horizontal East-facing Long-wave Radiation (W m$^{-2}$) | | surface | | | | | |
| | *Horizontal West-facing Long-wave Radiation (W m$^{-2}$) | | surface | | | | | |
| | *Horizontal South-facing Long-wave Radiation (W m$^{-2}$) | | surface | | | | | |
| | *Horizontal North-facing Long-wave Radiation (W m$^{-2}$) | | surface | | | | | |
| | **Turbulent Latent Heat Flux (W m$^{-2}$) | | | surface (EC) | surface (EC, bulk) | | | |
| | **Turbulent Sensible Heat Flux (W m$^{-2}$) | | | surface (EC) | surface (EC, bulk) | | | |
| | **Turbulent time-average eastward stress (Pa) | | | surface (EC) | surface | | | |
| | **Turbulent time-average northward stress (Pa) | | | | surface | | | |
| | *Friction Velocity (m s$^{-1}$) | | | surface (EC) | | | | |
| | Cloud Base Height (m) | ground-based remote sensing | ground-based remote sensing | ground-based remote sensing | | | ground-based remote sensing | |
| | Ground Heat Flux (W m$^{-2}$) | | | near-surface | near-surface | near-surface | | near-surface |
| | Visibility (m) | | | near-surface | | | | |

| | | | | | | | |
|---|---|---|---|---|---|---|---|
| **timeSeriesProfile Variables** | Atmospheric pressure (Pa) | near-surface (2m, 10m) | | | | | |
| | Total precipitation of water in all phases per unit area (kg m$^{-2}$ s$^{-1}$) | near-surface (2m, 10m) | | | | | |
| | Eastward Wind (m s$^{-1}$) | near-surface (2m, 10m) | near-surface (18m, 32m, 38m, 48m) | near-surface (2m, 10m, 20m, 40m) | | near-surface (2m, 10m) | near-surface (6m, 11m) |
| | Northward Wind (m s$^{-1}$) | near-surface (2m, 10m) | near-surface (18m, 32m, 38m, 48m) | near-surface (2m, 10m, 20m, 40m) | | near-surface (2m, 10m) | near-surface (6m, 11m) |
| | Temperature (K) | near-surface (2m, 10m) | near-surface (3m, 8m, 18m, 32m, 48m) | near-surface (2m, 10m, 20m, 40m) | near-surface (2m, 6m, 10m) | near-surface (2m, 10m) | near-surface (2m, 6m, 10m) |
| | Dew-point Temperature (K) | | | near-surface (2m, 10m, 20m, 40m) | | | |
| | Relative Humidity (1 or %) | near-surface (2m, 10m) | near-surface (3m, 8m, 18m, 32m, 48m) | near-surface (2m, 10m, 20m, 40m) | near-surface (2m, 6m, 10m) | | near-surface (2m, 6m, 10m) |
| | Soil Temperature Profile (K) | | sub-surface (5cm, 30cm) | sub-surface (5cm, 10cm, 15cm, 20cm, 25cm, 30cm, 45cm, 70cm, 95cm, 120cm) | sub-surface (5cm, 10cm, 15cm, 20cm, 25cm, 30cm, 45cm, 70cm, 95cm, 120cm) | | sub-surface (5cm, 10cm, 15cm, 20cm, 25cm, 30cm, 45cm, 70cm, 95cm, 120cm) |
| | Soil Moisture (kg m$^{-2}$) | | sub-surface (5cm, 30cm) | | | | |
| | Snow Temperature (K) | | near-surface (10cm, 20cm, 30cm, 40cm, 50cm, 60cm, 70cm, 80cm, 90cm, 100cm, 110cm) | | | | |
| **timeSeriesProfileSonde Variables** | Atmospheric pressure (Pa) | radiosonde | radiosonde | | radiosonde | radiosonde | |
| | Eastward Wind (m s$^{-1}$) | radiosonde | radiosonde | | radiosonde | radiosonde | |
| | Northward Wind (m s$^{-1}$) | radiosonde | radiosonde | | radiosonde | radiosonde | |
| | Temperature (K) | radiosonde | radiosonde | | radiosonde | radiosonde | |
| | Dew-point Temperature (K) | radiosonde | radiosonde | | radiosonde | radiosonde | |
| | Specific Humidity (1 or kg kg$^{-1}$) | | | | | Radiosonde | |
| | Relative Humidity (1 or %) | radiosonde | radiosonde | | radiosonde | radiosonde | |

*Denotes a variable NOT included in the H-K Table*
**Denotes a calculated variable (not a direct observation)**






**Table 4.** List of the instruments that contributed to the Whitehorse MODF, including details about the instrument manufacturer,
measured variables, configuration, temporal resolution, measurement uncertainty, and quality control applied.

| MODF featureType | Instrument | Manufacturer | Measured variables | Instrument Configuration | Temporal Resolution | Uncertainty (+/-) | Quality Control |
|---|---|---|---|---|---|---|---|
| timeSeries Variables | WXT520 | Vaisala | Atmospheric pressure (Pa) | Solid-state, all-in-one weather instrument in standard aspirated configuration mounted on a pole. No bird spike kit used. | 1 min | 0.5 hPa | Beyond the standard Vaisala processing, observations were checked against site-based climatology ranges and the rate of change thresholds, which were based on hourly criteria. Observations that fell outside of the 3-sigma normal climatological range were rejected, as were observations that had a rate of change greater than a seasonal-dependant threshold (e.g., >20 hPa/hr change). |
| timeSeries Variables | WXT520 | Vaisala | Total precipitation of water in all phases per unit area (kg $m^{-2}$ $s^{-1}$) | Solid-state, all-in-one weather instrument in standard aspirated configuration mounted on a pole. No bird spike kit used. | 1 min | 5% | Beyond the standard Vaisala processing, observations were checked against site-based climatology ranges and the rate of change thresholds, which were based on hourly criteria. Observations that fell outside of the 3-sigma normal climatological range were rejected, as were observations that had a rate of change greater than a seasonal-dependant threshold (e.g., > 10 mm/hr change). No corrections for solid precipitation under-catchment were performed (the dataset is raw in the MODF); where appropriate, users are recommended to process under-catchment corrections via Kochendorfer et al. (2020) (note: undercatchment is less of an issue for the WXT520 observations compared to Pluvio2). |
| timeSeries Variables | WXT520 | Vaisala | Eastward Wind (m $s^{-1}$) | Solid-state, all-in-one weather instrument in standard aspirated configuration mounted on a pole. No bird spike kit used. | 1 min | 0.3 ms$^{-1}$ | Beyond the standard Vaisala processing, observations were checked against site-based climatology ranges and the rate of change thresholds, which were based on hourly criteria. Observations that fell outside of the 3-sigma normal climatological range were rejected, as were observations that had a rate of change greater than a seasonal-dependant threshold (e.g., > 10 m/s/hr change). |
| timeSeries Variables | WXT520 | Vaisala | Northward Wind (m $s^{-1}$) | Solid-state, all-in-one weather instrument in standard aspirated configuration mounted on a pole. No bird spike kit used. | 1 min | 0.3 ms$^{-1}$ | Beyond the standard Vaisala processing, observations were checked against site-based climatology ranges and the rate of change thresholds, which were based on hourly criteria. Observations that fell outside of the 3-sigma normal climatological range were rejected, as were observations that had a rate of change greater than a seasonal-dependant threshold (e.g., > 10 m/s/hr change). |

| | | | | | | | |
|---|---|---|---|---|---|---|---|
| timeSeries Variables | WXT520 | Vaisala | Temperature (K) | Solid-state, all-in-one weather instrument in standard aspirated configuration mounted on a pole. No bird spike kit used. | 1 min | 0.3 K | The shelter heating effect is uncorrected beyond the Vaisala standard processing. Beyond the standard Vaisala processing, observations were checked against site-based climatology ranges and the rate of change thresholds, which were based on hourly criteria. Observations that fell outside of the 3-sigma normal climatological range were rejected, as were observations that had a rate of change greater than a seasonal-dependant threshold (e.g., > 5 K/hr change). |
| timeSeries Variables | WXT520 | Vaisala | Relative Humidity (1 or %) | Solid-state, all-in-one weather instrument in standard aspirated configuration mounted on a pole. No bird spike kit used. | 1 min | 3% | The humidity is not corrected in a sub-freezing environment, beyond the standard Vaisala processing. Beyond the standard Vaisala processing, observations were checked against site-based climatology ranges and the rate of change thresholds, which were based on hourly criteria. Observations that fell outside of the 3-sigma normal climatological range were rejected, as were observations that had a rate of change greater than a seasonal-dependant threshold (e.g., > 30 %/hr change). |
| timeSeries Variables | WXT520 | Vaisala | Dew-point Temperature (K) | Solid-state, all-in-one weather instrument in standard aspirated configuration mounted on a pole. No bird spike kit used. | 1 min | 0.5 K | The shelter heating effect is uncorrected and humidity is not corrected in a sub-freezing environment, beyond the standard Vaisala processing. Beyond the standard Vaisala processing, observations were checked against site-based climatology ranges and the rate of change thresholds, which were based on hourly criteria. Observations that fell outside of the 3-sigma normal climatological range were rejected, as were observations that had a rate of change greater than a seasonal-dependant threshold (e.g., > 5 K/hr change). |
| timeSeries Profile Variables | CL51 | Vaisala | Cloud Base Height (m) | Proprietary algorithm determines the lowest cloud base height | 1 min | 5 m | No QC was performed, beyond the standard Vaisala proprietary algorithm that retrieves cloud base height. |
| timeSeries ProfileSonde Variables | RS92 / DFM-09 | Vaisala / GRAW | Atmospheric pressure (Pa) | Standard radiosonde launch | 6 hr | 0.5 hPa | Vaisala also processed the raw data feed from the radiosonde observations, which was obtained at 2 s resolution. Data were binned into 10-meter intervals of geopotential height and all measurements within each bin were averaged. No additional QC was performed beyond Vaisala's standard radiosonde processing. |
| timeSeries ProfileSonde Variables | RS92 / DFM-09 | Vaisala / GRAW | Eastward Wind (m s$^{-1}$) | Standard radiosonde launch | 6 hr | 0.15 ms$^{-1}$ | Vaisala also processed the raw data feed from the radiosonde observations, which was obtained at 2 s resolution. Data were binned into 10-meter intervals of geopotential height and all measurements within each bin were averaged. No additional QC was performed beyond Vaisala's standard radiosonde processing. |

| timeSeries ProfileSonde Variables | RS92 / DFM-09 | Vaisala / GRAW | Northward Wind (m s$^{-1}$) | Standard launch | radiosonde | 6 hr | 0.15 ms$^{-1}$ | Vaisala also processed the raw data feed from the radiosonde observations, which was obtained at 2 s resolution. Data were binned into 10-meter intervals of geopotential height and all measurements within each bin were averaged. No additional QC was performed beyond Vaisala's standard radiosonde processing. |
|---|---|---|---|---|---|---|---|---|
| timeSeries ProfileSonde Variables | RS92 / DFM-09 | Vaisala / GRAW | Temperature (K) | Standard launch | radiosonde | 6 hr | 0.15 K | Vaisala also processed the raw data feed from the radiosonde observations, which was obtained at 2 s resolution. Data were binned into 10-meter intervals of geopotential height and all measurements within each bin were averaged. No additional QC was performed beyond Vaisala's standard radiosonde processing. |
| timeSeries ProfileSonde Variables | RS92 / DFM-09 | Vaisala / GRAW | Dew-point Temperature (K) | Standard launch | radiosonde | 6 hr | 0.5 K | Vaisala also processed the raw data feed from the radiosonde observations, which was obtained at 2 s resolution. Data were binned into 10-meter intervals of geopotential height and all measurements within each bin were averaged. No additional QC was performed beyond Vaisala's standard radiosonde processing. |




**Table 5.** List of the instruments that contributed to the Iqaluit MODF, including details about the instrument manufacturer, measured variables, configuration, temporal resolution, measurement uncertainty, and quality control applied.

| MODF featureType | Instrument | Manufacturer | Measured variables | Instrument Configuration | Temporal Resolution | Uncertainty (+/-) | Quality Control |
|---|---|---|---|---|---|---|---|
| timeSeries Variables | PTB110 | Vaisala | Pressure (Pa) | Installed within a naturally vented protective enclosure. | 1 min | 0.3 hPa | Beyond the standard Vaisala processing, observations were checked against site-based climatology ranges and the rate of change thresholds, which were based on hourly criteria. Observations that fell outside of the 3-sigma normal climatological range were rejected, as were observations that had a rate of change greater than a seasonal-dependant threshold (e.g., >20 hPa/hr change). |
| timeSeries Variables | Pluvio2 | OTT | Total precipitation of water in all phases per unit area ($kg\ m^{-2}\ s^{-1}$) | Single Alter shield | 1 min | 5% | Beyond the standard Vaisala processing, observations were checked against site-based climatology ranges and the rate of change thresholds, which were based on hourly criteria. Observations that fell outside of the 3-sigma normal climatological range were rejected, as were observations that had a rate of change greater than a seasonal-dependant threshold (e.g., > 10 mm/hr change). No corrections for solid precipitation under-catchment were performed (the dataset is raw in the MODF); where appropriate, users are recommended to process under-catchment corrections via Kochendorfer et al. (2020). |
| timeSeries Variables | Wind monitor 5103 | RM Young | Eastward Wind ($m\ s^{-1}$) | Four-blade helicoid propeller in standard configuration with a wind vane to measure wind direction | 1 min | 0.3 $ms^{-1}$ | Beyond the standard Vaisala processing, observations were checked against site-based climatology ranges and the rate of change thresholds, which were based on hourly criteria. Observations that fell outside of the 3-sigma normal climatological range were rejected, as were observations that had a rate of change greater than a seasonal-dependant threshold (e.g., > 10 m/s/hr change). |
| timeSeries Variables | Wind monitor 5103 | RM Young | Northward Wind ($m\ s^{-1}$) | Four-blade helicoid propeller in standard configuration with a wind vane to measure wind direction | 1 min | 0.3 $ms^{-1}$ | Beyond the standard Vaisala processing, observations were checked against site-based climatology ranges and the rate of change thresholds, which were based on hourly criteria. Observations that fell outside of the 3-sigma normal climatological range were rejected, as were observations that had a rate of change greater than a seasonal-dependant threshold (e.g., > 10 m/s/hr change). |
| timeSeries Variables | HMP35D | Vaisala | Temperature (K) | Sensor installed in shaded, naturally vented shelter. | 1 min | 0.1 K | The shelter heating effect is uncorrected beyond the Vaisala standard processing. Beyond the standard Vaisala processing, observations were checked against site-based climatology ranges and the rate of change thresholds, which were based on hourly criteria. Observations that fell outside of the 3-sigma normal climatological range were rejected, as were observations that |

had a rate of change greater than a seasonal-dependant threshold (e.g., > 5 K/hr change).

| | | | | | | | |
|---|---|---|---|---|---|---|---|
| timeSeries Variables | HMP35 D | Vaisala | Dew-point Temperature (K) | Sensor installed in shaded, naturally vented shelter. | 1 min | 0.2 K | The shelter heating effect is uncorrected and humidity is not corrected in a sub-freezing environment, beyond the standard Vaisala processing. Beyond the standard Vaisala processing, observations were checked against site-based climatology ranges and the rate of change thresholds, which were based on hourly criteria. Observations that fell outside of the 3-sigma normal climatological range were rejected, as were observations that had a rate of change greater than a seasonal-dependant threshold (e.g., > 5 K/hr change). |
| timeSeries Variables | HMP35 D | Vaisala | Relative Humidity (1 or %) | Sensor installed in shaded, naturally vented shelter. | 1 min | 0.8% | The humidity is not corrected in a sub-freezing environment, beyond the standard Vaisala processing. Beyond the standard Vaisala processing, observations were checked against site-based climatology ranges and the rate of change thresholds, which were based on hourly criteria. Observations that fell outside of the 3-sigma normal climatological range were rejected, as were observations that had a rate of change greater than a seasonal-dependant threshold (e.g., > 30 %/hr change). |
| timeSeries Variables | SR50A | Campbell Scientific | Snow thickness (m) | Sonic distance sensor at 50KHz with a perforated flat target base levelled at the surface (0 m a.g.l.) | 1 min | 1 cm | Observations were checked against site-based climatology ranges and the rate of change thresholds, which were based on hourly criteria. Observations that fell outside of the 3-sigma normal climatological range were rejected, as were observations that had a rate of change greater than a seasonal-dependant threshold (e.g., > 20 cm/hr change). |
| timeSeries Variables | CMP10 L | Kipp and Zonen | Upward Short-wave Radiation (W m$^{-2}$) | Integrated levelling included, dome, RM Young radiation shield (6 plate), and a CVF4L Ventilation System with Integrated Heater running when temperatures where near zero to prevent frost | 1 min | 7 W m$^{-2}$ | Data is raw and no additional QC was performed, other than the processing performed by Kipp and Zonen. No additional QC was performed on these observations to account for potential frost or snow deposition on the sensors. Data should be treated with caution since they typically require additional QC processing prior to analysis. |
| timeSeries Variables | CMP10 L | Kipp and Zonen | Downward Short-wave Radiation (W m$^{-2}$) | Integrated levelling included, dome, RM Young radiation shield (6 plate), and a CVF4L Ventilation System with Integrated Heater running when temperatures where | 1 min | 7 W m$^{-2}$ | Data is raw and no additional QC was performed, other than the processing performed by Kipp and Zonen. No additional QC was performed on these observations to account for potential frost or snow deposition on the sensors. Data should be treated with caution |

| | | | | | | | |
|---|---|---|---|---|---|---|---|
| | | | | near zero to prevent frost | | | since they typically require additional QC processing prior to analysis. |
| timeSeries Variables | CGR4L | Kipp and Zonen | Upward Long-wave Radiation (W m$^{-2}$) | Integrated levelling included, dome, RM Young radiation shield (6 plate), and a CVF4L Ventilation System with Integrated Heater running when temperatures where near zero to prevent frost | 1 min | 7 W m$^{-2}$ | Data is raw and no additional QC was performed, other than the processing performed by Kipp and Zonen. No additional QC was performed on these observations to account for potential frost or snow deposition on the sensors. Data should be treated with caution since they typically require additional QC processing prior to analysis. |
| timeSeries Variables | CGR4L | Kipp and Zonen | Downward Long-wave Radiation (W m$^{-2}$) | Integrated levelling included, dome, RM Young radiation shield (6 plate), and a CVF4L Ventilation System with Integrated Heater running when temperatures where near zero to prevent frost | 1 min | 7 W m$^{-2}$ | Data is raw and no additional QC was performed, other than the processing performed by Kipp and Zonen. No additional QC was performed on these observations to account for potential frost or snow deposition on the sensors. Data should be treated with caution since they typically require additional QC processing prior to analysis. |
| timeSeries Variables | CGR4L | Kipp and Zonen | *Horizontal East-facing Long-wave Radiation (W m$^{-2}$) | Integrated levelling included, dome, RM Young radiation shield (6 plate), and a CVF4L Ventilation System with Integrated Heater running when temperatures where near zero to prevent frost | 1 min | 7 W m$^{-2}$ | Data is raw and no additional QC was performed, other than the processing performed by Kipp and Zonen. No additional QC was performed on these observations to account for potential frost or snow deposition on the sensors. Data should be treated with caution since they typically require additional QC processing prior to analysis. |
| timeSeries Variables | CGR4L | Kipp and Zonen | *Horizontal West-facing Long-wave Radiation (W m$^{-2}$) | Integrated levelling included, dome, RM Young radiation shield (6 plate), and a CVF4L Ventilation System with Integrated Heater running when temperatures where near zero to prevent frost | 1 min | 7 W m$^{-2}$ | Data is raw and no additional QC was performed, other than the processing performed by Kipp and Zonen. No additional QC was performed on these observations to account for potential frost or snow deposition on the sensors. Data should be treated with caution since they typically require additional QC processing prior to analysis. |
| timeSeries Variables | CGR4L | Kipp and Zonen | *Horizontal South-facing Long-wave Radiation (W m$^{-2}$) | Integrated levelling included, dome, RM Young radiation shield (6 plate), and a CVF4L Ventilation System with Integrated Heater running when temperatures where near zero to prevent frost | 1 min | 7 W m$^{-2}$ | Data is raw and no additional QC was performed, other than the processing performed by Kipp and Zonen. No additional QC was performed on these observations to account for potential frost or snow deposition on the sensors. Data should be treated with caution since they typically require additional QC processing prior to analysis. |

| | | | | | | | |
|---|---|---|---|---|---|---|---|
| timeSeries Variables | CGR4L | Kipp and Zonen | *Horizontal North-facing Long-wave Radiation (W m$^{-2}$) | Integrated levelling included, dome, RM Young radiation shield (6 plate), and a CVF4L Ventilation System with Integrated Heater running when temperatures where near zero to prevent frost | 1 min | 7 W m$^{-2}$ | Data is raw and no additional QC was performed, other than the processing performed by Kipp and Zonen. No additional QC was performed on these observations to account for potential frost or snow deposition on the sensors. Data should be treated with caution since they typically require additional QC processing prior to analysis. |
| timeSeries Profile Variables | CL51 | Vaisala | Cloud Base Height (m) | Proprietary algorithm determines the lowest cloud base height | 1 min | 5 m | No QC was performed, beyond the standard Vaisala proprietary algorithm that retrieves cloud base height. |
| timeSeries Profile Variables | WXT520 | Vaisala | Atmospheric pressure (Pa) | Solid-state, all-in-one weather instrument in standard aspirated configuration mounted on a pole. No bird spike kit used. | 1 min | 0.5 hPa | Beyond the standard Vaisala processing, observations were checked against site-based climatology ranges and the rate of change thresholds, which were based on hourly criteria. Observations that fell outside of the 3-sigma normal climatological range were rejected, as were observations that had a rate of change greater than a seasonal-dependant threshold (e.g., >20 hPa/hr change). |
| timeSeries Profile Variables | WXT520 | Vaisala | Total precipitation of water in all phases per unit area (kg m$^{-2}$ s$^{-1}$) | Solid-state, all-in-one weather instrument in standard aspirated configuration mounted on a pole. No bird spike kit used. | 1 min | 5% | Beyond the standard Vaisala processing, observations were checked against site-based climatology ranges and the rate of change thresholds, which were based on hourly criteria. Observations that fell outside of the 3-sigma normal climatological range were rejected, as were observations that had a rate of change greater than a seasonal-dependant threshold (e.g., > 10 mm/hr change). No corrections for solid precipitation under-catchment were performed (the dataset is raw in the MODF); where appropriate, users are recommended to process under-catchment corrections via Kochendorfer et al. (2020) (note: undercatchment is less of an issue for the WXT520 observations compared to Pluvio2). |
| timeSeries Profile Variables | WXT520 | Vaisala | Eastward Wind (m s$^{-1}$) | Solid-state, all-in-one weather instrument in standard aspirated configuration mounted on a pole. No bird spike kit used. | 1 min | 0.3 ms$^{-1}$ | Beyond the standard Vaisala processing, observations were checked against site-based climatology ranges and the rate of change thresholds, which were based on hourly criteria. Observations that fell outside of the 3-sigma normal climatological range were rejected, as were observations that had a rate of change greater than a seasonal-dependant threshold (e.g., > 10 m/s/hr change). |
| timeSeries Profile Variables | WXT520 | Vaisala | Northward Wind (m s$^{-1}$) | Solid-state, all-in-one weather instrument in standard aspirated configuration mounted on a pole. No bird spike kit used. | 1 min | 0.3 ms$^{-1}$ | Beyond the standard Vaisala processing, observations were checked against site-based climatology ranges and the rate of change thresholds, which were based on hourly criteria. Observations that fell outside of the 3-sigma normal climatological range were rejected, as were observations that had a rate of change |

| | | | | | | | |
|---|---|---|---|---|---|---|---|
| | | | | | | | greater than a seasonal-dependant threshold (e.g., > 10 m/s/hr change). |
| timeSeries Profile Variables | WXT520 | Vaisala | Temperature (K) | Solid-state, all-in-one weather instrument in standard aspirated configuration mounted on a pole. No bird spike kit used. | 1 min | 0.3 K | The shelter heating effect is uncorrected beyond the Vaisala standard processing. Beyond the standard Vaisala processing, observations were checked against site-based climatology ranges and the rate of change thresholds, which were based on hourly criteria. Observations that fell outside of the 3-sigma normal climatological range were rejected, as were observations that had a rate of change greater than a seasonal-dependant threshold (e.g., > 5 K/hr change). |
| timeSeries Profile Variables | WXT520 | Vaisala | Relative Humidity (1 or %) | Solid-state, all-in-one weather instrument in standard aspirated configuration mounted on a pole. No bird spike kit used. | 1 min | 3% | The humidity is not corrected in a sub-freezing environment, beyond the standard Vaisala processing. Beyond the standard Vaisala processing, observations were checked against site-based climatology ranges and the rate of change thresholds, which were based on hourly criteria. Observations that fell outside of the 3-sigma normal climatological range were rejected, as were observations that had a rate of change greater than a seasonal-dependant threshold (e.g., > 30 %/hr change). |
| timeSeries ProfileSonde Variables | RS92 / DFM-09 | Vaisala / GRAW | Atmospheric pressure (Pa) | Standard radiosonde launch | 6 hr | 0.5 hPa | Vaisala also processed the raw data feed from the radiosonde observations, which was obtained at 2 s resolution. Data were binned into 10-meter intervals of geopotential height and all measurements within each bin were averaged. No additional QC was performed beyond Vaisala's standard radiosonde processing. |
| timeSeries ProfileSonde Variables | RS92 / DFM-09 | Vaisala / GRAW | Eastward Wind (m s$^{-1}$) | Standard radiosonde launch | 6 hr | 0.15 ms$^{-1}$ | Vaisala also processed the raw data feed from the radiosonde observations, which was obtained at 2 s resolution. Data were binned into 10-meter intervals of geopotential height and all measurements within each bin were averaged. No additional QC was performed beyond Vaisala's standard radiosonde processing. |
| timeSeries ProfileSonde Variables | RS92 / DFM-09 | Vaisala / GRAW | Northward Wind (m s$^{-1}$) | Standard radiosonde launch | 6 hr | 0.15 ms$^{-1}$ | Vaisala also processed the raw data feed from the radiosonde observations, which was obtained at 2 s resolution. Data were binned into 10-meter intervals of geopotential height and all measurements within each bin were averaged. No additional QC was performed beyond Vaisala's standard radiosonde processing. |

| timeSeries ProfileSonde Variables | RS92 / DFM-09 | Vaisala / GRAW | Temperature (K) | Standard launch | radiosonde | 6 hr | 0.15 K | Vaisala also processed the raw data feed from the radiosonde observations, which was obtained at 2 s resolution. Data were binned into 10-meter intervals of geopotential height and all measurements within each bin were averaged. No additional QC was performed beyond Vaisala's standard radiosonde processing. |
|---|---|---|---|---|---|---|---|---|
| timeSeries ProfileSonde Variables | RS92 / DFM-09 | Vaisala / GRAW | Dew-point Temperature (K) | Standard launch | radiosonde | 6 hr | 0.5 K | Vaisala also processed the raw data feed from the radiosonde observations, which was obtained at 2 s resolution. Data were binned into 10-meter intervals of geopotential height and all measurements within each bin were averaged. No additional QC was performed beyond Vaisala's standard radiosonde processing. |




**Table 6.** List of the instruments that contributed to the Sodankylä MODF, including details about the instrument manufacturer,
measured variables, configuration, temporal resolution, measurement uncertainty, and quality control applied.

| MODF featureType | Instrument | Manufacturer | Measured variables | Instrument Configuration | Temporal Resolution | Uncertainty (+/-) | Quality Control |
|---|---|---|---|---|---|---|---|
| timeSeries Variables | PT100 | Vaisala | Temperature (K) | Sensor installed in shaded, naturally vented shelter. | 10 min | 0.1 K | The shelter heating effect is uncorrected beyond the Vaisala standard processing. Beyond the standard Vaisala processing, observations were checked against site-based climatology ranges and the rate of change thresholds, which were based on hourly criteria. Observations that fell outside of the 3-sigma normal climatological range were rejected, as were observations that had a rate of change greater than a seasonal-dependant threshold (e.g., > 5 K/hr change). |
| timeSeries Variables | PT100 | generic | Temperature (K) | Sensor installed in shaded, naturally vented shelter. | 10 min | 0.3 K | The shelter heating effect is uncorrected beyond the Vaisala standard processing. Beyond the standard Vaisala processing, observations were checked against site-based climatology ranges and the rate of change thresholds, which were based on hourly criteria. Observations that fell outside of the 3-sigma normal climatological range were rejected, as were observations that had a rate of change greater than a seasonal-dependant threshold (e.g., > 5 K/hr change). |
| timeSeries Variables | PT100 | Pentronic | Temperature (K) | Sensor installed in shaded, naturally vented shelter. | 10 min | 0.3 K | The shelter heating effect is uncorrected beyond the Vaisala standard processing. Beyond the standard Vaisala processing, observations were checked against site-based climatology ranges and the rate of change thresholds, which were based on hourly criteria. Observations that fell outside of the 3-sigma normal climatological range were rejected, as were observations that had a rate of change greater than a seasonal-dependant threshold (e.g., > 5 K/hr change). |
| timeSeries Variables | HMP155 | Vaisala | Temperature (K) | Sensor installed in shaded, naturally vented shelter. | 10 min | 0.1 K | The shelter heating effect is uncorrected beyond the Vaisala standard processing. Beyond the standard Vaisala processing, observations were checked against site-based climatology ranges and the rate of change thresholds, which were based on hourly criteria. Observations that fell outside of the 3-sigma normal climatological range were rejected, as were observations that had a rate of change greater than a seasonal-dependant threshold (e.g., > 5 K/hr change). |

| | | | | | | | |
|---|---|---|---|---|---|---|---|
| timeSeries Variables | HMP155 | Vaisala | Relative Humidity (1 or %) | Sensor installed in shaded, naturally vented shelter. | 10 min | 1% | The humidity is not corrected in a sub-freezing environment, beyond the standard Vaisala processing. Beyond the standard Vaisala processing, observations were checked against site-based climatology ranges and the rate of change thresholds, which were based on hourly criteria. Observations that fell outside of the 3-sigma normal climatological range were rejected, as were observations that had a rate of change greater than a seasonal-dependant threshold (e.g., > 30 %/hr change). |
| timeSeries Variables | HMP35D | Vaisala | Relative Humidity (1 or %) | Sensor installed in shaded, naturally vented shelter. | 10 min | 0.8% | The humidity is not corrected in a sub-freezing environment, beyond the standard Vaisala processing. Beyond the standard Vaisala processing, observations were checked against site-based climatology ranges and the rate of change thresholds, which were based on hourly criteria. Observations that fell outside of the 3-sigma normal climatological range were rejected, as were observations that had a rate of change greater than a seasonal-dependant threshold (e.g., > 30 %/hr change). |
| timeSeries Variables | HMP45D | Vaisala | Relative Humidity (1 or %) | Sensor installed in shaded, naturally vented shelter. | 10 min | 2% (0-90 %RH) 3% (90-100 %RH) | The humidity is not corrected in a sub-freezing environment, beyond the standard Vaisala processing. Beyond the standard Vaisala processing, observations were checked against site-based climatology ranges and the rate of change thresholds, which were based on hourly criteria. Observations that fell outside of the 3-sigma normal climatological range were rejected, as were observations that had a rate of change greater than a seasonal-dependant threshold (e.g., > 30 %/hr change). |
| timeSeries Variables | SR50 | Campbell Scientific | Snow thickness (m) | Sonic distance sensor at 50KHz with a perforated flat target base levelled at the surface (0 m a.g.l.) | 10 min | 1 cm | Observations were checked against site-based climatology ranges, routine manual obervations, and the rate of change thresholds, which were based on hourly criteria. Observations that fell outside of the 3-sigma normal climatological range were rejected, as were observations that had a rate of change greater than a seasonal-dependant threshold (e.g., > 20 cm/hr change). |
| timeSeries Variables | Distrometer Model: 5.4110.01.200 | Thies Clima | Total precipitation of water in all phases per unit area (kg $m^{-2}$ $s^{-1}$) | Model with extended heating | 1 min | 5% | Beyond standard processing, observations were checked against site-based climatology ranges and the rate of change thresholds, which were based on hourly criteria. Observations that fell outside of the 3-sigma normal climatological range were rejected, as were observations that had a rate of change greater than a seasonal-dependant threshold (e.g., > 10 mm/hr change). |

| | | | | | | | |
|---|---|---|---|---|---|---|---|
| timeSeries Variables | Distrometer Model: 5.4110.01.200 | Thies Clima | Snowfall flux unit area (kg m$^{-2}$ s$^{-1}$) | Model with extended heating | 1 min | 5% | Beyond standard processing, observations were checked against site-based climatology ranges and the rate of change thresholds, which were based on hourly criteria. Observations that fell outside of the 3-sigma normal climatological range were rejected, as were observations that had a rate of change greater than a seasonal-dependant threshold (e.g., > 10 mm/hr change). |
| timeSeries Variables | SSG 1000 | Sommer Messtechnik | Snow water equivalent (m) | Sensor consists of seven perforated panels having a total measuring surface of 2.8 x 2.4 m with the measurement being made on the centre plate, | 1 min | 0.3% | Data is raw and no additional QC was performed, other than the processing performed by the sensor. |
| timeSeries Variables | CMA11 | Kipp and Zonen | Downward Short-wave Radiation (W m$^{-2}$) | Integrated levelling included, dome, RM Young radiation shield (6 plate), and a CVF4L Ventilation System with Integrated Heater running when temperatures where near zero to prevent frost | 10 min | 7 W m$^{-2}$ | Data is raw and no additional QC was performed, other than the processing performed by Kipp and Zonen. No additional QC was performed on these observations to account for potential frost or snow deposition on the sensors. Data should be treated with caution since they typically require additional QC processing prior to analysis. |
| timeSeries Variables | CMA11 | Kipp and Zonen | Upward Short-wave Radiation (W m$^{-2}$) | Integrated levelling included, dome, RM Young radiation shield (6 plate), and a CVF4L Ventilation System with Integrated Heater running when temperatures where near zero to prevent frost | 10 min | 7 W m$^{-2}$ | Data is raw and no additional QC was performed, other than the processing performed by Kipp and Zonen. No additional QC was performed on these observations to account for potential frost or snow deposition on the sensors. Data should be treated with caution since they typically require additional QC processing prior to analysis. |
| timeSeries Variables | CM11 | Kipp and Zonen | Downward Short-wave Radiation (W m$^{-2}$) | Integrated levelling included, dome, RM Young radiation shield (6 plate), and a CVF4L Ventilation System with Integrated Heater running when temperatures where near zero to prevent frost | 1 min | 7 W m$^{-2}$ | Data is raw and no additional QC was performed, other than the processing performed by Kipp and Zonen. No additional QC was performed on these observations to account for potential frost or snow deposition on the sensors. Data should be treated with caution since they typically require additional QC processing prior to analysis. |
| timeSeries Variables | CMP3 | Kipp and Zonen | Downward Short-wave Radiation (W m$^{-2}$) | Installed on a pole | 10 min | 15 W m$^{-2}$ | Data is raw and no additional QC was performed, other than the processing performed by Kipp and Zonen. No additional QC was performed on these observations to account for potential frost or snow deposition on the sensors. Data should be treated with |

| | | | | | | | |
|---|---|---|---|---|---|---|---|
| | | | | | | | caution since they typically require additional QC processing prior to analysis. |
| timeSeries Variables | CMP3 | Kipp and Zonen | Upward Short-wave Radiation (W m$^{-2}$) | Installed on a pole | 10 min | 15 W m$^{-2}$ | Data is raw and no additional QC was performed, other than the processing performed by Kipp and Zonen. No additional QC was performed on these observations to account for potential frost or snow deposition on the sensors. Data should be treated with caution since they typically require additional QC processing prior to analysis. |
| timeSeries Variables | CMP11 | Kipp and Zonen | Upward Short-wave Radiation (W m$^{-2}$) | Installed on a pole | 10 min | 7 W m$^{-2}$ | Data is raw and no additional QC was performed, other than the processing performed by Kipp and Zonen. No additional QC was performed on these observations to account for potential frost or snow deposition on the sensors. Data should be treated with caution since they typically require additional QC processing prior to analysis. |
| timeSeries Variables | CNR4 | Kipp and Zonen | Downward Short-wave Radiation (W m$^{-2}$) | Integrated 4-component system with temperature sensor | 10 min | 7 W m$^{-2}$ | Data is raw and no additional QC was performed, other than the processing performed by Kipp and Zonen. No additional QC was performed on these observations to account for potential frost or snow deposition on the sensors. Data should be treated with caution since they typically require additional QC processing prior to analysis. |
| timeSeries Variables | CNR4 | Kipp and Zonen | Upward Short-wave Radiation (W m$^{-2}$) | Integrated 4-component system with temperature sensor | 10 min | 7 W m$^{-2}$ | Data is raw and no additional QC was performed, other than the processing performed by Kipp and Zonen. No additional QC was performed on these observations to account for potential frost or snow deposition on the sensors. Data should be treated with caution since they typically require additional QC processing prior to analysis. |
| timeSeries Variables | CNR4 | Kipp and Zonen | Downward Long-wave Radiation (W m$^{-2}$) | Integrated 4-component system with temperature sensor | 10 min | 7 W m$^{-2}$ | Data is raw and no additional QC was performed, other than the processing performed by Kipp and Zonen. No additional QC was performed on these observations to account for potential frost or snow deposition on the sensors. Data should be treated with caution since they typically require additional QC processing prior to analysis. |
| timeSeries Variables | CNR4 | Kipp and Zonen | Upward Long-wave Radiation (W m$^{-2}$) | Integrated 4-component system with temperature sensor | 10 min | 7 W m$^{-2}$ | Data is raw and no additional QC was performed, other than the processing performed by Kipp and Zonen. No additional QC was performed on these observations to account for potential frost or snow deposition on the sensors. Data should be treated with caution since they typically require additional QC processing prior to analysis. |

| | | | | | | | |
|---|---|---|---|---|---|---|---|
| timeSeries Variables | NR-Lite | Kipp and Zonen | Net Short-wave Radiation (W m$^{-2}$) | Single-component thermopile net radiometer | 10 min | 25 W m$^{-2}$ | Data is raw and no additional QC was performed, other than the processing performed by Kipp and Zonen. No additional QC was performed on these observations to account for potential frost or snow deposition on the sensors. Data should be treated with caution since they typically require additional QC processing prior to analysis. |
| timeSeries Variables | NR-Lite2 | Kipp and Zonen | Net Short-wave Radiation (W m$^{-2}$) | Single-component thermopile net radiometer | 10 min | 15 W m$^{-2}$ | Data is raw and no additional QC was performed, other than the processing performed by Kipp and Zonen. No additional QC was performed on these observations to account for potential frost or snow deposition on the sensors. Data should be treated with caution since they typically require additional QC processing prior to analysis. |
| timeSeries Variables | PAR Lite | Kipp and Zonen | Photosynthetic Photon Flux density (mol m$^{-2}$ s$^{-1}$) | Quantum sensor | 10 min | 10% | Data is raw and no additional QC was performed, other than the processing performed by Kipp and Zonen. No additional QC was performed on these observations to account for potential frost or snow deposition on the sensors. Data should be treated with caution since they typically require additional QC processing prior to analysis. |
| timeSeries Variables | PQS1 | Kipp and Zonen | Photosynthetic Photon Flux density (mol m$^{-2}$ s$^{-1}$) | Quantum sensor | 10 min | 5% | Data is raw and no additional QC was performed, other than the processing performed by Kipp and Zonen. No additional QC was performed on these observations to account for potential frost or snow deposition on the sensors. Data should be treated with caution since they typically require additional QC processing prior to analysis. |
| timeSeries Variables | LI190SZ | Licor | Photosynthetic Photon Flux density (mol m$^{-2}$ s$^{-1}$) | Quantum sensor | 10 min | 5% | Data is raw and no additional QC was performed, other than the processing performed by Licor. No additional QC was performed on these observations to account for potential frost or snow deposition on the sensors. Data should be treated with caution since they typically require additional QC processing prior to analysis. |
| timeSeries Variables | PTB201A | Vaisala | Pressure (Pa) | Installed within a naturally vented protective enclosure. | 10 min | 0.3 hPa | Beyond the standard Vaisala processing, observations were checked against site-based climatology ranges and the rate of change thresholds, which were based on hourly criteria. Observations that fell outside of the 3-sigma normal climatological range were rejected, as were observations that had a rate of change greater than a seasonal-dependant threshold (e.g., >20 hPa/hr change). |
| timeSeries Variables | FD12P | Vaisala | Surface horizontal visibility (m) | Optical forward-scatter sensor installed on a pole | 10 min | 10% | Data is raw and no additional QC was performed, other than the processing performed by the sensor. |

| | | | | | | | |
|---|---|---|---|---|---|---|---|
| timeSeries Variables | WA25 (WAA25 and WAV25) | Vaisala | Eastward Wind (m s$^{-1}$) | Cup anemometer and vane designed for Arctic conditions with integrated heaters to prevent ice buildup | 10 min | 0.3 m s$^{-1}$ | Beyond the standard Vaisala processing, observations were checked against site-based climatology ranges and the rate of change thresholds, which were based on hourly criteria. Observations that fell outside of the 3-sigma normal climatological range were rejected, as were observations that had a rate of change greater than a seasonal-dependant threshold (e.g., > 10 m/s/hr change). |
| timeSeries Variables | WA25 (WAA25 and WAV25) | Vaisala | Northward Wind (m s$^{-1}$) | Cup anemometer and vane designed for Arctic conditions with integrated heaters to prevent ice buildup | 10 min | 0.3 m s$^{-1}$ | Beyond the standard Vaisala processing, observations were checked against site-based climatology ranges and the rate of change thresholds, which were based on hourly criteria. Observations that fell outside of the 3-sigma normal climatological range were rejected, as were observations that had a rate of change greater than a seasonal-dependant threshold (e.g., > 10 m/s/hr change). |
| timeSeries Variables | UA2D | Thies Clima | Eastward Wind (m s$^{-1}$) | 2-D sonic anemometer | 10 min | 2% | Data is raw and no additional QC was performed, other than the processing performed by the sensor. |
| timeSeries Variables | UA2D | Thies Clima | Northward Wind (m s$^{-1}$) | 2-D sonic anemometer | 10 min | 2% | Data is raw and no additional QC was performed, other than the processing performed by the sensor. |
| timeSeries Variables | USA-1 | Metek | Eastward Wind (m s$^{-1}$) | 3-D sonic anemometer | 10 min | 0.1 m s$^{-1}$ | Data is raw and no additional QC was performed, other than the processing performed by the sensor. |
| timeSeries Variables | USA-1 | Metek | Northward Wind (m s$^{-1}$) | 3-D sonic anemometer | 10 min | 0.1 m s$^{-1}$ | Data is raw and no additional QC was performed, other than the processing performed by the sensor. |
| timeSeries Variables | USA-1 | Metek | Vertical velocity (m s$^{-1}$) | 3-D sonic anemometer | 10 min | 0.1 m s$^{-1}$ | Data is raw and no additional QC was performed, other than the processing performed by the sensor. |
| timeSeries Variables | USA-1 | Metek | Surface friction velocity (eddy covariance method) (m s$^{-1}$) | 3-D sonic anemometer | 10 min | 0.1 m s$^{-1}$ | Additional filtering of output from eddy covariance processing not performed |
| timeSeries Variables | USA-1 | Metek | Surface turbulent latent heat flux (eddy covariance method) (W m$^{-2}$) | 3-D sonic anemometer | 10 min | 20% | Additional filtering of output from eddy covariance processing not performed |
| timeSeries Variables | USA-1 | Metek | Surface turbulent sensible heat flux (eddy covariance | 3-D sonic anemometer | 10 min | 20% | Additional filtering of output from eddy covariance processing not performed |

| | | | method) (W m$^{-2}$) | | | | |
|---|---|---|---|---|---|---|---|
| timeSeries Variables | USA-1 | Metek | Surface momentum flux (eddy covariance method) (W m$^{-2}$) | 3-D sonic anemometer | 10 min | 25% | Additional filtering of output from eddy covariance processing not performed |
| timeSeries Variables | HFP01 | Huseflux | Ground heat flux (W m$^{-2}$) | Thermopile buried in soil | 10 min | 3% | Data is raw and no additional QC was performed, other than the processing performed by the sensor. |
| timeSeriesProfile Variables | QMT103 | Vaisala | Bulk soil temperature (K) | Thin steel sheath incorporating sensor, buried in soil | 10 min | 0.3 K | Data is raw and no additional QC was performed, other than the processing performed by the sensor. |
| timeSeriesProfile Variables | Hydra Probe II | Stevens | Bulk soil temperature (K) | 4-needle sensor buried in soil | 10 min | 0.3 K | Data is raw and no additional QC was performed, other than the processing performed by the sensor. |
| timeSeriesProfile Variables | Hydra Probe II | Stevens | Average layer soil moisture (kg m$^{-2}$) | 4-needle sensor buried in soil | 10 min | 5% | Data is raw and no additional QC was performed, other than the processing performed by the sensor. |
| timeSeriesProfile Variables | GS3 | Decagon Devices | Bulk soil temperature (K) | Sensor encapsulated in an epoxy body with stainless steel needles. Buried in soil | 10 min | 1 K | Data is raw and no additional QC was performed, other than the processing performed by the sensor. |
| timeSeriesProfile Variables | GTE | Decagon Devices | Bulk soil temperature (K) | Sensor encapsulated in an epoxy body with stainless steel needles. Buried in soil | 10 min | 1 K | Data is raw and no additional QC was performed, other than the processing performed by the sensor. |
| timeSeriesProfile Variables | 109-L | Campbell Scientific | Bulk soil temperature (K) | Thermistor encapsulated in an epoxy-filled aluminum housing and buried in soil | 10 min | 0.3 K | Data is raw and no additional QC was performed, other than the processing performed by the sensor. |
| timeSeriesProfile Variables | CS655 | Campbell Scientific | Bulk soil temperature (K) | Two 12-cm-long stainless steel rods connected to a printed circuit board encapsulated in epoxy attached to a shielded cable. Buried in soil | 10 min | 0.3 K | Data is raw and no additional QC was performed, other than the processing performed by the sensor. |
| timeSeriesProfile Variables | PT100 | Pentronic | Bulk soil temperature (K) | Thin steel sheath incorporating sensor, buried in soil | 10 min | 0.3 K | Data is raw and no additional QC was performed, other than the processing performed by the sensor. |
| timeSeriesProfile Variables | IKES PT100 | Nokeval | Bulk soil temperature (K) | Thin steel sheath incorporates a Pt100 sensor with double insulation moulded in | 10 min | 0.3 K | Data is raw and no additional QC was performed, other than the processing performed by the sensor. |

| | | | | solid rubber with the cable. Buried in soil | | | |
|---|---|---|---|---|---|---|---|
| timeSeriesProfile Variables | ThetaProbe ML2x | Delta-T Devices | Average layer soil moisture (kg m$^{-2}$) | 4-needle sensor buried in soil | 10 min | 5.00% | Data is raw and no additional QC was performed, other than the processing performed by the sensor. |
| timeSeriesProfile Variables | 107-L | Campbell Scientific | Snow temperature (K) | Thermistor encapsulated in an epoxy-filled aluminum housing and buried in snow | 10 min | 0.5 K | Data is raw and no additional QC was performed, other than the processing performed by the sensor. |
| timeSeriesProfile Variables | PT100 | generic | Air temperature (K) | Sensor installed in shaded, naturally vented shelter. | 10 min | 0.3 K | Data is raw and no additional QC was performed, other than the processing performed by the sensor. |
| timeSeriesProfile Variables | HMP | Vaisala | Relative Humidity (1 or %) | Sensor installed in shaded, naturally vented shelter. | 10 min | 0.80% | Data is raw and no additional QC was performed, other than the processing performed by the sensor. |
| timeSeriesProfile Variables | WAA25 | Vaisala | Wind speed (m s$^{-1}$) | Cup anemometer with integrated heater to prevent ice buildup | 10 min | 0.17 m s$^{-1}$ | Data is raw and no additional QC was performed, other than the processing performed by the sensor. |




**Table 7.** List of the instruments that contributed to the Utqiaġvik MODF, including details about the instrument manufacturer, measured variables, configuration, temporal resolution, measurement uncertainty, and quality control applied.

| MODF featureType | Instrument | Manufacturer | Measured variables | Instrument Configuration | Temporal Resolution | Uncertainty (+/-) | Quality Control |
|---|---|---|---|---|---|---|---|
| timeSeries Variables | PTB-220 | Vaisala | Pressure (Pa) | The Barrow meteorology station (BMET) uses mainly conventional in situ sensors mounted at four different heights (2m, 10m, 20m and 40m) on a 40 m tower to obtain profiles of wind speed, wind direction, air temperature, dew point and humidity. It also obtains barometric pressure, visibility and precipitation data from sensors at the base of the tower. https://www.arm.gov/capabilities/instruments/twr | 1 min | 0.15 hPa | Beyond the standard Vaisala processing, observations were checked against other instrumentation on the tower and compared with the surface meteorological instruments and the energy balance bowen ratio. Data was also compared with the SONDE data that was launched some distance away from the tower: https://www.arm.gov/publications/tech_reports/handbooks/twr_handbook.pdf |
| timeSeries Variables | WS425 | Vaisala | Near-surface (2m) eastward wind (m s$^{-1}$) | The Barrow meteorology station (BMET) uses mainly conventional in situ sensors mounted at four different heights (2m, 10m, 20m and 40m) on a 40 m tower to obtain profiles of wind speed, wind direction, air temperature, dew point and humidity. It also obtains barometric pressure, visibility and precipitation data from sensors at the base of the tower. https://www.arm.gov/capabilities/instruments/twr | 1 min | 0.135 ms$^{-1}$ | Beyond the standard Vaisala processing, observations were checked against other instrumentation on the tower and compared with the surface meteorological instruments and the energy balance bowen ratio. Data was also compared with the SONDE data that was launched some distance away from the tower: https://www.arm.gov/publications/tech_reports/handbooks/twr_handbook.pdf |

| | | | | | | | | |
|---|---|---|---|---|---|---|---|---|
| timeSeries Variables | WS425 | Vaisala | Near-surface (2m) north ward wind (m s$^{-1}$) | The Barrow meteorology station (BMET) uses mainly conventional in situ sensors mounted at four different heights (2m, 10m, 20m and 40m) on a 40 m tower to obtain profiles of wind speed, wind direction, air temperature, dew point and humidity. It also obtains barometric pressure, visibility and precipitation data from sensors at the base of the tower. https://www.arm.gov/capabilities/instruments/twr | 1 min | 0.135 ms$^{-1}$ | Beyond the standard Vaisala processing, observations were checked against other instrumentation on the tower and compared with the surface meteorological instruments and the energy balance bowen ratio. Data was also compared with the SONDE data that was launched some distance away from the tower: https://www.arm.gov/publications/tech_reports/handbooks/twr_handbook.pdf |
| timeSeries Variables | HMT337 (previous ly HMP35D /HMP45 D) | Vaisala | Near-surface (2m) air tempe rature (K) | The Barrow meteorology station (BMET) uses mainly conventional in situ sensors mounted at four different heights (2m, 10m, 20m and 40m) on a 40 m tower to obtain profiles of wind speed, wind direction, air temperature, dew point and humidity. It also obtains barometric pressure, visibility and precipitation data from sensors at the base of the tower. https://www.arm.gov/capabilities/instruments/twr | 1 min | 0.2 K | Beyond the standard Vaisala processing, observations were checked against other instrumentation on the tower and compared with the surface meteorological instruments and the energy balance bowen ratio. Data was also compared with the SONDE data that was launched some distance away from the tower: https://www.arm.gov/publications/tech_reports/handbooks/twr_handbook.pdf |
| timeSeries Variables | HMT337 (previous ly HMP35D /HMP45 D) | Vaisala | Near-surface (2m) dew point tempe rature (K) | The Barrow meteorology station (BMET) uses mainly conventional in situ sensors mounted at four different heights (2m, 10m, 20m and 40m) on a 40 m tower to obtain profiles of wind speed, wind direction, air temperature, dew point and humidity. It also obtains barometric pressure, visibility and precipitation data from sensors at the base of the tower. https://www.arm.gov/ca | 1 min | 0.2 K | Beyond the standard Vaisala processing, observations were checked against other instrumentation on the tower and compared with the surface meteorological instruments and the energy balance bowen ratio. Data was also compared with the SONDE data that was launched some distance away from the tower: https://www.arm.gov/publications/tech_reports/handbooks/twr_handbook.pdf |

| timeSeries Variables | HMT337 (previously HMP35D /HMP45 D) | Vaisala | Near-surface (2m) relative humidity (%) | The Barrow meteorology station (BMET) uses mainly conventional in situ sensors mounted at four different heights (2m, 10m, 20m and 40m) on a 40 m tower to obtain profiles of wind speed, wind direction, air temperature, dew point and humidity. It also obtains barometric pressure, visibility and precipitation data from sensors at the base of the tower. https://www.arm.gov/capabilities/instruments/twr | 1 min | 1.7 % | Beyond the standard Vaisala processing, observations were checked against other instrumentation on the tower and compared with the surface meteorological instruments and the energy balance bowen ratio. Data was also compared with the SONDE data that was launched some distance away from the tower: https://www.arm.gov/publications/tech_reports/handbooks/twr_handbook.pdf |
|---|---|---|---|---|---|---|---|
| timeSeries Variables | TEI 49i | Thermo Scientific | Ozone concentration in air (mole fraction) | Inlet line samples air from roof of station through filter, while instrument is housed inside station building | 1 min | 1 ppb | This data set contains continuous UV photometric data of surface level ozone collected at 6m above ground level. Data records consist of UTC time, date, and processed ozone mixing ratio (parts per billion). Data is collected from global locations and is provided in 1 minute and 1 hour averages. Data are archived at the NOAA National Climatic Data Center (NCDC), but are produced and available from NOAA Earth System Research Laboratory (ESRL). https://www.ncei.noaa.gov/access/metadata/landing-page/bin/iso?id=gov.noaa.ncdc:C00894 |
| timeSeries Variables | Toughsonic 30 | Senix | Surface snow thickness (m) | Instrument is located on broadband radiation albedo rack | 1 min | n/a | Data is compared against meteorological and global radiation data to verify accuracy; pollution/technical events are flagged and/or removed from data set; data values not physically possible are removed |
| timeSeries Variables | IRT | Apogee | Surface (skin) temperature (K) | Data collected from US Climate Reference Network (CRN) | 1 min | 0.5 K | Inter-comparison of the 3 temperature sensors: Sensors should be within 0.3° C of one another. An hourly flag message is generated for any departure greater than 0.30° C (i.e., 0.301° C and greater). IR max should exceed the ambient temperature, and IR min should be less than ambient temperature. https://www1.ncdc.noaa.gov/pub/data/uscrn/documentation/program/ManualMonitoringHandbook.pdf |

| | | | | | | | |
|---|---|---|---|---|---|---|---|
| timeSeries Variables | GNDRAD | PSP | Upward surface short-wave radiation (W m$^{-2}$) | https://www.arm.gov/capabilities/instruments/gndrad | 1 min | 2.0 W m$^{-2}$ | SIRS Instrument mentors review the Data Quality Office's (DQO) weekly Data Quality Assessment Reports (DQAR). If a problem is detected, a Data Quality Problem Report (DQPR) is issued. The DQPR system is a web-based system by which the mentor, local site operations staff, and the DQO are informed and communicate to resolve a data quality problem (e.g., instrument failure, data collection issue, etc.). A DQPR is typically initiated by the DQO or instrument mentor during data review. Data Quality Reports (DQR) are prepared by instrument mentors as needed to close out corresponding DQPRs. https://www.arm.gov/capabilities/instruments/gndrad |
| timeSeries Variables | SKYRAD | PSP | Downward short-wave radiation at the surface (W m$^{-2}$) | https://www.arm.gov/capabilities/instruments/skyrad | 1 min | 4.0 W m$^{-2}$ | SIRS Instrument mentors review the Data Quality Office's (DQO) weekly Data Quality Assessment Reports (DQAR). If a problem is detected, a Data Quality Problem Report (DQPR) is issued. The DQPR system is a web-based system by which the mentor, local site operations staff, and the DQO are informed and communicate to resolve a data quality problem (e.g., instrument failure, data collection issue, etc.). A DQPR is typically initiated by the DQO or instrument mentor during data review. Data Quality Reports (DQR) are prepared by instrument mentors as needed to close out corresponding DQPRs. https://www.arm.gov/capabilities/instruments/skyrad |
| timeSeries Variables | GNDRAD | PIR | Upward surface long-wave radiation (W m$^{-2}$) | https://www.arm.gov/capabilities/instruments/gndrad | 1 min | 2.0 W m$^{-2}$ | SIRS Instrument mentors review the Data Quality Office's (DQO) weekly Data Quality Assessment Reports (DQAR). If a problem is detected, a Data Quality Problem Report (DQPR) is issued. The DQPR system is a web-based system by which the mentor, local site operations staff, and the DQO are informed and communicate to resolve a data quality problem (e.g., instrument failure, data collection issue, etc.). A DQPR is typically initiated by the DQO or instrument mentor during data review. Data Quality Reports (DQR) are prepared by instrument mentors as needed to close out corresponding DQPRs. https://www.arm.gov/capabilities/instruments/gndrad |
| timeSeries Variables | SKYRAD | PIR | Downward surface long-wave radiation (W m$^{-2}$) | https://www.arm.gov/capabilities/instruments/skyrad | 1 min | 4.0 W m$^{-2}$ | SIRS Instrument mentors review the Data Quality Office's (DQO) weekly Data Quality Assessment Reports (DQAR). If a problem is detected, a Data Quality Problem Report (DQPR) is issued. The DQPR system is a web-based system by which the mentor, local site operations staff, and the DQO are informed and communicate to resolve a data quality problem (e.g., instrument failure, data collection issue, etc.). A DQPR is typically initiated by the DQO or instrument mentor during data review. Data Quality Reports (DQR) are prepared by instrument mentors as needed to close out corresponding DQPRs. https://www.arm.gov/capabilities/instruments/skyrad |

| timeSeries Variables | Windmaster Pro Anemometer | Gill | Surface turbulent latent heat flux (eddy covariance method) (W m$^{-2}$) | Standard ARM site arrangement is sonic sensor "North" mark pointing along the boom to the tower; the boom is usually pointing due south; u wind component is north-south with positive toward the north; v wind component is east-west with positive toward the west. NOTE: no correction is made to convert u and v component into meteorological "north" and "east" wind components when tower boom is not aligned to south; u wind component is "along boom", v wind component is "cross boom https://www.arm.gov/publications/tech_reports/doe-sc-arm-tr-223.pdf | 1 min | <1.5 % | The QCECOR VAP currently contains two variables: surface latent heat flux (LH) and sensible heat flux (SH), together with their QC flags. When SEBS are collocated with ECOR, the wetness measurements from SEBS are used to flag the LH that may be incorrect due to hydrometeors such as precipitation, dew, or frost. An indeterminate flag is given to those that fail the wetness test. chrome-extension://efaidnbmnnnibpcajpcglclefindmkaj/https://www.arm.gov/publications/tech_reports/doe-sc-arm-tr-223.pdf |
| timeSeries Variables | Windmaster Pro Anemometer | Gill | Surface turbulent sensible heat flux (eddy covariance method) (W m$^{-2}$) | Standard ARM site arrangement is sonic sensor "North" mark pointing along the boom to the tower; the boom is usually pointing due south; u wind component is north-south with positive toward the north; v wind component is east-west with positive toward the west. NOTE: no correction is made to convert u and v component into meteorological "north" and "east" wind components when tower boom is not aligned to south; u wind component is "along boom", v wind component is "cross boom https://www.arm.gov/publications/tech_reports/doe-sc-arm-tr-223.pdf | 1 min | <1.5 % | The QCECOR VAP currently contains two variables: surface latent heat flux (LH) and sensible heat flux (SH), together with their QC flags. When SEBS are collocated with ECOR, the wetness measurements from SEBS are used to flag the LH that may be incorrect due to hydrometeors such as precipitation, dew, or frost. An indeterminate flag is given to those that fail the wetness test. chrome-extension://efaidnbmnnnibpcajpcglclefindmkaj/https://www.arm.gov/publications/tech_reports/doe-sc-arm-tr-223.pdf |

| timeSeries Variables | HFT-3, SMP1, STP-1 | Radiation and Energy Balance Systems, Inc. | Ground heat flux (W m$^{-2}$) | Soil measurements are performed by three sets of soil heat flow (5 cm depth), soil temperature (0–5 cm average), and soil moisture (centered at 2.5 cm) probes. Soil heat flow is adjusted for the effect of soil moisture above the soil heat flow plate. The storage of energy in the soil above the soil heat flow plate is determined from the change in soil temperature with time. | 1 min | 10 mV | Instrument mentor routinely views graphic displays that include plots (day courses) of all calculated quantities and comparison plots (time series or scatter plots) of relevant parameters with data from collocated ECOR, SEBS, EBBR (SGP CF and EF39 only), and surface meteorological instrumentation (MET) (Cook et al. 2006). chrome-extension://efaidnbmnnnibpcajpcglclefindmkaj/https://www.arm.gov/publications/tech_reports/handbooks/sebs_handbook.pdf |
|---|---|---|---|---|---|---|---|
| timeSeries Profile Variables | WS425 | Vaisala | Eastward wind component (m s$^{-1}$) | The Barrow meteorology station (BMET) uses mainly conventional in situ sensors mounted at four different heights (2m, 10m, 20m and 40m) on a 40 m tower to obtain profiles of wind speed, wind direction, air temperature, dew point and humidity. It also obtains barometric pressure, visibility and precipitation data from sensors at the base of the tower. https://www.arm.gov/capabilities/instruments/twr | 1 min | 0.135 ms$^{-1}$ | Beyond the standard Vaisala processing, observations were checked against other instrumentation on the tower and compared with the surface meteorological instruments and the energy balance bowen ratio. Data was also compared with the SONDE data that was launched some distance away from the tower: https://www.arm.gov/publications/tech_reports/handbooks/twr_handbook.pdf |
| timeSeries Profile Variables | WS425 | Vaisala | Northward wind component (m s$^{-1}$) | The Barrow meteorology station (BMET) uses mainly conventional in situ sensors mounted at four different heights (2m, 10m, 20m and 40m) on a 40 m tower to obtain profiles of wind speed, wind direction, air temperature, dew point and humidity. It also obtains barometric pressure, visibility and precipitation data from sensors at the base of the tower. https://www.arm.gov/capabilities/instruments/twr | 1 min | 0.135 ms$^{-1}$ | Beyond the standard Vaisala processing, observations were checked against other instrumentation on the tower and compared with the surface meteorological instruments and the energy balance bowen ratio. Data was also compared with the SONDE data that was launched some distance away from the tower: https://www.arm.gov/publications/tech_reports/handbooks/twr_handbook.pdf |

| timeSeries Profile Variables | HMT337 (previously HMP35D /HMP45D) | Vaisala | Air temperature (K) | The Barrow meteorology station (BMET) uses mainly conventional in situ sensors mounted at four different heights (2m, 10m, 20m and 40m) on a 40 m tower to obtain profiles of wind speed, wind direction, air temperature, dew point and humidity. It also obtains barometric pressure, visibility and precipitation data from sensors at the base of the tower. https://www.arm.gov/capabilities/instruments/twr | 1 min | 0.2 K | Beyond the standard Vaisala processing, observations were checked against other instrumentation on the tower and compared with the surface meteorological instruments and the energy balance bowen ratio. Data was also compared with the SONDE data that was launched some distance away from the tower: https://www.arm.gov/publications/tech_reports/handbooks/twr_handbook.pdf |
|---|---|---|---|---|---|---|---|
| timeSeries Profile Variables | HMT337 (previously HMP35D /HMP45D) | Vaisala | Dew-point temperature (K) | The Barrow meteorology station (BMET) uses mainly conventional in situ sensors mounted at four different heights (2m, 10m, 20m and 40m) on a 40 m tower to obtain profiles of wind speed, wind direction, air temperature, dew point and humidity. It also obtains barometric pressure, visibility and precipitation data from sensors at the base of the tower. https://www.arm.gov/capabilities/instruments/twr | 1 min | 0.2 K | Beyond the standard Vaisala processing, observations were checked against other instrumentation on the tower and compared with the surface meteorological instruments and the energy balance bowen ratio. Data was also compared with the SONDE data that was launched some distance away from the tower: https://www.arm.gov/publications/tech_reports/handbooks/twr_handbook.pdf |
| timeSeries Profile Variables | HMT337 (previously HMP35D /HMP45D) | Vaisala | Relative humidity (%) | The Barrow meteorology station (BMET) uses mainly conventional in situ sensors mounted at four different heights (2m, 10m, 20m and 40m) on a 40 m tower to obtain profiles of wind speed, wind direction, air temperature, dew point and humidity. It also obtains barometric pressure, visibility and precipitation data from sensors at the base of the tower. https://www.arm.gov/ca | 1 min | 1.7 % | Beyond the standard Vaisala processing, observations were checked against other instrumentation on the tower and compared with the surface meteorological instruments and the energy balance bowen ratio. Data was also compared with the SONDE data that was launched some distance away from the tower: https://www.arm.gov/publications/tech_reports/handbooks/twr_handbook.pdf |

| | | | | | | | | |
|---|---|---|---|---|---|---|---|---|
| timeSeries Profile Variables | PT100 | in-house | Soil tempe rature profil e (K) | Instrument is located on broadband radiation albedo rack | 1 min | n/a | Data is compared against meteorological and global radiation data to verify accuracy; pollution/technical events are flagged and/or removed from data set; data values not physically possible are removed |
| timeSeries Profile Variables | KAZR | KAZR | Snow fall flux per unit area | Installed on top of the ARM facility roof | 1 min | n/a | https://doi.org/10.1525/elementa.2021.00101 chrome-extension://efaidnbmnnnibpcajpcglclefindmkaj/https://www.arm.gov/publications/tech_reports/handbooks/kazr_handbook.pdf |
| timeSeries ProfileSon de Variables | SONDE | Radios onde | Atmo spheri c press ure (Pa) | The SONDE system originally located at Barrow was an old CLASS-type that was originally operated by NOAA's Climate Measurements and Diagnostics Laboratory on TWP's Manus site. | 30 min | 1 hPa | The manufacturer defines the cumulative sensor uncertainty at the 2-sigma (95.5%) confidence level. Repeatability is estimated from the standard deviation of differences between two successive repeated calibrations (2-sigma). Reproducibility is estimated from the standard deviation of differences in twin soundings. Citation recommendation: Atmospheric Radiation Measurement (ARM) user facility. 2002. Balloon-Borne Sounding System (SONDEWNPN). 2002-04-28 to 2022-11-17, North Slope Alaska (NSA) Central Facility, Barrow AK (C1). Compiled by K. Burk. ARM Data Center. Data set accessed 2022-11-18 at http://dx.doi.org/10.5439/1595321. |
| timeSeries ProfileSon de Variables | SONDE | Radios onde | Eastw ard wind comp onent (m s⁻¹) | The SONDE system originally located at Barrow was an old CLASS-type that was originally operated by NOAA's Climate Measurements and Diagnostics Laboratory on TWP's Manus site. | 30 min | n/a | The manufacturer defines the cumulative sensor uncertainty at the 2-sigma (95.5%) confidence level. Repeatability is estimated from the standard deviation of differences between two successive repeated calibrations (2-sigma). Reproducibility is estimated from the standard deviation of differences in twin soundings. Citation recommendation: Atmospheric Radiation Measurement (ARM) user facility. 2002. Balloon-Borne Sounding System (SONDEWNPN). 2002-04-28 to 2022-11-17, North Slope Alaska (NSA) Central Facility, Barrow AK (C1). Compiled by K. Burk. ARM Data Center. Data set accessed 2022-11-18 at http://dx.doi.org/10.5439/1595321. |
| timeSeries ProfileSon de Variables | SONDE | Radios onde | North ward wind comp onent (m s⁻¹) | The SONDE system originally located at Barrow was an old CLASS-type that was originally operated by NOAA's Climate Measurements and Diagnostics Laboratory on TWP's Manus site. | 30 min | n/a | The manufacturer defines the cumulative sensor uncertainty at the 2-sigma (95.5%) confidence level. Repeatability is estimated from the standard deviation of differences between two successive repeated calibrations (2-sigma). Reproducibility is estimated from the standard deviation of differences in twin soundings. Citation recommendation: Atmospheric Radiation Measurement (ARM) user facility. 2002. Balloon-Borne Sounding System (SONDEWNPN). 2002-04-28 to 2022-11-17, North Slope Alaska (NSA) Central Facility, Barrow AK (C1). Compiled by K. Burk. ARM Data Center. Data set accessed 2022-11-18 at http://dx.doi.org/10.5439/1595321. |

| timeSeries ProfileSonde Variables | SONDE | Radiosonde | Temperature (K) | The SONDE system originally located at Barrow was an old CLASS-type that was originally operated by NOAA's Climate Measurements and Diagnostics Laboratory on TWP's Manus site. | 30 min | 0.5 K | The manufacturer defines the cumulative sensor uncertainty at the 2-sigma (95.5%) confidence level. Repeatability is estimated from the standard deviation of differences between two successive repeated calibrations (2-sigma). Reproducibility is estimated from the standard deviation of differences in twin soundings. Citation recommendation: Atmospheric Radiation Measurement (ARM) user facility. 2002. Balloon-Borne Sounding System (SONDEWNPN). 2002-04-28 to 2022-11-17, North Slope Alaska (NSA) Central Facility, Barrow AK (C1). Compiled by K. Burk. ARM Data Center. Data set accessed 2022-11-18 at http://dx.doi.org/10.5439/1595321. |
| timeSeries ProfileSonde Variables | SONDE | Radiosonde | Dew-point temperature (K) | The SONDE system originally located at Barrow was an old CLASS-type that was originally operated by NOAA's Climate Measurements and Diagnostics Laboratory on TWP's Manus site. | 30 min | 0.5 K | The manufacturer defines the cumulative sensor uncertainty at the 2-sigma (95.5%) confidence level. Repeatability is estimated from the standard deviation of differences between two successive repeated calibrations (2-sigma). Reproducibility is estimated from the standard deviation of differences in twin soundings. Citation recommendation: Atmospheric Radiation Measurement (ARM) user facility. 2002. Balloon-Borne Sounding System (SONDEWNPN). 2002-04-28 to 2022-11-17, North Slope Alaska (NSA) Central Facility, Barrow AK (C1). Compiled by K. Burk. ARM Data Center. Data set accessed 2022-11-18 at http://dx.doi.org/10.5439/1595321. |
| timeSeries ProfileSonde Variables | SONDE | Radiosonde | Relative humidity (%) | The SONDE system originally located at Barrow was an old CLASS-type that was originally operated by NOAA's Climate Measurements and Diagnostics Laboratory on TWP's Manus site. | 30 min | 5% | The manufacturer defines the cumulative sensor uncertainty at the 2-sigma (95.5%) confidence level. Repeatability is estimated from the standard deviation of differences between two successive repeated calibrations (2-sigma). Reproducibility is estimated from the standard deviation of differences in twin soundings. Citation recommendation: Atmospheric Radiation Measurement (ARM) user facility. 2002. Balloon-Borne Sounding System (SONDEWNPN). 2002-04-28 to 2022-11-17, North Slope Alaska (NSA) Central Facility, Barrow AK (C1). Compiled by K. Burk. ARM Data Center. Data set accessed 2022-11-18 at http://dx.doi.org/10.5439/1595321. |




**Table 8.** List of the instruments that contributed to the Tiksi MODF, including details about the instrument manufacturer,
measured variables, configuration, temporal resolution, measurement uncertainty, and quality control applied.

| MODF featureType | Instrument | Manufacturer | Measured variables | Instrument Configuration | Temporal Resolution | Uncertainty (+/-) | Quality Control |
|---|---|---|---|---|---|---|---|
| timeSeries Variables | PTB110 | Vaisala | Surface pressure (Pa) | Located on the fluxtower at 5m height | 1 min | 0.3 hPa | Data are manually QC'ed to identify and eliminate instrument malfunction; outliers are filtered out if values are physically impossible; values are compared to other local variables if/when possible |
| timeSeries Variables | 3001 | RM Young | Near-surface (4m) eastward wind (m s$^{-1}$) | Located on the fluxtower at 4m height | 1 min | 0.5 m s$^{-1}$ | Data are manually QC'ed to identify and eliminate instrument malfunction; outliers are filtered out if values are physically impossible; values are compared to other local variables if/when possible |
| timeSeries Variables | 3001 | RM Young | Near-surface (4m) northward wind (m s$^{-1}$) | Located on the fluxtower at 4m height | 1 min | 0.5 m s$^{-1}$ | Data are manually QC'ed to identify and eliminate instrument malfunction; outliers are filtered out if values are physically impossible; values are compared to other local variables if/when possible |
| timeSeries Variables | HMT330 | Vaisala | Near-surface (2m) air temperature (K) | Located on the fluxtower at 2m height | 1 min | 0.2 K | Data are manually QC'ed to identify and eliminate instrument malfunction; outliers are filtered out if values are physically impossible; values are compared to other local variables if/when possible |
| timeSeries Variables | HMT330 | Vaisala | Near-surface (2m) relative humidity (%) | Located on the fluxtower at 2m height | 1 min | 1.5 + 0.015 × reading | Data are manually QC'ed to identify and eliminate instrument malfunction; outliers are filtered out if values are physically impossible; values are compared to other local variables if/when possible |
| timeSeries Variables | SR50A | Campbell Scientific | Surface snow thickness (m) | Located on the albedo rack | 1 min | 1 cm | Data are manually QC'ed to identify and eliminate instrument malfunction; outliers are filtered out if values are physically impossible; values are compared to other local variables if/when possible |
| timeSeries Variables | SI-111 | Apogee | Surface (skin) temperature (K) | Located on the fluxtower at 2m height | 1 min | 0.2 K | Data are manually QC'ed to identify and eliminate instrument malfunction; outliers are filtered out if values are physically impossible; values are compared to |

| | | | | | | | | |
|---|---|---|---|---|---|---|---|---|
| | | | | | | | | other local variables if/when possible |
| timeSeries Variables | PSP | Eppley | Upward surface short-wave radiation (W m$^{-2}$) | Located on the albedo rack | 1 min | 2.0 | W m$^{-2}$ | Data are manually QC'ed to identify and eliminate instrument malfunction; outliers are filtered out if values are physically impossible; values are compared to other local variables if/when possible |
| timeSeries Variables | CM22 | Kipp & Zonen | Downward surface short-wave radiation (W m$^{-2}$) | Located on the tracker at the MET station building | 1 min | 5.0 | W m$^{-2}$ | Data are manually QC'ed to identify and eliminate instrument malfunction; outliers are filtered out if values are physically impossible; values are compared to other local variables if/when possible |
| timeSeries Variables | PIR | Eppley | Upward surface long-wave radiation (W m$^{-2}$) | Located on the albedo rack | 1 min | 2.0 | W m$^{-2}$ | Data are manually QC'ed to identify and eliminate instrument malfunction; outliers are filtered out if values are physically impossible; values are compared to other local variables if/when possible |
| timeSeries Variables | PIR | Eppley | Downward surface long-wave radiation (W m$^{-2}$) | Located on the tracker at the MET station building | 1 min | 4.0 | W m$^{-2}$ | Data are manually QC'ed to identify and eliminate instrument malfunction; outliers are filtered out if values are physically impossible; values are compared to other local variables if/when possible |
| timeSeries Variables | HPF01 | Hukseflux | Ground heat flux (W m$^{-2}$) | Located at the base of the fluxtower at 5cm depth | 1 min | 3 % | | Data are manually QC'ed to identify and eliminate instrument malfunction; outliers are filtered out if values are physically impossible; values are compared to other local variables if/when possible |
| timeSeries Variables | HPF01 | Hukseflux | Ground heat flux (W m$^{-2}$) | Located at the base of the fluxtower at 5cm depth | 1 min | 3 % | | Data are manually QC'ed to identify and eliminate instrument malfunction; outliers are filtered out if values are physically impossible; values are compared to other local variables if/when possible |
| timeSeriesProfile Variables | HMT330, HMP155 | Vaisala | Air temperature (K) | Located on the fluxtower at 2m, 6m, 10m height | 1 min | 0.2 K | | Data are manually QC'ed to identify and eliminate instrument malfunction; outliers are filtered out if values are physically impossible; values are compared to |

| | | | | other local variables if/when possible | | | |
|---|---|---|---|---|---|---|---|
| timeSeriesProfile Variables | HMT330, HMP155 | Vaisala | Relative humidity (%) | Located on the fluxtower at 2m, 6m, 10m height | 1 min | 1.5 + 0.015 × reading | Data are manually QC'ed to identify and eliminate instrument malfunction; outliers are filtered out if values are physically impossible; values are compared to other local variables if/when possible |
| timeSeriesProfile Variables | TP-101 | MRC | Soil temperature profile (K) | Located at albedo rack at depths: 5cm, 10cm, 15cm, 20cm, 25cm, 30cm, 45cm, 70cm, 95cm, 120cm | 1 min | n/a | Data are manually QC'ed to identify and eliminate instrument malfunction; outliers are filtered out if values are physically impossible; values are compared to other local variables if/when possible |
| timeSeriesProfileSonde Variables | SONDE | Radiosonde | Atmospheric pressure (Pa) | https://www.ncei.noaa.gov/pub/data/igra/data/data-por/ | 30 min | 1 hPa | https://www.ncei.noaa.gov/pub/data/igra/data/data-por/ |
| timeSeriesProfileSonde Variables | SONDE | Radiosonde | Eastward wind component (m s$^{-1}$) | https://www.ncei.noaa.gov/pub/data/igra/data/data-por/ | 30 min | n/a | https://www.ncei.noaa.gov/pub/data/igra/data/data-por/ |
| timeSeriesProfileSonde Variables | SONDE | Radiosonde | Northward wind component (m s$^{-1}$) | https://www.ncei.noaa.gov/pub/data/igra/data/data-por/ | 30 min | n/a | https://www.ncei.noaa.gov/pub/data/igra/data/data-por/ |
| timeSeriesProfileSonde Variables | SONDE | Radiosonde | Temperature (K) | https://www.ncei.noaa.gov/pub/data/igra/data/data-por/ | 30 min | 0.5 K | https://www.ncei.noaa.gov/pub/data/igra/data/data-por/ |
| timeSeriesProfileSonde Variables | SONDE | Radiosonde | Dew-point temperature (K) | https://www.ncei.noaa.gov/pub/data/igra/data/data-por/ | 30 min | 0.5 K | https://www.ncei.noaa.gov/pub/data/igra/data/data-por/ |
| timeSeriesProfileSonde Variables | SONDE | Radiosonde | Relative humidity (%) | https://www.ncei.noaa.gov/pub/data/igra/data/data-por/ | 30 min | 5% | https://www.ncei.noaa.gov/pub/data/igra/data/data-por/ |




**Table 9.** List of the instruments that contributed to the Ny-Ålesund MODF, including details about the instrument
manufacturer, measured variables, configuration, temporal resolution, measurement uncertainty, and quality control applied.

| MODF featureType | Instrument | Manufacturer | Measured variables | Instrument Configuration | Temporal Resolution | Uncertainty (+/-) | Quality Control |
|---|---|---|---|---|---|---|---|
| **timeSeries Variables** | Digiquarz 6000-16B | Paroscientific, Inc. | Pressure (Pa) | Installed within a naturally vented protective enclosure. | 1 min | 0.08 hPa | Observations were checked against site-based climatology ranges and the rate of change thresholds. |
| **timeSeries Variables** | Pluvio2 | OTT | Total precipitation of water in all phases per unit area (kg $m^{-2}$ $s^{-1}$) | Single Alter shield | 1 min | 5% | Operated and analysed by the University of Cologne. No additional QC was applied; data is raw and should be treated with caution. |
| **timeSeries Variables** | Combined Wind Transmitter 4.3324.32.073 | Thies Clima | Eastward Wind (m $s^{-1}$) | Opto-electronically scanned three-cup anemometer with low starting speed. The position of the wind vane is detected opto-electronically. | 1 min | 0.4 $ms^{-1}$ | Instrument is checked on a daily basis. Observations were checked against site-based climatology ranges, the rate of change thresholds, and redundant measurements in close proximity. |
| **timeSeries Variables** | Combined Wind Transmitter 4.3324.32.073 | Thies Clima | Northward Wind (m $s^{-1}$) | Opto-electronically scanned three-cup anemometer with low starting speed. The position of the wind vane is detected opto-electronically. | 1 min | 0.4 $ms^{-1}$ | Instrument is checked on a daily basis. Observations were checked against site-based climatology ranges, the rate of change thresholds, and redundant measurements in close proximity. |
| **timeSeries Variables** | Ventilated air temperature transmitter 2.1265.20.000 | Thies Clima | Temperature (K) | The sensor is protected by a double thermal radiation shield. A built-in ventilator provides for the necessary air flow. | 1 min | 0.1 K | Instrument is checked on a daily basis. Observations were checked against site-based climatology ranges, the rate of change thresholds, and redundant measurements in close proximity. |
| **timeSeries Variables** | HMP155 | Vaisala | Relative Humidity (1 or %) | The sensor with additional temperature sensor is installed in a vented radiation shelter. | 1 min | 0.80% | Instrument is checked on a daily basis. Observations were checked against site-based climatology ranges, the rate of change thresholds, and redundant measurements in close proximity. |
| **timeSeries Variables** | CMP22 | Kipp and Zonen | Upward Short-wave Radiation (W $m^{-2}$) | Sensor installed in an Eigenbrodt ventilation system to prevent from icing. | 1 min | 5 $Wm^{-2}$ | Instrument is checked on a daily basis. Data quality check is performed according to BSRN requirements. |

| | | | | | | | |
|---|---|---|---|---|---|---|---|
| **timeSeries Variables** | CMP22 | Kipp and Zonen | Downward Short-wave Radiation (W m$^{-2}$) | Sensor installed in an Eigenbrodt ventilation system to prevent from icing. | 1 min | 5 Wm$^{-2}$ | Instrument is checked on a daily basis. Data quality check is performed according to BSRN requirements. |
| **timeSeries Variables** | Precision Infared Radiometer | Eppley | Upward Long-wave Radiation (W m$^{-2}$) | Sensor installed in an Eigenbrodt ventilation system to prevent from icing. | 1 min | 5 Wm$^{-2}$ | Instrument is checked on a daily basis. Data quality check is performed according to BSRN requirements. |
| **timeSeries Variables** | Precision Infared Radiometer | Eppley | Downward Long-wave Radiation (W m$^{-2}$) | Sensor is shaded and installed in an Eigenbrodt ventilation system to prevent from icing. | 1 min | 5 W/m2 | Instrument is checked on a daily basis. Data quality check is performed according to BSRN requirements. |
| **timeSeriesProfile Variables** | CL51 | Vaisala | Cloud Base Height (m) | Proprietary algorithm determines the lowest cloud base height | 1 min | ~10 m | Operated with the standard Vaisala proprietary algorithm that retrieves cloud base height. Additional check for unphysical outliers. |
| **timeSeriesProfile Sonde Variables** | RS41 | Vaisala | Atmospheric pressure (Pa) | Standard radiosonde launch | 6 hr | 0.5 hPa | No additional QC beyond the standard Vaisala proprietary algorithm. |
| **timeSeriesProfile Sonde Variables** | RS41 | Vaisala | Eastward Wind (m s$^{-1}$) | Standard radiosonde launch | 6 hr | 0.15 ms$^{-1}$ | No additional QC beyond the standard Vaisala proprietary algorithm. |
| **timeSeriesProfile Sonde Variables** | RS41 | Vaisala | Northward Wind (m s$^{-1}$) | Standard radiosonde launch | 6 hr | 0.15 ms$^{-1}$ | No additional QC beyond the standard Vaisala proprietary algorithm. |
| **timeSeriesProfile Sonde Variables** | RS41 | Vaisala | Temperature (K) | Standard radiosonde launch | 6 hr | 0.3 K | No additional QC beyond the standard Vaisala proprietary algorithm. |
| **timeSeriesProfile Sonde Variables** | RS41 | Vaisala | Relative Humidity (1 or %) | Standard radiosonde launch | 6 hr | 4% | No additional QC beyond the standard Vaisala proprietary algorithm. |




**Table 10.** List of the instruments that contributed to the Eureka MODF, including details about the instrument manufacturer, measured variables, configuration, temporal resolution, measurement uncertainty, and quality control applied.

| MODF featureType | Instrument | Manufacturer | Measured variables | Instrument Configuration | Temporal Resolution | Uncertainty (+/-) | Quality Control |
|---|---|---|---|---|---|---|---|
| timeSeries Variables | PTB220 | Vaisala | Surface pressure (Pa) | Located on Flux Tower at 2 m height | 1 min | 0.3 hPa | Data are manually QC'ed to identify and eliminate instrument malfunction; outliers are filtered out if values are physically impossible; values are compared to other local variables if/when possible |
| timeSeries Variables | VENTUS-UMB Ultrasonic | Lufft | Near-surface (6m) eastward wind (m s$^{-1}$) | Located on Flux Tower at 6 m | 1-10 s | 0.1 ms$^{-1}$ | Data are manually QC'ed to identify and eliminate instrument malfunction; outliers are filtered out if values are physically impossible; values are compared to other local variables if/when possible |
| timeSeries Variables | VENTUS-UMB Ultrasonic | Lufft | Near-surface (6m) northward wind (m s$^{-1}$) | Located on Flux Tower at 6 m t | 1-10 s | 0.1 ms$^{-1}$ | Data are manually QC'ed to identify and eliminate instrument malfunction; outliers are filtered out if values are physically impossible; values are compared to other local variables if/when possible |
| timeSeries Variables | HMT-337 | Vaisala | Near-surface (2m) air temperature (K) | Located on Flux Tower at 2 m height | 1 min | 0.2 K | Data are manually QC'ed to identify and eliminate instrument malfunction; outliers are filtered out if values are physically impossible; values are compared to other local variables if/when possible |
| timeSeries Variables | HMT-337 | Vaisala | Near-surface (2m) relative humidity (%) | Located on Flux Tower at 2 m height | 1 min | 1.5 + 0.015 × reading | Data are manually QC'ed to identify and eliminate instrument malfunction; outliers are filtered out if values are physically impossible; values are compared to other local variables if/when possible |
| timeSeries Variables | SR50A | Campbell Scientific | Surface Snow Thickness | Located on Flux Tower at 2 m height | 1 min | 1 cm | Manually QC'ed to identify and eliminate instrument malfunction and remove non-physical values. |
| timeSeries Variables | IRTS-P | Apogee | Surface (skin) temperature (K) | Located on Flux Tower at 2 m height | 1 min | 0.2 K | Data are manually QC'ed to identify and eliminate instrument malfunction; outliers are filtered out if values are physically impossible; values are compared to other local variables if/when possible |
| timeSeries Variables | PSP | Eppley | Upward surface short-wave radiation (W m$^{-2}$) | Located on Flux Tower at 11 m height | 1 min | 2.0 W m$^{-2}$ | Processed through Long QCRad; Historical Quality Control Techniques: Long, C. N., & Shi, Y. (2008). An Automated Quality Assessment and Control Algorithm for Surface Radiation Measurements. OASJ, 2, 23- |

37. doi: 10.2174/1874282300802010023

Younkin, K., & Long, C. N. (2004). Improved Correction of IR Loss in Diffuse Shortwave Measurements: An ARM Value Added Product.

| | | | | | | | |
|---|---|---|---|---|---|---|---|
| timeSeries Variables | CM22 | Kipp and Zonen | Downward surface short-wave radiation (W m$^{-2}$) | Located on Flux Tower at 11 m height | 1 min | 5.0 W m$^{-2}$ | Processed through Long QCRad; Historical Quality Control Techniques: Long, C. N., & Shi, Y. (2008). An Automated Quality Assessment and Control Algorithm for Surface Radiation Measurements. OASJ, 2, 23-37. doi: 10.2174/1874282300802010023 Younkin, K., & Long, C. N. (2004). Improved Correction of IR Loss in Diffuse Shortwave Measurements: An ARM Value Added Product. |
| timeSeries Variables | CM22 | Kipp and Zonen | Upward surface long-wave radiation (W m$^{-2}$) | Located on Flux Tower at 11 m height | 1 min | 5.0 W m$^{-2}$ | Processed through Long QCRad; Historical Quality Control Techniques: Long, C. N., & Shi, Y. (2008). An Automated Quality Assessment and Control Algorithm for Surface Radiation Measurements. OASJ, 2, 23-37. doi: 10.2174/1874282300802010023 Younkin, K., & Long, C. N. (2004). Improved Correction of IR Loss in Diffuse Shortwave Measurements: An ARM Value Added Product. |
| timeSeries Variables | PIR | Eppley | Downward surface long-wave radiation (W m$^{-2}$) | Located on Flux Tower at 11 m height | 1 min | 4.0 W m$^{-2}$ | Processed through Long QCRad; Historical Quality Control Techniques: Long, C. N., & Shi, Y. (2008). An Automated Quality Assessment and Control Algorithm for Surface Radiation Measurements. OASJ, 2, 23-37. doi: 10.2174/1874282300802010023 Younkin, K., & Long, C. N. (2004). Improved Correction of IR Loss in Diffuse Shortwave Measurements: An ARM Value Added Product. |
| timeSeries Variables | HPFO1 | Hukseflux | Ground heat flux (W m$^{-2}$) | Depth 3 cm | 1 min | 3 % | Manually QC'ed to identify and eliminate instrument malfunction |
| timeSeriesProfile Variables | HMT-337 | Vaisala | Air temperature (K) | Located on Flux Tower at 2, 6, 10 m | 1 min | 0.2 K | Manually QC'ed to identify and eliminate instrument malfunction. The lowest level of temperature profile is also saved in the timeSeries file as surface temperature tas |
| timeSeriesProfile Variables | HMT-337 | Vaisala | Relative humidity (%) | Located on Flux Tower at 2, 6, 10 m | 1 min | 1.5 + 0.015 × reading | Manually QC'ed to identify and eliminate instrument malfunction. The lowest level of humidity profile is also saved in the timeSeries file as surface humidity hurs |

| | | | | | | | | |
|---|---|---|---|---|---|---|---|---|
| timeSeriesProfile Variables | TP-101 | MRC | Soil temperature profile (K) | Depth: 5cm [mV], 10cm [mV], 15cm [mV], 20cm [mV], 25cm [mV], 30cm [mV], 45cm [mV], 70cm [mV], 95cm [mV], 120cm [mV] | 1 min | n/a | Manually QC'ed to identify and eliminate instrument malfunction. |
| timeSeriesProfile Variables | VENTUS -UMB Ultrasonic | Lufft | Eastward wind component (m s$^{-1}$) | Located on Flux Tower at 6 m and 11 m | 1-10 s | 0.1 ms$^{-1}$ | Manually QC'ed to identify and eliminate instrument malfunction. The lowest level of wind speed profile is also saved in the timeSeries file as surface wind wpeed uas |
| timeSeriesProfile Variables | VENTUS -UMB Ultrasonic | Lufft | Northward wind component (m s$^{-1}$) | Located on Flux Tower at 6 m and 11 m | 1-10 s | 0.1 ms$^{-1}$ | Manually QC'ed to identify and eliminate instrument malfunction. The lowest level of wind speed profile is also saved in the timeSeries file as surface wind speed vas |






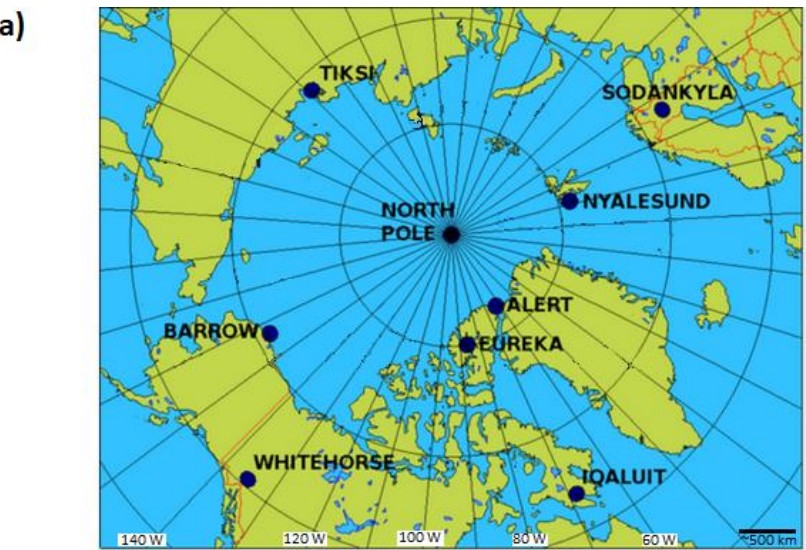

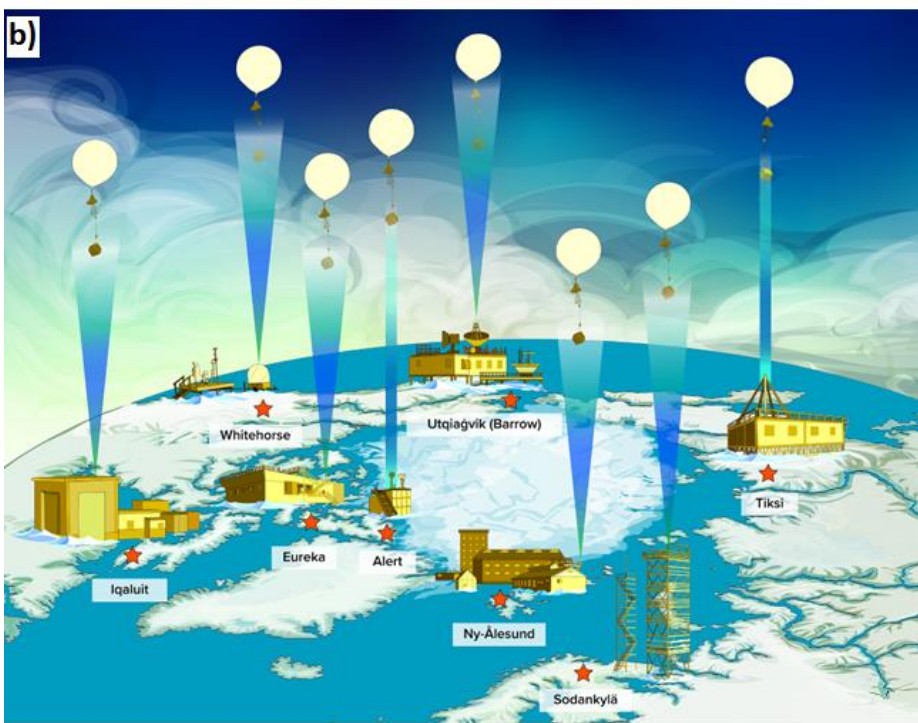


**Figure 1.** a) Locations of the MODF$_{ysm}$ YOPP supersites (Antarctic sites not shown). (b) Infographic depicting iconic building(s) at each site. The infographic is roughly centred around the North Pole (centre). All locations shown have generated a MODF$_{ysm}$, with the exception of Alert (in progress).



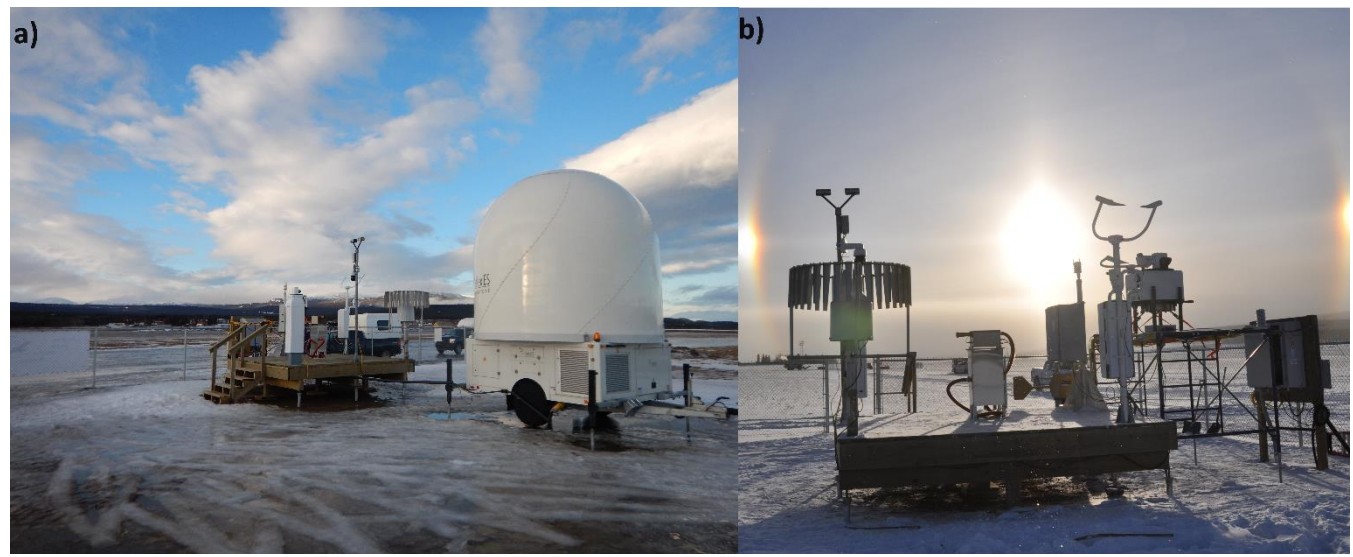


**Figure 2.** The Whitehorse site and the surrounding airfield in early spring 2018 with an X-band radar (white dome) in the foreground (a),
and the main instrument platform, including a Pluvio2, Parsivel, FS11P, WXT520, and CL51 ceilometer (from left to right) with a sundog
in the background (b). Photos adapted from Figure 5 in Mariani et al. (2022).

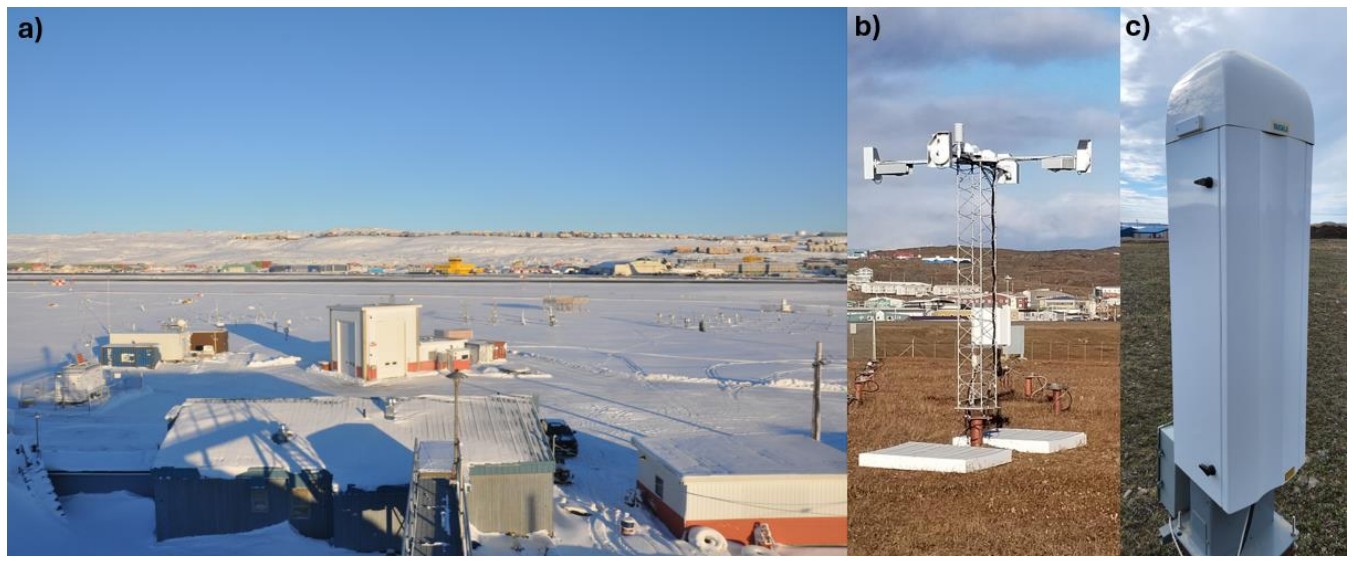


**Figure 3.** The Iqaluit site surroundings taken in winter 2018 with the Iqaluit airport in the background (a), the radiation flux sensor suite
during the summer, consisting of several CMP10Ls, CGR4Ls, and SR50As (b), and the CL51 ceilometer during the summer (c). Photos
adapted from Figure 2 (Mariani et al., 2022).

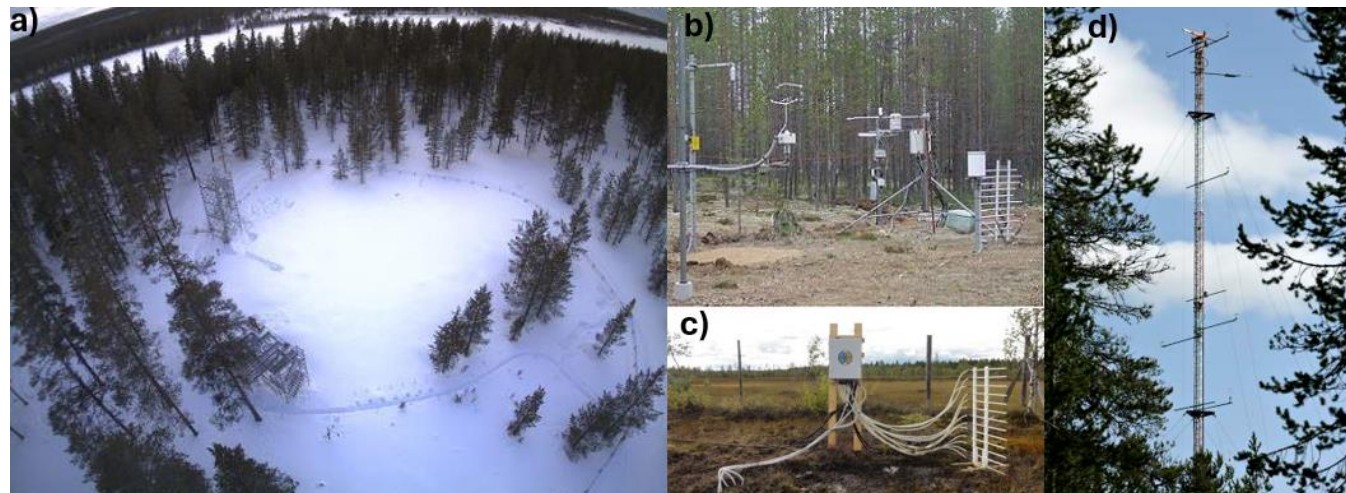

Figure 4. The Sodankylä site surroundings during the winter at the Intensive Observation Area, IOA, in the boreal forest (a), snow, soil and meteorological measurements in the MET measurement field (b), multi-level snow and soil measurements at the Peatland site, SUO, (c) and the meteorological tower with meteorological and radiation sensors (d). Photos: FMI (litdb.fmi.fi).

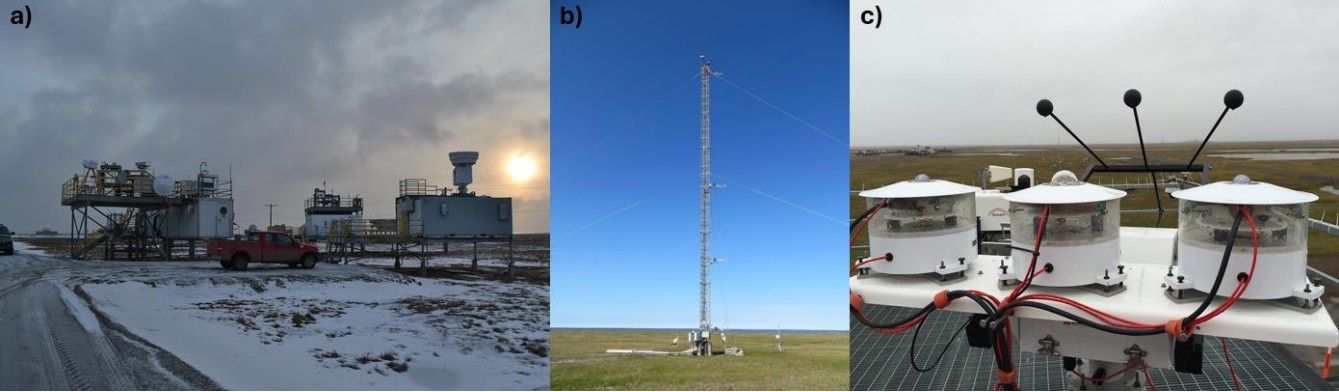

Figure 5. The Utqiaġvik site surroundings during the winter, including the main observation stations and their rooftop instrument suites (a), the meteorological tower with radiation flux sensors deployed in the summer (b), and the SKYRAD downward longwave radiation sensor deployed on the roof in the spring (c). Photos: www.arm.gov.

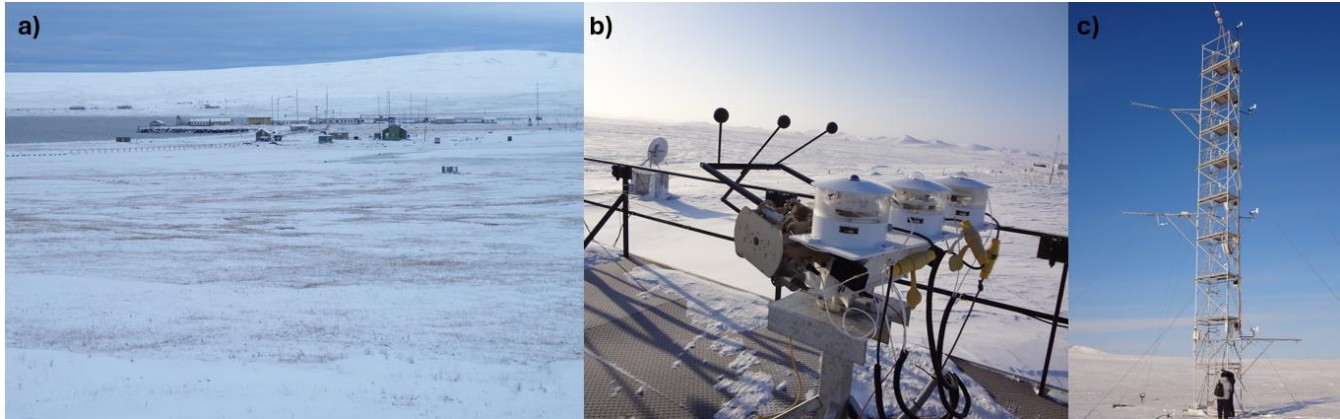

**Figure 6.** The Tiksi site surroundings, taken from afar in the winter (a), the SKYRAD downward longwave radiation sensor deployed on the roof of the Tiksi observation building (b), and the meteorological tower equipped with radiation flux sensors (c). Photos: Taneil Uttal (NOAA).

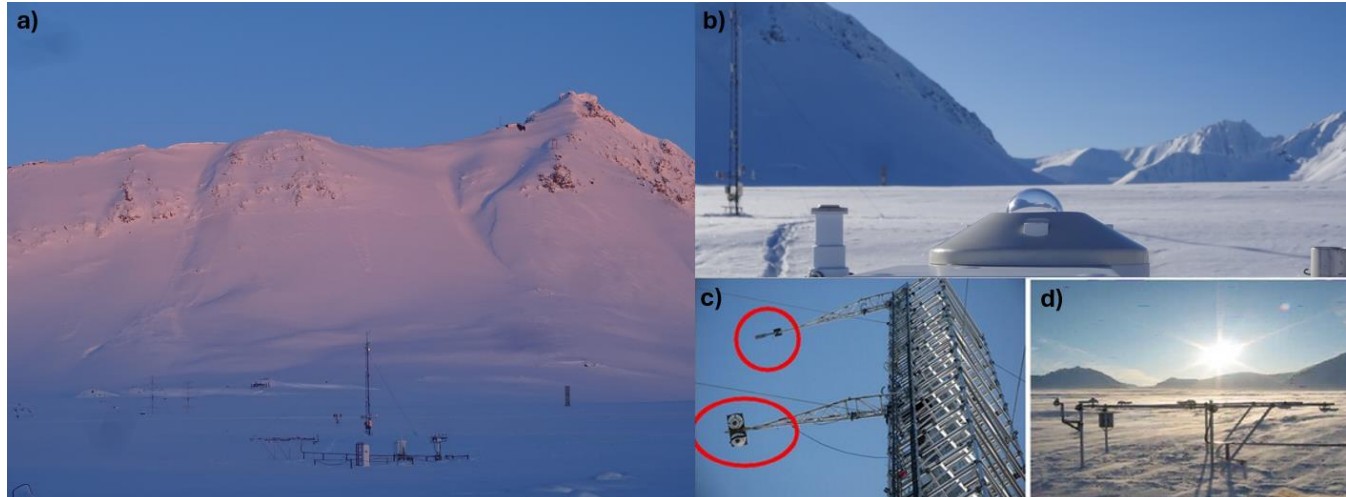

**Figure 7.** The Ny-Ålesund site surroundings taken in the winter with the meteorological sensors and radiation tower in the foreground (a), the CMP22 downward shortwave radiation sensor at the site (b), the meteorological tower with the radiation flux sensors circled (c), and several surface meteorological and albedo-measuring sensors at the BSRN station (d). Photos (c-d) are adapted from Figure 1 in Becherini et al., 2021.

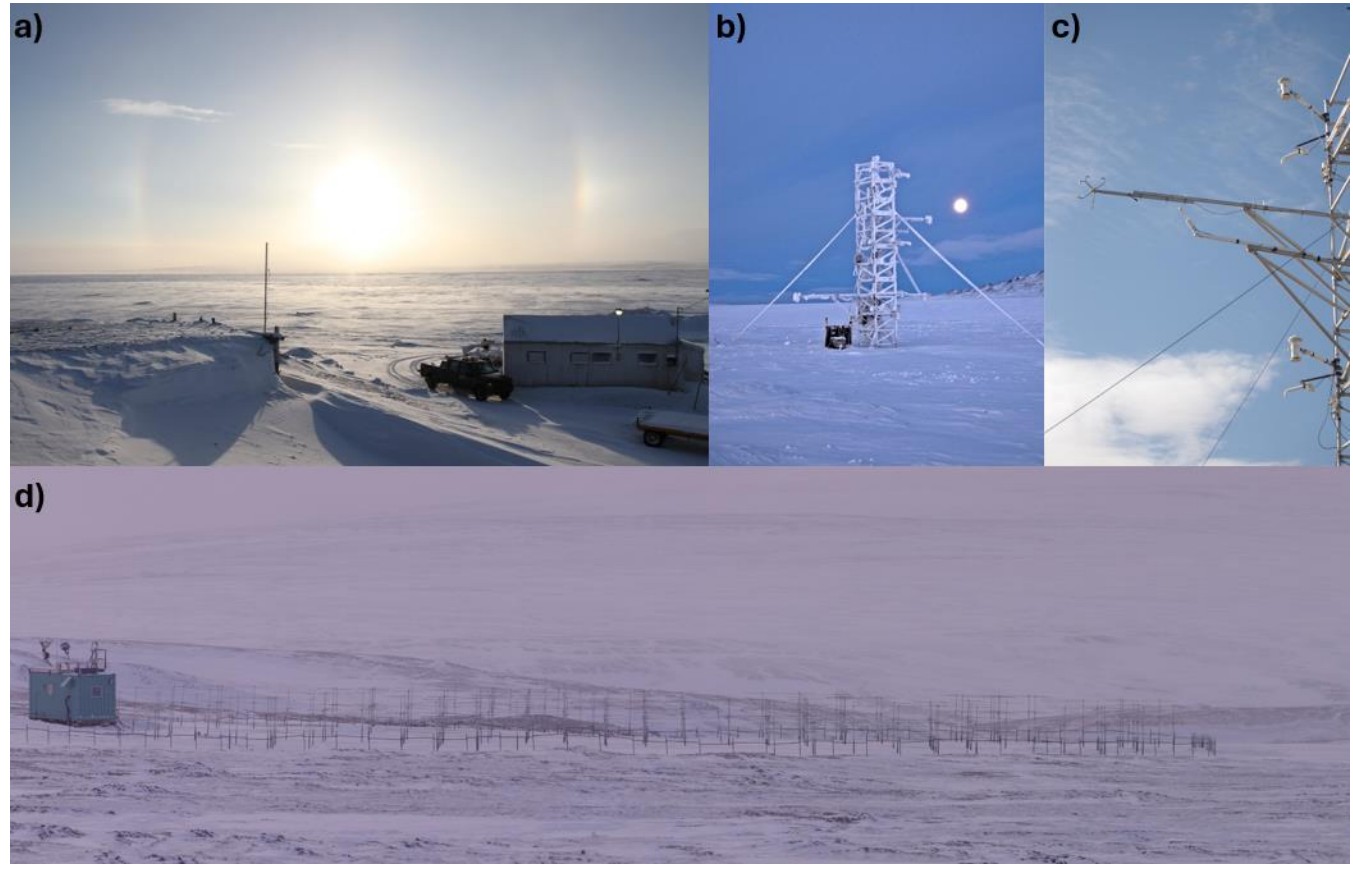


**Figure 8.** The Eureka site surroundings in the winter, facing south from the Eureka Weather Station (EWS) looking over the frozen fjord
with a sundog in the background (a), the meteorological tower at the Surface and Atmospheric Flux Irradiance Extension (SAFIRE)
(b) with radiation flux (e.g., PSP) and meteorological sensors deployed (c), and the SAFIRE site surroundings taken from afar (d).


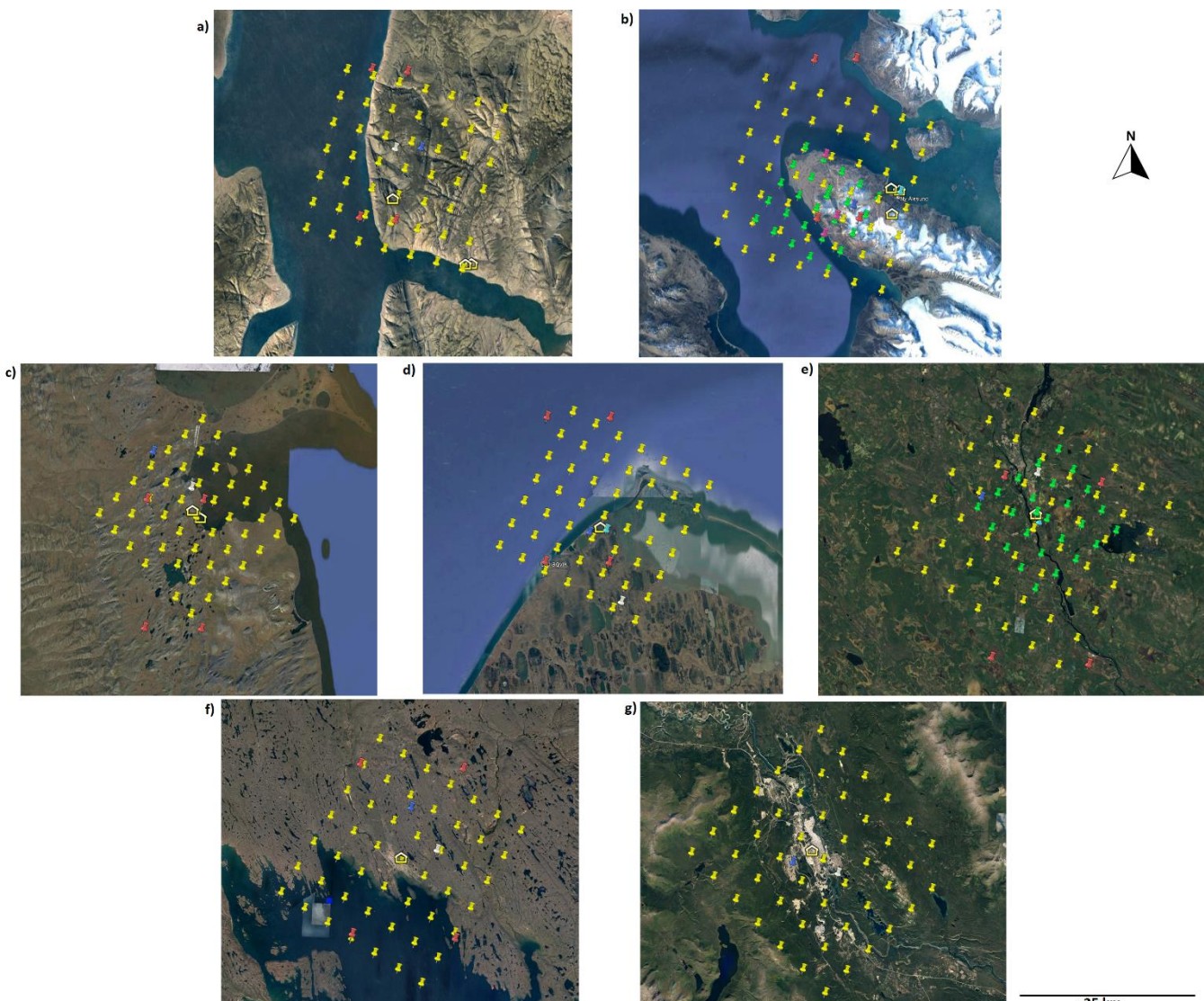

**Figure 9.** Model grid points at and around each site (a) Eureka, (b) Ny-Ålesund, (c) Tiksi, (d) Utqiaġvik, (e) Sodankylä, (f) Iqaluit, and (g) Whitehorse, displayed through the Google Earth web-platform: *Image Landsat / Copernicus, Image ©2023 Maxar Technologies*. Sites are organized from highest latitude (Eureka) to lowest (Whitehorse). Yellow building icons represent the location of the facility on-site which contains all co-located instruments. Similarly, icons for the AROME-Arctic model grid are indicated by a green pin, ARPEGE pins are in white, DWD-ICON pins are light blue, ECCC-CAPS pins are yellow, ECMWF-IFS pins are dark blue, and SL-AV pins are in red. All images are north-aligned, nadir view.


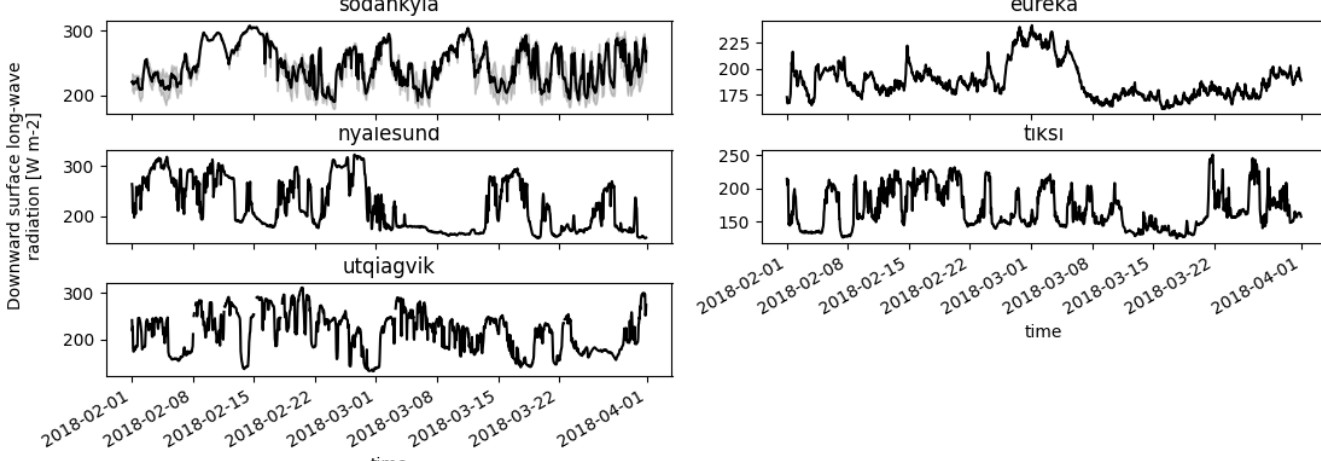


**Figure 10.** Observations (30-min) of downward surface long-wave radiation ("rlds") conducted during SOP1 at each site. Observations from
Whitehorse and Iqaluit were not available during SOP1. Sodankylä conducts multiple observations of rlds; the mean (black line) and min/max
spread in observed rlds (grey shaded area) are shown.



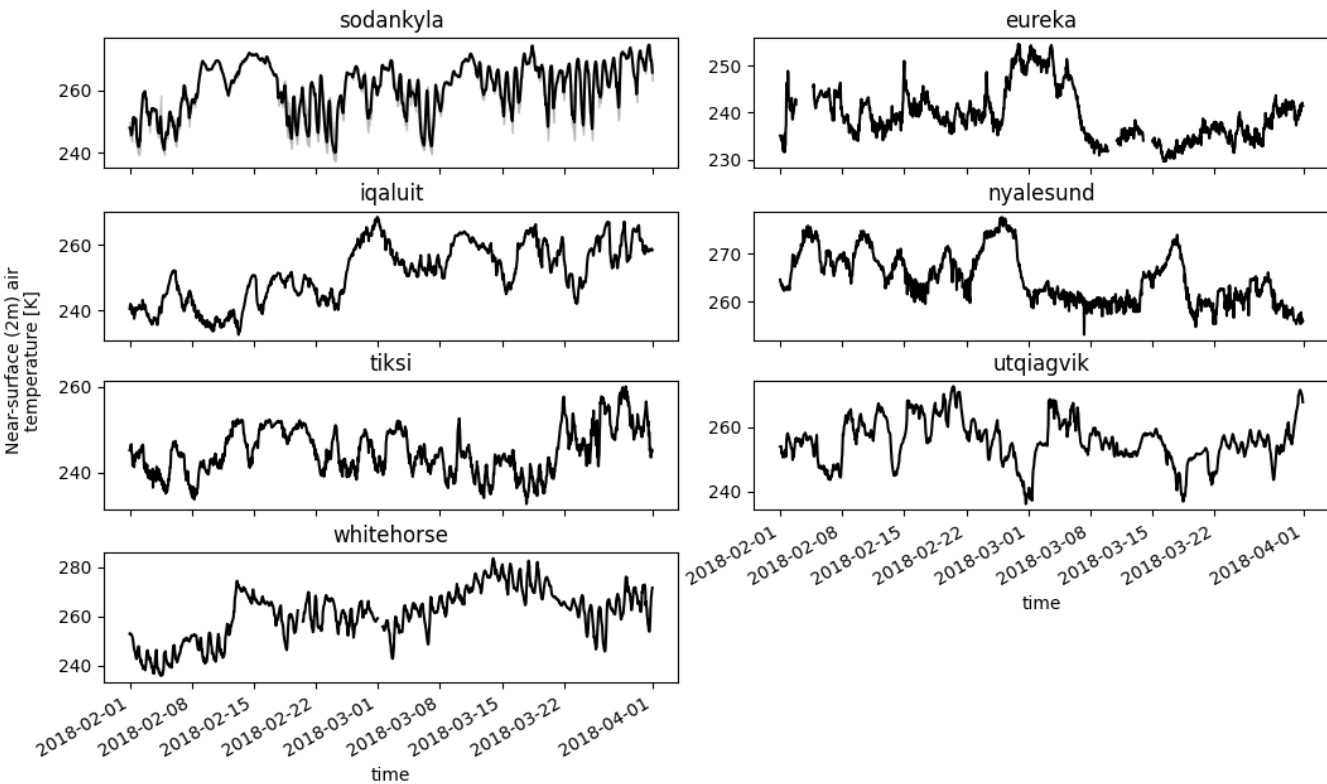


**Figure 11.** Similar to Figure 3, except for observations of near-surface (2 m) air temperature ("tas") conducted at each site during SOP1.


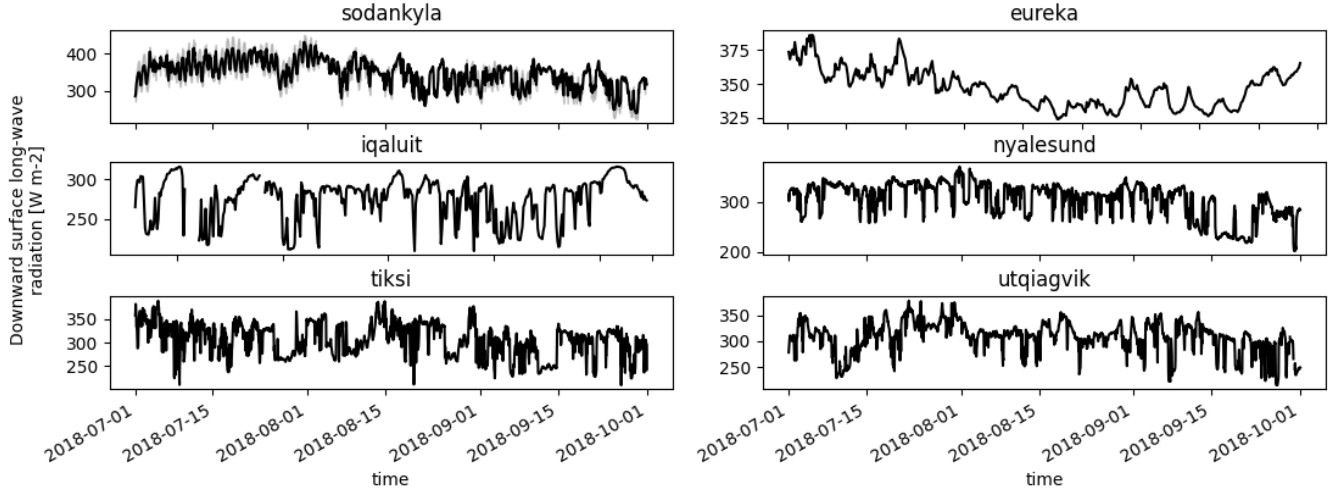


**Figure 12.** Similar to Figure 3, except for observations of downward surface long-wave radiation ("rlds") conducted during SOP2 at each site. Observations from Whitehorse were not available during SOP2.



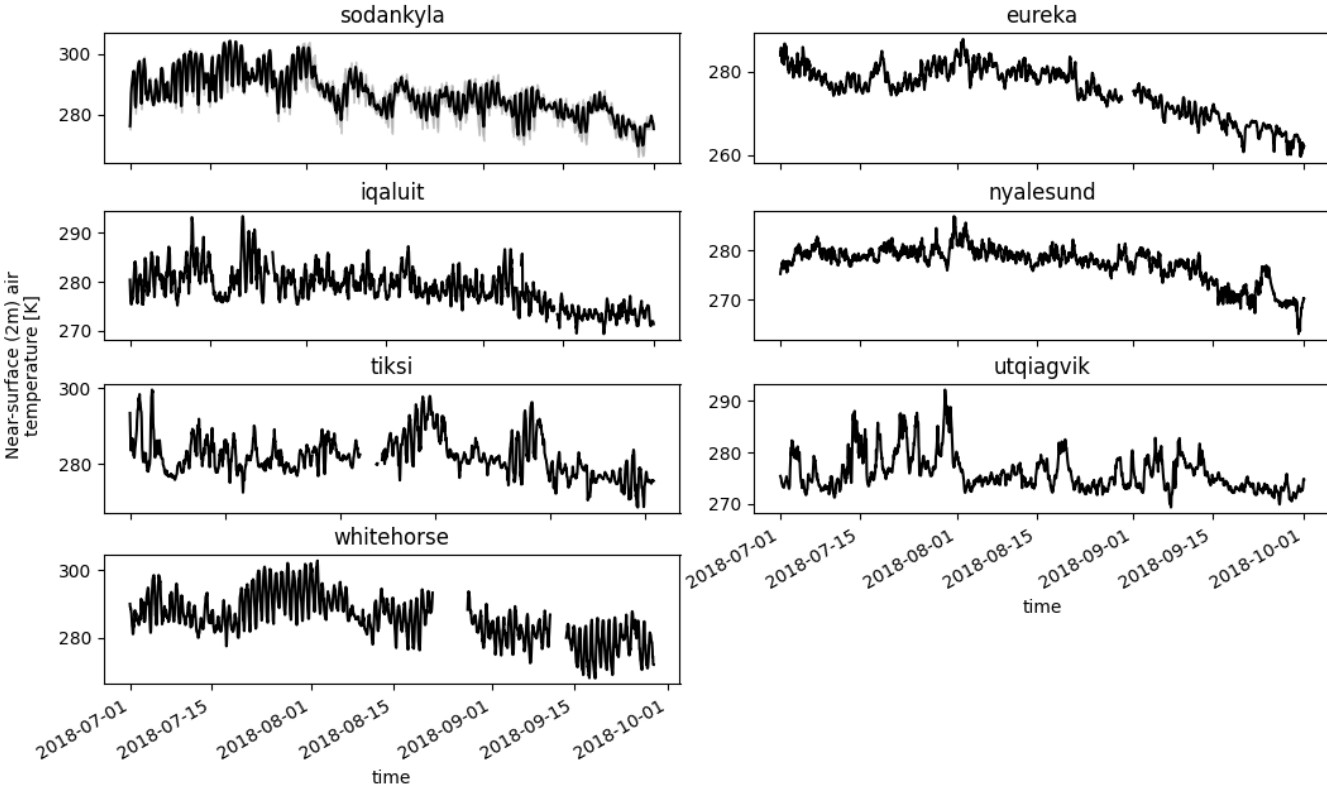


**Figure 13.** Similar to Figure 3, except for observations of near-surface (2 m) air temperature ("tas") conducted at each site during SOP2.