# Peer review of "Special Observing Period (SOP) Data for the Year of Polar Prediction site Model Intercomparison Project (YOPPsiteMIP)"

_Earth System Science Data, 2023_

## Referee Comment (RC1)

Review comments

Manuscript information:
Title: Special Observing Period (SOP) Data for the Year of Polar Prediction site Model Intercomparison Project (YOPPsiteMIP)
Author(s): Zen Mariani, Sara Morris, Taneil Uttal, Elena Akish, Robert Crawford, Laura Huang, Jonathan Day, Johanna Tjernström, Øystein Godøy, Lara Ferrighi, Leslie Hartten, Jareth Holt, Christopher Cox, Ewan O'Connor, Roberta Pirazzini, Marion Maturilli, Giri Prakash, James Mather, Kimberly Strong, Pierre Fogal, Vasily Kustov, Gunilla Svensson, Michael Gallagher, and Brian Vasel
MS No.: essd-2023-497
MS type: Data description paper

**General Comment:**

This paper provides meteorological observation data related to the YOPP project at seven sites in high-latitude regions. The data are being provided as a proposal for a unified data format and usage. Under the recent global-scale climate change, continuous meteorological observation data at high latitudes, where the environment is severe and local anthropogenic influences are relatively less likely, is of great scientific value. Therefore, this paper and its accompanying data set are worthy of publication and being open.

However, in publishing this paper, some of the information necessary to ensure data availability is missing. This point cannot be ignored when considering the paper's position as a data description paper. In order to publish this paper, it is necessary to add such information and revise the manuscript so that readers or data users can understand the usability of the data appropriately.

The following Specific Comments (Major and Minor Comments) are mainly suggestions for related revisions, but these are not individual comments but requests for consideration of revisions throughout the paper. Based on these comments, please revise the paper to make the data more accessible to readers and users of the data. Other easy revisions are listed as Technical Corrections; please consider revising these as well.

**Specific Comments:**
[Major Comments]
1. Information on the environment in which the observation equipment is installed should be added.

Photographs of the observation equipment and its location should be included to show the conditions at the observation site. The installation and surrounding environments are important factors

in meteorological observation data. Depending on the environment, the interpretation of the data may change. In this manuscript, each meteorological observation site is described a little in the text, but that is not enough. Please include photographs of the observation equipment in the text, Appendix, or Supplement.

In addition, Fig. 2 should be replaced with a clearer photo or image for the same reason. The photo in Fig. 2 hints at the cover information around the installation site but does not provide other important information, such as the specific surface cover type or topographic information. The reader needs to interpret the meteorological data, so the figure should be changed to include such information.

**2. Please describe the QC, even briefly.**

Related to Major Comment 1, without a description of the environment in which the data was processed, data users cannot determine whether they can fully rely on this dataset. In an appropriate observing environment, a simple QC can make the dataset more complete, but in a complex or harsh observing environment, the reliability of the dataset will depend on how carefully the QC was performed. Although there is a brief description of the QC method in this manuscript, it should be explained more carefully in this paper (rather than just citing the method) due to the aim of data description papers. Since it is difficult to judge the usability of the overall dataset at present, authors should describe how the QC was performed, even briefly, along with a description of the surrounding environment (also related to Major Comment 1).

**3. Lack of information on observation sensors and data processing**

Whether or not information on the type of meteorological sensor is specified is an important aspect in interpreting the validity of the observation data, along with information on the observation environment. Depending on the characteristics of the sensor, the observed data may not faithfully reflect the surrounding meteorological conditions. The lack of easy access to the list of observation items and sensors used at different sites is a major obstacle to the main goal of organizing observation data at the super sites in a unified manner. Although information is provided in fragments in the text, it would be easier for readers to read if the information were systematically organized in tables.

In addition, there is a lack of information on the sensor installation environment. For example, whether the air temperature sensor is installed in a forced or natural ventilation shelter, whether the shelter heating effect is corrected for natural ventilation, whether the humidity is corrected in a sub-freezing environment, whether the flux measured by the eddy correlation method eliminates the influence of the surrounding environment, etc. It is important to specify whether or not the data set has been corrected for the factors mentioned above in addition to QC to ensure data availability.

[Minor Comments]

- L55 Polar prediction

What prediction?

- L128 super site

Is "super site" defined somewhere? If not, mentioning "super site" every time is redundant, so it should be rewritten as just "site".

- L128 the Canadian Arctic Weather Science (CAWS) project

Please add a brief description of the purpose of the CAWS project.

- L135, 149 etc. roughness length

What is the intention of including roughness length? Roughness varies greatly depending on the surface condition (snow cover, vegetation, bare ground, etc.), so simply showing the average roughness value has less meaning. If there is an intention, please modify it so that the intention is clear; otherwise, consider deleting it as it is redundant.

- L139 average monthly precipitation

It should be stated in terms of annual total precipitation rather than monthly average precipitation. Temperature and precipitation amounts as climatic values are important and valuable data, and their inclusion is strongly recommended. However, they are generally presented as annual mean temperature and total precipitation as climate values. It is acceptable to include monthly averages, but it would be better to include them as additional information to the annual precipitation totals.

- L251 snowfall of 60.3 cm

Would this "snowfall of 60.3 cm" be 46.6 mm w.e. in water equivalent? It should be stated in terms of water equivalent to describe precipitation as a climatic value. That information can be listed if the authors want to list the snow depth.

- L296 The radiation flux observations were processed using the eddy correlation and bulk method

This statement is inappropriate since the eddy correlation and bulk methods are related to heat fluxes due to atmospheric turbulence and have nothing to do with radiation. I did not fully understand the author's intention to refer to the heat balance method, but please correct this part.

- L374 (such as snow/frost deposition...)

The content of the parentheses is too long and instead impairs readability. I request that the text be revised.

- L403 assessed by the site scientist/data quality office

I do not understand what is intended by this expression. Please revise the wording specifically to describe it more.

- Table1

It is difficult to understand what "Measured Variable" means, starting with "All" in "Measured Variable" at Whitehouse and Iqaluit, "Total precipitation of water" at other sites, "all wind", "all radiation", "all wind," and "all radiation" and so on, at other locations. Please consider correcting the element names, including correcting the terminology pointed out in the Minor Comment to L280.

- Table3

(1) Please cite the H-K table or add a brief explanation in the caption. The explanations of figures and tables should be in a style that can be understood to some extent without reading the paper by looking at the figures and captions.

(2) Please correct the superscripts of the units in "Measured variable" as they are not superscripted.

(3) Please explain what you mean by "EC" and "bulk" in the Table for the method of Flux measurement.

- Figure1

Please increase the resolution of the figures in both (a) and (b). Also, it cannot be understood what the 1000 km scale in (b) means. In the notation of figure (b), the distance in the figure should not be constant. The scale should not be notated on such a diagram. Please delete it.

- Figure3–6

Please include the time step of the data in the caption. Daily? Hourly?

- Figure3

I do not understand what the mean and spread indicate. Please add an explanation.

**Technical Corrections:**

-L55 Jung et al., 2014

I could not find this reference in the reference section.

- L148, L217, L281, L292 "/s"

-> "s$^{-1}$"

- L198 NOAA

Since "NOAA" has already appeared in L178, there is no need to write the official name in parentheses.

- L237 The CAnadian Network...

I understand the intention of capitalizing the first letter of the abbreviation, but it should be "Canadian..." as the official name.

- L243, L253, L264, L309, L384, L392, L407, L412, L413, L417 "&"

-> "and"

- L280, L292 wind

Does it mean "wind speed and direction"?

- L280, L292 pressure

Does it mean "atmospheric pressure"?

- L301 "/m$^2$"

-> m$^{-2}$

- L301 "μV/W/m$^2$"

-> μV/(W/m$^{-2}$)$^{-1}$

- L327 CMPI6

CMIP6? Please confirm and correct if necessary.

-L357 Huang et al., 2023b; 2023a

-> Huang et al., 2023a; 2023b

- L367 missing_value

-> "missing_value"

I understood that "missing_value" is a flag indicating a missing value. I think it is better to enclose it with " in the text to distinguish it from a flag with a proper noun role and an ordinary word.

- L394 Cook et al. (2008) と Fuehrer and Friehe (2002)

Cook et al. (2008) and Fuehrer and Friehe (2002) were not on the list of cited references. Please add them.

- Reference

Formatting is not standardized. For example, the following points. Please again check the submission policies and correct them to the prescribed format.

(1) The format of "doi" is not unified.

(2) The word [dataset] is not written in the dataset's reference.

(3) The following points should be corrected as required.

- L413 Younkin & Long, 2004

-> Is 2003 wrong? Please check.

- L413 Maturilli, 2020a, 2020b, 2020c, 2022

-> Maturilli (2020a, 2020b, 2020c, 2022)

- All Tables

(1) The caption of the Table should be written at the top of the Table. Please modify it.

(2) The Tables do not include vertical lines. Please remove the vertical bars and modify them to a refined appearance.

- Table2 "&"

-> "and"

---

## Author Comment (AC1)

**Reviewer 1**

Thank you for your work in helping us improve the manuscript. We have made changes to the manuscript for your consideration. We have responded to all R1 comments below and outlined changes in the manuscript (via tracked changes).

- Information on the environment in which the observation equipment is installed should be added. Photographs of the observation equipment and its location should be included to show the conditions at the observation site. The installation and surrounding environments are important factors in meteorological observation data. Depending on the environment, the interpretation of the data may change. In this manuscript, each meteorological observation site is described a little in the text, but that is not enough. Please include photographs of the observation equipment in the text, Appendix, or Supplement.
  - The description of each site has been significantly increased. In particular, seven new detailed Tables (Tables 4-10) outlining all of the instruments, their manufacturers, accuracy, operating configuration, temporal resolution, and quality control at each site is now included. Note these Tables are extensive (several pages long) so we would also suggest having them in an Appendix instead. We have noted in the manuscript that the information in these tables is also documented in the attributes of the MODFs themselves. Photographs of each site, including the surrounding topography, are now included as well (Figures 2-8). Additional photographs of several key instruments at each site, as an example of the operating conditions, are also included. Combined, this enables the user to better understand the surrounding environment, equipment, relative locations of instruments, and surface conditions.
- Fig. 2 should be replaced with a clearer photo or image for the same reason. The photo in Fig. 2 hints at the cover information around the installation site but does not provide other important information, such as the specific surface cover type or topographic information. The reader needs to interpret the meteorological data, so the figure should be changed to include such information.
  - Seven new Figures (Figures 2-8 in the new manuscript) have been added to the manuscript to address this. The new figures clearly illustrate each site's topography and cover information at and around the site, showing the specific surface cover type and other topographic information. Note these new Figures are numerous (several sub-figures were requested to show instruments as well as the surrounding area), so we suggest having them in an Appendix instead.
  - The original Figure 2 (now Figure 9) has been mostly left as-is to provide a "zoomed out" perspective of the synoptic region, km-scale land cover, and the NWP model grid points that fall within this region; it also appears later in the paper when NWP models are discussed.
- Please describe the QC, even briefly. Related to Major Comment 1, without a description of the environment in which the data was processed, data users cannot determine whether they can fully rely on this dataset. In an appropriate observing environment, a simple QC can make the dataset more complete, but in a complex or harsh observing environment, the reliability of the dataset will depend on how carefully the QC was performed. Although there is a brief description of the QC method in this manuscript, it should be explained more carefully in this paper (rather than just citing the method) due

to the aim of data description papers. Since it is difficult to judge the usability of the overall dataset at present, authors should describe how the QC was performed, even briefly, along with a description of the surrounding environment (also related to Major Comment 1).

- o The discussion about QC has been significantly increased and expanded in the paper. The new Tables 4-10 are extensive and provide a dedicated quality control column including a full description of how QC was applied for every single instrument at each site (i.e., all measured variables). The QC method is no loner just cited but now explained in detail with references provided. There also exists an in-depth discussion of QC in Section 4 to highlight processing and QC differences between some of the sites. Details pertaining to how the QC was performed, thresholds and metrics used, etc., is now included in the new Tables 4-10. A description and pictures of the surrounding environment is now provided in seven new Figures 2-8 (see previous two comments) to improve context of the observations and their conditions.
- Lack of information on observation sensors and data processing. Whether or not information on the type of meteorological sensor is specified is an important aspect in interpreting the validity of the observation data, along with information on the observation environment. Depending on the characteristics of the sensor, the observed data may not faithfully reflect the surrounding meteorological conditions. The lack of easy access to the list of observation items and sensors used at different sites is a major obstacle to the main goal of organizing observation data at the super sites in a unified manner. Although information is provided in fragments in the text, it would be easier for readers to read if the information were systematically organized in tables.
  - o Seven new Tables (4-10) have been added to the paper which describes each individual instrument that was used in the MODF for each site. Details about each site's instruments, including the make, model, accuracy, uncertainty, processing technique, and QC are now provided in full detail. Where possible, references to the manufacturer's datasheets and other reference material is provided in the text, including additional comments on methodologies and relevant intricacies of each instrument's dataset. This long list of instruments at each site is quite extensive and detailed; as such it may also be suitable as an Appendix. Overall, these new Tables improve the organization and clarity regarding the source of each observation in a clear and standardized manner.
- There is a lack of information on the sensor installation environment. For example, whether the air temperature sensor is installed in a forced or natural ventilation shelter, whether the shelter heating effect is corrected for natural ventilation, whether the humidity is corrected in a sub-freezing environment, whether the flux measured by the eddy correlation method eliminates the influence of the surrounding environment, etc. It is important to specify whether or not the data set has been corrected for the factors mentioned above in addition to QC to ensure data availability.
  - o The new Tables (4-10) added to the manuscript now provide this information (see previous comment). Details regarding the setup, environment, and configuration for each instrument is now provided in full detail to clearly indicate the precise observing configuration as well as the level of QC that was / was not performed. The new Figures 2-8 have photographs of several of these instruments (as an example) and the surrounding area; they provide visual context of their operating configuration and surrounding environment.
- L55 Polar prediction: What prediction?
  - o The text now reads "Polar weather forecast prediction" to clarify this.

- L128 super site: Is "super site" defined somewhere? If not, mentioning "super site" every time is redundant, so it should be rewritten as just "site".
  - L71 now provides a brief description to distinguish 'supersite' from 'site': "in general, the suite of several additional instruments that enable an enhanced measurement program, including remote sensing, radiation, and other meteorological sensors, is what distinguishes a 'supersite' from a typical weather site."
  - The text now reads 'supersite (hereafter referred to as "sites")' in the Introduction immediately following this sentence, with "sites" being used for the remainder of the text to reduce redundancy.
- L128 the Canadian Arctic Weather Science (CAWS) project: please add a brief description of the purpose of the CAWS project.
  - CAWS was initiated to evaluate upper air observing technologies that can complement and improve Polar forecasts, perform satellite calibration / validation over Arctic terrain, and to provide recommendations to optimize the Canadian Arctic observing network. This text is now included in the manuscript.
- L135, 149 etc. roughness length: what is the intention of including roughness length? Roughness varies greatly depending on the surface condition (snow cover, vegetation, bare ground, etc.), so simply showing the average roughness value has less meaning. If there is an intention, please modify it so that the intention is clear; otherwise, consider deleting it as it is redundant.
  - This has been removed from the text to reduce redundancy.
- L139 average monthly precipitation: it should be stated in terms of annual total precipitation rather than monthly average precipitation. Temperature and precipitation amounts as climatic values are important and valuable data, and their inclusion is strongly recommended. However, they are generally presented as annual mean temperature and total precipitation as climate values. It is acceptable to include monthly averages, but it would be better to include them as additional information to the annual precipitation totals.
  - The total annual precipitation is now provided for each site, as well as the annual average temperature. This information is now reported for all sites in a systematic manner, following R2's comment on the structure of the site descriptions.
- L251 snowfall of 60.3 cm: would this "snowfall of 60.3 cm" be 46.6 mm w.e. in water equivalent? It should be stated in terms of water equivalent to describe precipitation as a climatic value. That information can be listed if the authors want to list the snow depth.
  - In order to standardize the site descriptions (standard statements for precipitation, temperature, etc.) and improve clarity, the reported snowfall amount for Eureka has been removed from the text, in response to R2's comments.
- L296 The radiation flux observations were processed using the eddy correlation and bulk method: this statement is inappropriate since the eddy correlation and bulk methods are related to heat fluxes due to atmospheric turbulence and have nothing to do with radiation. I did not fully understand the author's intention to refer to the heat balance method, but please correct this part.
  - The text has been changed to "heat flux;" the use of "radiation flux" with respect to using EC and bulk was an error, thank you for catching this error.
- L374 (such as snow/frost deposition…): the content of the parentheses is too long and instead impairs readability. I request that the text be revised.

- o   The parentheses have been moved to a new sentence to improve clarity: "These sources of error may include snow/frost deposition on radiation and temperature sensors or absorption of solar radiation by unsheltered temperature sensors."
- L403 assessed by the site scientist/data quality office: I do not understand what is intended by this expression. Please revise the wording specifically to describe it more.
  - o   The sentence has been changed to "A second level of manual QC was performed whereby data was reviewed by instrument mentors and visually assessed by the site scientist/data quality office" to improve clarity. This assessment is conducted manually via visual inspection by the instrument / site PI as a means to verify the QC process.
- Table1: it is difficult to understand what "Measured Variable" means, starting with "All" in "Measured Variable" at Whitehouse and Iqaluit, "Total precipitation of water" at other sites, "all wind", "all radiation", "all wind," and "all radiation" and so on, at other locations. Please consider correcting the element names, including correcting the terminology pointed out in the Minor Comment to L280.
  - o   "All" refers to the entire list of the measured variables in Table 3, whereas "All radiation" refers to all radiation-relevant measured variables (for instance, all upward/downward longwave/shortwave radiation observations). This description is now included in the Table caption to clarify what 'all' refers to and which particular subsets (e.g., "all wind" refers to all the wind speed and direction observations) are being referred to. Using the short form "all" is required to significantly shorten the length of the table, reduce redundancy, and improve readability. The terminology regarding wind (minor comment to L280) has been resolved; the text now reads "wind speed and direction" here and elsewhere throughout the manuscript.
- Table3: (1) Please cite the H-K table or add a brief explanation in the caption. The explanations of figures and tables should be in a style that can be understood to some extent without reading the paper by looking at the figures and captions. (2) Please correct the superscripts of the units in "Measured variable" as they are not superscripted. (3) Please explain what you mean by "EC" and "bulk" in the Table for the method of Flux measurement.
  - o   (1) The reference for the H-K table is now provided in the caption.
  - o   (2) Missing superscripts of the units in "measured variable" are now fixed, thank you for catching this.
  - o   (3) "EC" and "bulk" are now explained in the Table caption.
- Figure 1: please increase the resolution of the figures in both (a) and (b). Also, it cannot be understood what the 1000 km scale in (b) means. In the notation of figure (b), the distance in the figure should not be constant. The scale should not be notated on such a diagram. Please delete it.
  - o   The resolution for Figure 1 has been increased to its maximum. The length scale in (b) has been removed.
- Figures 3-6: please include the time step of the data in the caption. Daily? Hourly?
  - o   The time step of the data used for these Figures is 30 minutes. This is now stated in their Figure caption.
- Figure 3: I do not understand what the mean and spread indicate. Please add an explanation.
  - o   The Sodankylä site conducts more than one measurement of the downward surface long-wave radiation at several locations (see the site description in Sect 2). Instead of plotting just one of these timeseries, this plot provides their mean (black line) and their spread (min – max) as the

grey shaded area. The Figure caption's text has been clarified to indicate that the spread is the min to max value of these observations.

- L55 Jung et al., 2014: I could not find this reference in the reference section.
  - The text should read "2016" – it has been fixed in the manuscript. Thank you for catching this error.
- L148, L217, L281, L292 "/s" -> "s–1"
  - Resolved throughout the manuscript as suggested.
- L198 NOAA: since "NOAA" has already appeared in L178, there is no need to write the official name in parentheses.
  - The official name no longer appears here and instead the acronym is defined at L178 during the first instance of NOAA.
- L237 The CAnadian Network…: I understand the intention of capitalizing the first letter of the abbreviation, but it should be "Canadian..." as the official name.
  - Changed to "Canadian" as suggested.
- L243, L253, L264, L309, L384, L392, L407, L412, L413, L417 "& &" -> "and"
  - All instances of "&" have been replaced with "and"
- L280, L292 wind: does it mean "wind speed and direction"?
  - The text has been clarified to "wind speed and direction."
- L280, L292 pressure: does it mean "atmospheric pressure"?
  - Yes; this is now clarified in the manuscript.
- L301 "/m2" -> m–2
  - Resolved as suggested
- L301 "μV/W/m2" -> μV/(W/m–2)–1
  - Resolved as suggested
- L327 CMPI6: CMIP6? Please confirm and correct if necessary.
  - The text has been changed to CMIP6. Thank you for catching this typo.
- L357 Huang et al., 2023b; 2023a -> Huang et al., 2023a; 2023b
  - Resolved as suggested.
- L367 missing_value -> "missing_value": I understood that "missing_value" is a flag indicating a missing value. I think it is better to enclose it with " in the text to distinguish it from a flag with a proper noun role and an ordinary word.
  - "missing_value" is now enclosed with quotes in the text, as suggested.
- L394 Cook et al. (2008) と Fuehrer and Friehe (2002): Cook et al. (2008) and Fuehrer and Friehe (2002) were not on the list of cited references. Please add them.
  - These two missing references have been added to the reference list, thank you for catching this oversight.
- Reference formatting is not standardized. For example, the following points. Please again check the submission policies and correct them to the prescribed format. (1) The format of "doi" is not unified. (2) The word [dataset] is not written in the dataset's reference. (3) The following points should be corrected as required: L413 Younkin & Long, 2004 -> Is 2003 wrong? Please check; L413 Maturilli, 2020a, 2020b, 2020c, 2022 -> Maturilli (2020a, 2020b, 2020c, 2022).

- o The reference formatting has been manually checked and modified to adhere to the journal's standards.
    (1) Doi format is now standardized, where possible, as much as possible.
    (2) The word dataset is now written in the dataset's reference,
    (3) 2003 is correct for Younkin; this typo is now fixed. The Maturilli reference is now fixed with parentheses, as suggested.
- All Tables: (1) The caption of the Table should be written at the top of the Table. Please modify it. (2) The Tables do not include vertical lines. Please remove the vertical bars and modify them to a refined appearance.
    - o (1) Table captions have been moved to the top of the Table.
    (2) We have removed the vertical lines in all the Tables. Further formatting will be achieved during the typesetting process, particularly for the new and lengthy Tables.
- Table2 -> "and"
    - o All instances of "&" have now been replaced with "and."

Thank you for your work in helping us improve the manuscript. We have made changes to the manuscript for your consideration. We have responded to all R2 comments below and outlined changes in the manuscript (via tracked changes).

- The abstract contains long sentences (ex: Lines 30-33; line 37-42) and the aim of improving numerical weather prediction (NWP) is also mentioned twice. The abstract should be improved for clarity and to summarize the material presented in the manuscript.
  - The abstract has been revised to improve the flow and clarity overall. The abstract has been shortened and the material presented in the manuscript is now clearly summarized towards the end of the abstract. The long sentences were adjusted (broken into multiple sentences) to improve flow of the text. NWP improvement is now only mentioned once in the text.
- The goal of the study should be clarified. While reading the manuscript, I thought that the dataset also included the model outputs, but it is only the field measurements. To improve this, the paragraph starting at line 89, could be divided into 2. One paragraph about the MODF files and a shorter paragraph stating the goal of the manuscript. Also, is a little long and could be shorten. The first few sentences 107-111 seemed to be out of place. Should it be justifying the need to describing such database and, therefore, placed before stating the goal of the study. Additionally, when reading the conclusion, it seems that the authors are doing 2 things: 1) describing a new database and 2) describing a new way to organize/compress data. The goal should probably reflect this because there is a section in the manuscript describing the type of file.
  - This paragraph is now divided into two, as suggested. The first paragraph focuses on the description of the MODFs and their contents. The second paragraph focuses on the goal of the manuscript and the motivation for developing the MODFs. This second paragraph was revised to improve clarity regarding the goal of this study and the purpose of the MODFs. The goals stated in the Introduction (and abstract) now more closely reflect the statements made in the conclusion to help emphasize the goals and achieved outcomes of the study.
- Description of the sites (section 2). I suggestion to use a standard structure for the description of each site. For example, the first sentence of the Whitehorse site is about the platform and instruments while not other site description has this information, at least at the beginning of the section. For each site, 1) there should be a photo of the site with the instruments used, 2) short geographical description, 3) the climatology and 4) other relevant information about the site. Since that the manuscript describe field data, photos of the sites/instruments should come before the grid points used by NWP.
  - The description for each site has been significantly changed to be more systematic, with a standardized structure, as suggested. In the revised version, each site now follows the suggested structure of (1) photos and list of instruments, (2) geographical description, (3) climatology, and (4) other relevant information. Seven new Figures (2-8) containing photos for each site, including the site's topography and individual sensor configurations, are now included prior to the Figure with grid points used by NWP (now Figure 9) with seven new Tables to provide this information for each site.
- Figure 2 should probably be later in manuscript when the authors describe the structure of the datasets in sections 4 and 5 to not confuse the field data and the model data.

- o Figure 2 has been moved to later in the manuscript, as suggested. After the addition of the new requested Figures (photos of the sites and instruments in Figures 2-8), this is now Figure 9 in the revised manuscript.
- The authors should carefully double check the use of full name/acronyms. These are most of the minor comments below.
  - o Acronyms and names have been double-checked and inconsistencies and/or typos have been fixed.
- Line 119: The sentence can start with "To properly..." and "It is important" can be deleted.
  - o The sentence now starts with "to properly" as suggested.
- Line 177: Add DOE in parenthesis after "Department of energy" and one can use the acronym later.
  - o Resolved as suggested
- Line 178: Define NOAA here and use the acronym later.
  - o Resolved as suggested
- Line 193: Use NOAA instead of the full name.
  - o Resolved as suggested
- Line 194: FMI is already defined.
  - o FMI is no longer re-defined, as suggested
- Line 271: I wonder if the author should add a Table to describe these. Then, the reader does not have to download the data to get the information.
  - o A series of new detailed Tables (4-10) is now provided in the manuscript for each site, as suggested by R1. We have noted in the manuscript that the information in these tables is also documented in the attributes of the MODFs themselves.
- Lines 338 and 339: Use "m" instead of "meters".
  - o 'm' is now used instead of 'meters,' as suggested
- Line 340: Add commas between "respectively".
  - o Resolved as suggested
- Line 341: Change "2" for "two".
  - o Resolved as suggested
- Line 347: The beginning of the sentence is awkward. Just start the sentence with "The present phase..." and add "used" after "concept"?
  - o The sentence has been revised, as suggested. It now reads "The present phase of the MODF concept is to use standardized…"
- Line 385: DOE/ARM already defined no need to write the full name.
  - o DOE, ARM, and GML are no longer re-defined, as suggested.
- Line 464: I suggest removing "excellent".
  - o "Excellent" is now removed from the text, as suggested.
- Line 522: Use the acronym because they are already defined.
  - o DOE and ARM are no longer re-defined, as suggested
- Line 549: Delete "see" in front of Figures 3 to 6 and use e.g. instead? One can also delete "as an example".
  - o "see" is now replaced with e.g., and "as an example" has been deleted, as suggested

- Line 550: Delete "see".
  - o "See" is now removed, as suggested.